# On the Fundamental Limits of LLMs at Scale

**Muhammad Ahmed Mohsin**[1], **Muhammad Umer**[1], **Ahsan Bilal**[2], **Zeeshan Memon**[3], **Muhammad Ibtsaam Qadir**[4], **Sagnik Bhattacharya**[1], **Hassan Rizwan**[5], **Abhiram R. Gorle**[1], **Maahe Zehra Kazmi**[6], **Ayesha Mohsin**[7], **Ali Subhan** [8], **Muhammad Usman Rafique**[9], **Zihao He**[10], **Pulkit Mehta**[11], **Jinda Han**[12], **Muhammad Ali Jamshed**[13], **Dean Hougen**[2], **John M. Cioffi**[1]

[1]*Stanford University*   [2]*The University of Oklahoma*   [3]*Emory University*   [4]*Purdue University*   [5]*UC Riverside*
[6]*UC Berkeley*   [7]*National University of Sciences & Technology*   [8]*Universtat Pompeu Fabra*   [9]*Zoox*   [10]*Meta*
[11]*Google DeepMind*   [12]*University of Illinois at Urbana-Champaign*   [13]*University of Glasgow*

**Reviewed on OpenReview:** *https://openreview.net/forum?id=BIRDGVrom8*

## Abstract

Large Language Models (LLMs) have benefited enormously from scaling, yet these gains are bounded by five fundamental limitations: (1) hallucination, (2) context compression, (3) reasoning degradation, (4) retrieval fragility, and (5) multimodal misalignment. While existing surveys describe these phenomena empirically, they lack a rigorous theoretical synthesis connecting them to the foundational limits of computation, information, and learning. This work closes that gap by presenting a unified, proof-informed framework that formalizes the innate theoretical ceilings of LLM scaling. First, computability and uncomputability imply an irreducible residue of error: for any computably enumerable model family, diagonalization guarantees inputs on which some model must fail, and undecidable queries (e.g., halting-style tasks) induce infinite failure sets for all computable predictors. Second, information-theoretic and statistical constraints bound attainable accuracy even on decidable tasks, finite description length enforces compression error, and long-tail factual knowledge requires prohibitive sample complexity. Third, geometric and computational effects compress long contexts far below their nominal size due to positional under-training, encoding attenuation, and softmax crowding. We further show how likelihood-based training favors pattern completion over inference, how retrieval under token limits suffers from semantic drift and coupling noise, and how multimodal scaling inherits shallow cross-modal alignment. Across sections, we pair theorems and empirical evidence to outline where scaling helps, where it saturates, and where it cannot progress, providing both theoretical foundations and practical mitigation paths like bounded-oracle retrieval, positional curricula, and sparse or hierarchical attention.

## 1 Introduction

The past half-decade has witnessed an unprecedented surge in the scale and influence of Large Language Models (LLMs). Parameter counts, training datasets, and compute budgets have all expanded by orders of magnitude, leading to large qualitative shifts in observed behavior across many downstream tasks. Whether such shifts should be interpreted as true emergence, however, depends on the evaluation metric and whether the apparent discontinuity remains robust under alternative scoring rules (Jiang et al., 2024a). For instance, OpenAI's GPT series has grown from 117 million parameters in GPT-1 (Radford et al., 2018) to over a trillion in GPT-4 (OpenAI Achiam et al., 2023): a thousand-fold rise in representational capacity. Empirical *scaling laws* suggest that training loss and downstream performance improve predictably with model size, dataset volume, and compute (Hoffmann et al., 2022a). The transition from GPT-3.5 to GPT-4, for example, achieved a 16-point gain on Measuring Massive Multitask Language Understanding (MMLU) and a 35-point leap on GSM-8K (OpenAI Achiam et al., 2023). These successes have inspired the prevailing belief that scale itself can indefinitely extend intelligence, reducing every failure mode to an engineering obstacle solvable by more data, parameters, or alignment.

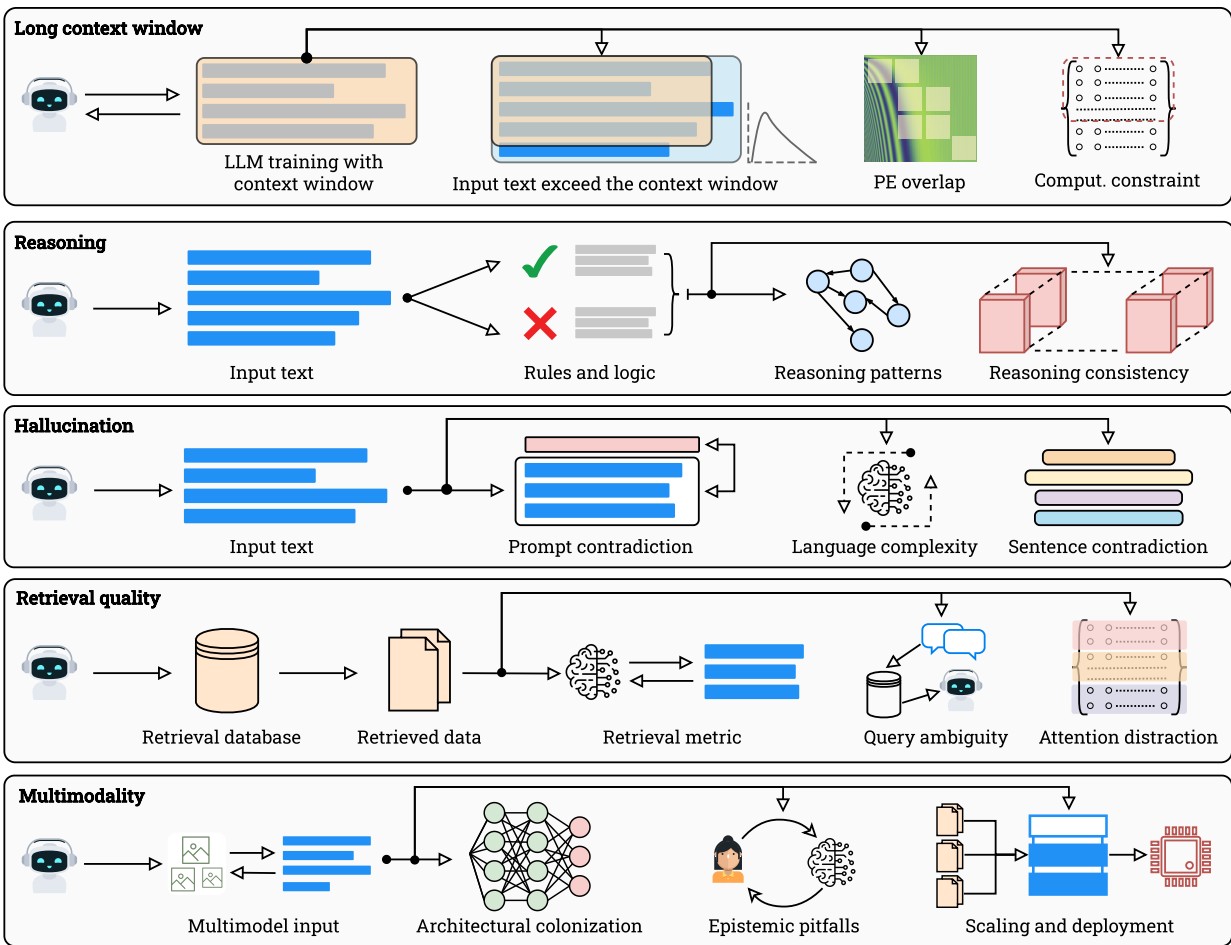

Figure 1: Five interacting fronts that bound LLM reliability. The figure is organizational rather than strictly taxonomic: several failure modes overlap and interact. **Context utilization**: effective use of long inputs is constrained by finite training windows, context extrapolation, positional-encoding degradation, and computational limits. **Reasoning reliability**: rule adherence, compositional generalization, and cross-step consistency remain brittle under complexity and distribution shift. Many failures lie at the intersection of these first two fronts, since long-context reasoning requires both retaining relevant information and manipulating it coherently. **Hallucination**: unsupported content, fabricated details, and contradictions arise from weak grounding or context misintegration. **Retrieval quality**: external evidence is limited by retrieval, ranking, query ambiguity, and integration quality. **Multimodality**: cross-modal inputs introduce added challenges in grounding, alignment, and scalable deployment. Arrows indicate couplings among factors analyzed in later sections.

Yet as models approach trillion-parameter regimes, the very process that powers their ascent also exposes **fundamental limits** that scale cannot surmount. Larger models not only perform better but also *fail more confidently*: they hallucinate, misreason, forget, and misalign in increasingly systematic ways. These pathologies persist even under massive data, suggesting deeper computational and statistical origins. In this paper, we argue that such behaviors are not transient artifacts of optimization or data curation but manifestations of *intrinsic theoretical barriers*, constraints imposed by computability, information theory, and learnability itself. We identify five main limitations that capture distinct failure modes that persist with scaling:

**(1) Hallucination.** LLMs often generate fluent yet fabricated content. Beyond data or alignment flaws (Dziri et al., 2023; Banerjee et al., 2025), we prove hallucination is *inevitable*: diagonalization over enumerable

model classes (Tong et al., 2024b) ensures at least one failure input for every model; uncomputability of problems like the Halting task (Turing et al., 1936) yields infinite failure sets; and finite information capacity and compression bounds (Sahoo et al., 2024) force distortion on complex or rare facts. Thus, no computable LLM can be universally correct over open-ended queries.

**(2) Context compression.** Even with 128K-token windows (Grattafiori et al., 2024), *positional under-training*, *encoding saturation*, and *softmax crowding* (Xiong et al., 2023; Bai et al., 2024a) jointly limit effective context utilization far below its nominal capacity. Gradient decay at rare positions, vanishing sinusoidal/RoPE overlap, and logarithmic score-margin growth show that effective context scales sub-linearly with nominal length.

**(3) Reasoning degradation.** Despite surface fluency, LLMs favor correlation completion over true inference. Likelihood training rewards local coherence, not logical entailment, producing syntactic rather than semantic generalization (Wei et al., 2022). Token-level objectives and lack of explicit reasoning loss drive this systematic "reasoning collapse" out of distribution.

**(4) Retrieval fragility.** Retrieval-augmented models (Lewis et al., 2020a; Borgeaud et al., 2022) inherit theoretical fragilities: bounded token budgets induce semantic drift, ranking noise, and weak coupling between retrieved and generated text. Information-theoretically, as retrieval breadth increases, mutual information with the target decays, imposing an upper limit on factual grounding.

**(5) Multimodal misalignment.** Joint vision–language models suffer cross-modal imbalance, language channels dominate gradients, while visual features under-adapt. Differing modality entropies and misaligned latent manifolds cause perceptual illusions and symbolic confusion, showing that multimodal scaling amplifies rather than removes single-modality brittleness.

Across these axes, we uncover a unifying principle: **LLM failures scale with capability** because they stem from the very theoretical roots that enable language modeling itself. Each failure mode reflects a projection of the same underlying triad: computational undecidability, statistical sample insufficiency, and finite information capacity. Despite extensive empirical documentation, prior surveys (Matarazzo & Torlone, 2025; Kostikova et al., 2025) remain descriptive and lack a formal synthesis connecting these observations to the mathematical foundations of computation and learning. We close this gap through a proof-informed framework that derives a hierarchy of impossibility and saturation results, jointly characterizing when scaling improves, when it plateaus, and when it provably cannot advance. Specifically, we show that no enumerable model class can be universally hallucination-free, as dictated by computability and diagonalization limits. We also demonstrate that finite description length and sample complexity enforce an irreducible generalization error, reflecting the information-theoretic bounds on learnability. Finally, we show that context, reasoning, retrieval, and multimodal grounding each follow identifiable degradation laws determined by architectural constraints and data entropy.

Throughout the paper, theorems and lemmas serve two roles. Some restate well-known limits from computability, information theory, and learning theory in a form tailored to LLM analysis, while others provide unifying formalizations that make explicit assumptions under which commonly observed scaling failures arise. These results should therefore be interpreted as assumption-explicit lenses that organize the surrounding empirical literature rather than universal claims about all LLMs in full generality. Table 1 makes this classification explicit, labeling every formal object in the paper by its epistemic status.

Together, these results reframe scaling not as an unbounded engineering problem but as a process bounded by **intrinsic computational and epistemic constraints**. The remainder of this paper systematically formalizes each limitation, and each section is designed to be self-contained so that readers may engage selectively:

- **Section 2** proves the inevitability of hallucination via a three-tier hierarchy: diagonalization guarantees at least one failure per model, Cantor-pairing–based construction extends this to infinitely many failures per model, and reduction to the Halting Problem shows that undecidable queries force infinite error sets

Table 1: Epistemic status of the formal claims in this paper. We distinguish *definitions*, *classical results* adapted to the LLM setting, *results we prove* under explicitly stated assumptions, *modeling assumptions*, and *empirical observations* drawn from the literature. "Fundamental limit" claims should be read as applying to the first three categories under their stated assumptions, not to the modeling frameworks or empirical trends.

| Status | Claims |
|---|---|
| *Definition* | Hallucination taxonomy (Def. 1); calibrated answer confidence and calibration (§2.3); coverage ratio and temporal staleness (§2.2). |
| *Classical result, adapted* | Diagonalization inevitability (Thms. 1, 2); undecidability-forced error (Thm. 3); Kolmogorov-complexity bottleneck (Lemma 1); VC generalization bound and PAC sample complexity (Thm. 4). |
| *Proved here, under stated assumptions* | Noise floor for atomic facts (Lemma 2); positional undertraining (Lemma 3); sinusoidal/RoPE attenuation (Lemma 4); softmax crowding and $\ln N$ margin (Lemma 5, Cor. 1); incentive property of cost-based grading (Prop. 1). |
| *Modeling assumption / framework* | Creativity–factuality capacity identity (§2.4); improvisation-requires-hallucination argument (§2.4); exposure-bias compounding inequality (§2.2); reasoning-efficiency functional $\eta = \mathbb{E}[Q/C]$ and the unified reasoning objective (§4). |
| *Empirical observation* | Long-tail accuracy collapse (<40% for rare entities); temporal decay (>50% staleness within months); effective context well below nominal length ($\sim$64K of 128K); PAL vs. CoT on GSM8K (§2–4). |

for *all* computable predictors. Statistical and information-theoretic bounds then quantify the residual hallucination risk even on decidable, learnable tasks.

- **Section 3** derives three complementary laws governing effective context length: positional undertraining from left-skewed training distributions (Lemma 3), sinusoidal and RoPE encoding attenuation at long ranges (Lemma 4), and softmax crowding that requires $O(\ln N)$ score margins to maintain attention on relevant tokens (Lemma 5). Together these show that effective context scales sub-linearly with nominal window size.

- **Section 4** analyzes why likelihood-based training produces fluent but logically brittle outputs, formalizes four failure modes (objective mismatch, spurious correlations, search pathology, metric fragility), and presents a unified objective augmenting likelihood with verification and cost regularization. Practical instantiations, i.e., solver-based, prompt-based, and fine-tuning methods, are reviewed as concrete operationalizations.

- **Section 5** dissects how bounded token budgets, semantic drift, and weak coupling between retrieved and generated text impose information-theoretic upper bounds on the factual grounding achievable through retrieval-augmented generation.

- **Section 6** shows how cross-modal entropy imbalance and misaligned latent manifolds cause language channels to dominate gradients, producing perceptual illusions and symbolic confusion that scaling amplifies rather than resolves.

- **Section 7** examines how current evaluation practices interact with the theoretical limits established in earlier sections, highlighting where benchmarks measure genuine capability versus where they obscure fundamental ceilings.

- **Section 8** synthesizes these findings to delineate where further scaling in parameters, data, or modalities yields diminishing or zero returns, and outlines mitigation paths that operate within the identified constraints.

- **Section 9** summarizes the key results and identifies open problems for future work.

## 2 What Makes LLMs Hallucinate?

To ground the subsequent analysis, we first provide a precise definition of hallucination adopted throughout this work.

**Definition 1** (Hallucination)**.** Let $h : \Sigma^* \to \mathcal{Y}$ be a language model mapping input strings to outputs. We distinguish three types of hallucination:

1. **Factual hallucination.** The model output contradicts an external ground-truth function $f : \Sigma^* \to \mathcal{Y}$, i.e., $h(x) \neq f(x)$ for some input $x \in \Sigma^*$.
2. **Faithfulness hallucination.** Given a context $C$ provided at inference time, the model output contradicts or is unsupported by $C$, i.e., $h(x \mid C) \not\models C$.
3. **Intrinsic hallucination.** The model produces outputs that are internally inconsistent, e.g., $h(x) \neq h(x')$ for semantically equivalent inputs $x \simeq x'$, or the model contradicts its own prior statements within a single generation.

The theoretical impossibility results in Section 2 concern *factual hallucination*: they construct ground-truth functions $f$ against which every computable model must err. Data-induced hallucinations (Section 2.2) and evaluation misalignment (Section 2.3) exacerbate all three types, while the creativity-factuality trade-off (Section 2.4) primarily drives factual and faithfulness hallucinations. We note this distinction explicitly because much of the empirical literature conflates these categories, which obscures where theoretical limits apply.

Hallucination in large language models arises from four interconnected sources (Figure 2): (1) Fundamental limits from computability theory, uncomputability, and statistical learning that prove hallucination is mathematically inevitable (Xu et al., 2024e; Banerjee et al., 2025); (2) data-induced hallucinations from incomplete coverage, noise, long-tail distributions, temporal decay, and conflicting information in training corpora (Huang et al., 2025a; Wang et al., 2023); (3) evaluation misalignment where benchmarks reward confident fabrication over calibrated uncertainty (Xu et al., 2024a; Kirichenko et al., 2025); and (4) creativity-factuality trade-offs where mechanisms enabling creative generation necessarily increase hallucination risk (Nguyen et al., 2024; Peeperkorn et al., 2024). Together, these establish hallucination not as a transient engineering problem but as an intrinsic property of probabilistic language models.

## 2.1 Fundamental limits

Hallucination in LLMs is not merely an engineering artifact of insufficient data or suboptimal training; rather, it reflects intrinsic computational and statistical limits that no (present) architecture, scale, or optimization can overcome (Xu et al., 2024e; Banerjee et al., 2025). We formalize these intrinsic boundaries through three complementary lenses: *computability theory*, which shows that no enumerable class of models can correctly answer all computable queries (Peng et al., 2024; Karpowicz, 2025); *uncomputability*, which demonstrates that certain problems lie beyond the reach of any algorithm (Melo et al., 2025); and *statistical learnability*, which reveals that even learnable functions require sample complexity that often exceeds practical limits (Asher et al., 2023; Su, 2025; Khakhar et al., 2023; Goldblum et al., 2023). These results together establish hallucination as an inevitable feature of any learning system operating over open-ended domains (Huang et al., 2025a; Dziri et al., 2023).

**Diagonalization boundary.** We begin with the most fundamental limit: the impossibility of perfect learning for any enumerable collection of models. Modern LLMs, viewed as computable functions mapping input strings to output distributions, belong to computably enumerable sets (e.g., all polynomial-time Turing machines). A classical diagonalization argument originating from Cantor's proof of uncountable infinities and adapted to learning theory reveals that *any* such enumerable collection must hallucinate on some inputs. We formalize this as follows.

**Theorem 1** (Inevitability for enumerable LLMs). *For any computably enumerable set of LLMs $\{h_0, h_1, h_2, \ldots\}$, where each $h_i : \Sigma^* \to \mathcal{Y}$ maps input strings to outputs, there exists a computable ground-truth function $f : \Sigma^* \to \mathcal{Y}$ such that every model state $h_i^{[j]}$ (at training step $j$) hallucinates on at least one input.*

*Proof.* Since both the set of all computable LLMs and the set of all input strings over finite alphabet $\Sigma$ are countable, we can enumerate them: models $\{h_0, h_1, h_2, \ldots\}$ and inputs $\{s_0, s_1, s_2, \ldots\}$. This enumeration is computable.

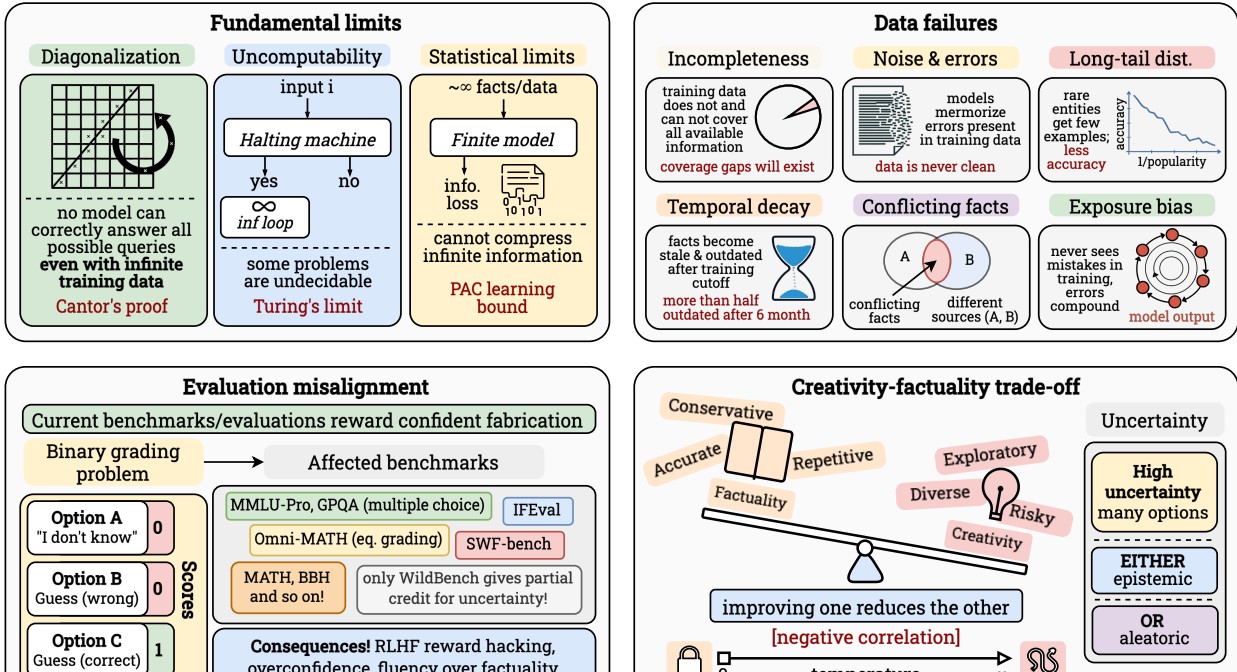

Figure 2: **Taxonomy of hallucination sources in LLMs. (Fundamental limits.)** Diagonalization (no enumerable model set answers all queries), uncomputability (undecidable problems force infinite failures), and statistical constraints (finite models cannot compress infinite information). **(Data failures.)** Incomplete coverage, noise (2–3% error rates), long-tail distributions, temporal decay (>50% staleness after 6 months), conflicts, and exposure bias. **(Evaluation misalignment.)** Binary grading equates uncertainty with wrong answers, incentivizing fabrication across benchmarks {MMLU-Pro, Graduate-Level Google-Proof Q&A (GPQA), MATH}, causing reinforcement learning from human feedback (RLHF) reward hacking and overconfidence. **(Creativity-factuality trade-off.)** Low temperature yields accurate but repetitive outputs; high temperature enables diversity but increases errors.

Construct the ground-truth function $f : \Sigma^* \to \mathcal{Y}$ by diagonalization as follows. For each index $i \in \mathbb{N}$:

$$f(s_i) := \begin{cases} y_{\text{alt}} & \text{if } h_i(s_i) = y_{\text{default}} \\ y_{\text{default}} & \text{otherwise} \end{cases} \tag{1}$$

where $y_{\text{default}}, y_{\text{alt}} \in \mathcal{Y}$ are two distinct outputs. For inputs $s_j$ with $j \neq i$, define $f(s_j)$ arbitrarily (this choice does not affect the argument for model $h_i$).

By construction, $f(s_i) \neq h_i(s_i)$ for all $i$. Since $f$ is defined by case analysis on computable functions (enumeration and $h_i$), $f$ itself is computable. Each model $h_i$ produces incorrect output on input $s_i$, hence hallucinates. This holds for any training state $h_i^{[j]}$ since the enumeration includes all states of all models. This proves that no matter how large or how well-trained an LLM becomes, there will always exist specific queries on which it hallucinates. The adversarial input is constructible for every model architecture and training regime, indicating that hallucination-free LLMs are mathematically impossible. This argument presumes the existence of a ground-truth function that can be defined outside the enumerable set of models, a necessary premise for this proof structure. $\square$

This establishes that at least one adversarial input exists for each model. However, practically, the situation is more broad: hallucination is not an isolated phenomenon but occurs on infinitely many inputs.

**Theorem 2** (Infinite hallucinations). *For any computably enumerable set of LLMs $\{h_0, h_1, \ldots\}$, there exists a computable ground-truth function $f'$ such that each model $h_i$ hallucinates on infinitely many inputs.*

*Proof.* Let $\pi : \mathbb{N} \to \mathbb{N} \times \mathbb{N}$ be a computable bijection (for example, the inverse of the Cantor pairing function $\langle a, b \rangle = \frac{(a+b)(a+b+1)}{2} + a$). For each $k \in \mathbb{N}$, write $\pi(k) = (a_k, b_k)$ and set $i_k := a_k$.

Construct $f' : \Sigma^* \to \mathcal{Y}$ as follows. For each input $s_k$ where $k \in \mathbb{N}$, define:

$$f'(s_k) := \text{flip}(h_{i_k}(s_k)) \tag{2}$$

where $\text{flip}(\cdot)$ returns a different output from its argument (e.g., if output space $\mathcal{Y} = \{y_0, y_1\}$, then $\text{flip}(y_0) = y_1$ and $\text{flip}(y_1) = y_0$).

Since $\pi$ is a bijection onto $\mathbb{N} \times \mathbb{N}$, every pair $(i, b)$ is the image of exactly one $k$. Therefore, for each model index $i$, the set $\{k \in \mathbb{N} : i_k = i\} = \{\pi^{-1}(i, b) : b \in \mathbb{N}\}$ is infinite (containing one element for each $b \in \mathbb{N}$). For each such $k$, we have $f'(s_k) = \text{flip}(h_i(s_k)) \neq h_i(s_k)$ by definition of flip. Since the Cantor pairing, its inverse, and the flip function are all computable, $f'$ is computable. Therefore, each model $h_i$ hallucinates on infinitely many inputs. The situation is worse than isolated failures since each model fails on infinitely many inputs, not just rare edge cases. This establishes hallucination as pervasive rather than exceptional. $\square$

Note that these results are independent of architecture (transformers, RNNs, state-space models), training procedure (supervised, reinforcement learning), or prompt engineering. Even using another LLM to detect and correct hallucinations cannot eliminate them, as the correcting model is itself subject to Theorem 1.

**Uncomputability boundary.** Beyond enumeration-based limitations, certain problems are *undecidable*: no algorithm can solve them for all inputs, regardless of computational resources. The canonical example is the Halting Problem, which asks whether a given computer program will finish running or continue to run forever on a given input. This was proven undecidable by Turing in 1936 (Turing et al., 1936). When an LLM encounters such queries, it faces an impossible dilemma: refusing to answer reveals incompleteness, while attempting an answer inevitably leads to hallucination (Xu et al., 2024e). We formalize this inherent limitation.

**Theorem 3** (Undecidable problems force hallucination)**.** *Let $\Pi$ denote the set of all program-input pairs, and let $f_{\text{halt}} : \Pi \to \{0, 1\}$ be the characteristic function of the Halting Problem (outputting 1 if the program halts, 0 otherwise). For any computable LLM $h : \Pi \to \{0, 1\}$ attempting to approximate $f_{\text{halt}}$, the set $S_h := \{\pi \in \Pi : h(\pi) \neq f_{\text{halt}}(\pi)\}$ of inputs on which $h$ hallucinates is infinite.*

*Proof.* Assume for contradiction that $S_h$ is finite, i.e., $|S_h| = k < \infty$. Then there exists a finite exception set $E := \{(\pi_1, b_1), \ldots, (\pi_k, b_k)\}$ where $b_i = f_{\text{halt}}(\pi_i)$ are the correct answers, and for all $\pi \notin \{\pi_1, \ldots, \pi_k\}$, we have $h(\pi) = f_{\text{halt}}(\pi)$.

Construct a Turing machine $M'$ that decides the Halting Problem:

1. On input $\pi$, check if $\pi \in \{\pi_1, \ldots, \pi_k\}$ (finite check).
2. If yes, output the precomputed correct answer $b_i$ from table $E$.
3. If no, run $h(\pi)$ and output its result.

By assumption, $M'$ correctly decides $f_{\text{halt}}(\pi)$ for all $\pi \in \Pi$. Since $h$ is computable and the table lookup is computable, $M'$ is a computable decider for the Halting Problem. This contradicts the undecidability of the Halting Problem (Turing et al., 1936). Therefore, $S_h$ must be infinite. $\square$

Theorem 3 shows that undecidability creates an insurmountable barrier: no matter how sophisticated an LLM becomes, infinite failures are guaranteed on such problems. Moreover, variants of undecidable problems permeate practical applications: code analysis tasks ("Will this loop terminate?"), logical consistency checking ("Does this axiom set entail a contradiction?"), and self-referential queries ("Generate a sentence you cannot generate") all inherit this fundamental impossibility (Xu et al., 2024e). These are not contrived examples but questions users naturally pose to LLMs, making uncomputability-induced hallucination practically relevant. Yet even when we restrict attention to computable and decidable problems, statistical barriers remain.

**Information-theoretic & statistical limits.** Even for functions that are both *computable* and *learnable in principle*, resource constraints impose hallucination risk. A model with finite descriptive complexity cannot faithfully reproduce arbitrary functions of unbounded complexity without compression-induced distortion. This information-theoretic bottleneck complements the computational barriers above.

**Lemma 1** (Kolmogorov complexity bottleneck). *Let $h$ be an LLM with Kolmogorov complexity $K(h) = c < \infty$. For any $\tau > 0$, there exists a ground-truth function $f$ such that $h$ exhibits hallucination with error exceeding $\tau$ on some input.*

*Proof.* Let $\mathcal{F}_n := \{f : \Sigma^{\leq n} \to \mathcal{Y}\}$ be the set of all functions on inputs of length at most $n$. The cardinality $|\mathcal{F}_n| = |\mathcal{Y}|^{|\Sigma|^{n+1}}$ grows exponentially. The number of functions with Kolmogorov complexity at most $c$ is bounded by $O(2^c)$, since each can be described by a program of length at most $c$.

For sufficiently large $n$, we have $|\mathcal{F}_n| \gg 2^c$. Thus, there exist functions $f \in \mathcal{F}_n$ with $K(f) > c = K(h)$. For such $f$, the model $h$ cannot encode $f$ exactly; by the pigeonhole principle, there must exist inputs where $h$'s output differs from $f$'s output. The fraction of functions with $K(f) > c$ approaches 1 as $n \to \infty$, making hallucination on incompressible functions inevitable. Specifically, if $h$ attempts to approximate a random function $f$ with $K(f) \gg K(h)$, the expected error can be made arbitrarily large by choosing sufficiently complex $f$, exceeding any threshold $\tau$. A model with finite parameters cannot perfectly memorize all facts in an unbounded knowledge domain. Compression is mandatory, and compression introduces errors—particularly on incompressible (random or arbitrary) facts like specific dates, numerical constants, or rare entity attributes. $\square$

Lemma 1 captures the intuition that finite-capacity models must compress, and compression introduces errors on incompressible data. To quantify this limitation in a learning-theoretic framework, we turn to *probably approximately correct (PAC) learning*, which formalizes sample complexity. Let $\mathcal{R}_{\mathrm{hal}}(h)$ denote the hallucination risk, i.e., the probability that $h$ produces a factually incorrect output on a random query drawn from distribution $\mathcal{D}$. The Vapnik-Chervonenkis (VC) dimension is a measure of the complexity or capacity of a class of functions, which reflects its ability to fit diverse patterns. Standard VC-dimension arguments provide generalization bounds:

$$\mathcal{R}_{\mathrm{hal}}(h) \;\leq\; \widehat{\mathcal{R}}_{\mathrm{hal}}(h) \;+\; O\!\left(\sqrt{\frac{d\log(n/d) + \log(1/\delta)}{n}}\right), \tag{3}$$

where $\widehat{\mathcal{R}}_{\mathrm{hal}}(h)$ is the empirical hallucination rate on $n$ training samples, $d$ is the VC dimension of the hypothesis class, and the bound holds with probability at least $1 - \delta$ (Sahoo et al., 2024). For high-capacity models (large $d$) or distributions with long tails (rare facts appearing in $\ll n$ examples), the generalization term remains large even with low training error.

This provides an upper bound on risk given sufficient samples, *but how many samples are required to achieve low hallucination?* For arbitrary facts, say, information with no compressible structure, the answer is prohibitively large.

**Theorem 4** (Sample complexity for arbitrary facts). *Consider a distribution over $m$ independent binary facts, each with a correct answer chosen uniformly at random and independently. To learn a classifier that achieves hallucination probability at most $\epsilon$ across all $m$ facts simultaneously, with confidence at least $1 - \delta$, requires*

$$n \;=\; \Omega\!\left(\frac{m}{\epsilon^2}\log\frac{m}{\delta}\right) \tag{4}$$

*training examples.*

*Proof.* Each of the $m$ facts defines a binary classification problem. Since answers are chosen uniformly and independently, there is no compressible pattern; each fact must be memorized independently. The VC dimension of the hypothesis class capable of representing all $m$ independent facts is at least $m$ (the class can shatter $m$ points).

By the VC inequality, to achieve error at most $\epsilon$ on each fact, we need approximately $O(d/\epsilon^2)$ samples where $d$ is the VC dimension. With $d = m$ and applying a union bound over $m$ facts to achieve simultaneous correctness with confidence $1 - \delta$, we obtain:

$$n \;=\; \Omega\!\left(\frac{m}{\epsilon^2} \cdot \left(\log m + \log \frac{1}{\delta}\right)\right) \;=\; \Omega\!\left(\frac{m}{\epsilon^2} \cdot \log \frac{m}{\delta}\right). \tag{5}$$

For large $m$ (e.g., millions of rare entities, dates, or numerical facts), this bound becomes prohibitive, exceeding the size of any feasible training corpus. $\qquad\square$

Theorem 4 can essentially be translated as: when knowledge lacks structure (e.g., birthdates of millions of individuals, precise numerical constants, arbitrary historical events), sample requirements scale linearly with the number of facts. Real-world corpora, while vast, are finite and contain each rare fact only sparsely, making hallucination on long-tail queries much more statistically likely (Su, 2025).

The PAC framework can be refined further through PAC-Bayesian bounds, which incorporate prior knowledge. Letting $P$ be a prior distribution over models and $Q$ a posterior (concentrated near the trained model), we obtain:

$$\mathbb{E}_{h \sim Q}[\mathcal{R}_{\mathrm{hal}}(h)] \;\leq\; \mathbb{E}_{h \sim Q}[\widehat{\mathcal{R}}_{\mathrm{hal}}(h)] \;+\; \sqrt{\frac{\mathrm{KL}(Q\|P) + \log(2\sqrt{n}/\delta)}{2n}}. \tag{6}$$

The Kullback-Leibler (KL) divergence term $\mathrm{KL}(Q\|P)$ penalizes models that deviate significantly from the prior, capturing a complexity-accuracy tradeoff. While fine-tuning on high-quality factual data can reduce $\widehat{\mathcal{R}}_{\mathrm{hal}}$ and tighten the bound, the sample complexity constraint remains: complete elimination of hallucination over open-ended queries with arbitrary facts is infeasible without exponential data, i.e., a requirement no corpus can satisfy.

Combining the aforementioned results, hallucination emerges from a rigorous three-tier hierarchy, each layer imposing its own constraints:

1. Any enumerable set of models fails on adversarially constructed queries (Theorems 1 and 2), ensuring that no finite or countable collection of LLMs can be universally correct.
2. Undecidable problems (such as the Halting Problem) force infinite-failure sets regardless of model capacity or training data (Theorem 3), making hallucination unavoidable on natural problem classes.
3. Finite model capacity cannot compress infinite-complexity functions without distortion (Lemma 1), and sample complexity for arbitrary facts scales prohibitively (Theorem 4), rendering exhaustive memorization impractical.

Mitigation strategies, including retrieval-augmented generation (oracle access), continual learning (adaptive capacity expansion), and constraint-based decoding can *reduce* hallucination in specific domains but cannot *eliminate* it universally (Béchard & Ayala, 2024). The takeaway is that: hallucination is an intrinsic property of learning systems operating over unbounded, open-ended query spaces, and any deployment of LLMs must account for this irreducible uncertainty.

Having established the inevitability of hallucination from first principles, we now turn to mechanisms that exacerbate this phenomenon in practice.

## 2.2 Data-induced hallucinations

While the preceding analysis establishes that hallucination is theoretically inevitable, training data exacerbates this through systematic imperfections. Even if we could construct an arbitrarily large model with infinite capacity, the *training corpus itself* introduces hallucination pathways that compound the irreducible baseline. We examine how data incompleteness, quality degradation, distributional skew, temporal decay, and internal conflicts create fertile ground for factual errors that persist even in well-trained models.

**Incompleteness.** No training corpus, regardless of size, can encode all knowledge. The set of facts about the world grows continuously, while training datasets represent finite snapshots. Even static domains suffer from coverage gaps; i.e., rare entities, niche topics, and low-resource languages receive sparse representation,

forcing models to interpolate or guess when queried about underrepresented content (Onoe et al., 2022; Mousavi et al., 2024; Cheng et al., 2024).

Formally, let $\mathcal{K}_{\text{world}}$ denote the set of all factual propositions and $\mathcal{K}_{\text{train}} \subset \mathcal{K}_{\text{world}}$ the subset present in the training data. The *knowledge gap* $\mathcal{K}_{\text{world}} \setminus \mathcal{K}_{\text{train}}$ is necessarily non-empty and, in practice, vast. Define the *coverage ratio* as:

$$\rho_{\text{cov}} := \frac{|\mathcal{K}_{\text{train}}|}{|\mathcal{K}_{\text{world}}|} \in [0, 1], \tag{7}$$

which, for any finite corpus and unbounded knowledge domain, satisfies $\rho_{\text{cov}} \to 0$ as $|\mathcal{K}_{\text{world}}| \to \infty$. Queries targeting the knowledge gap force the model to extrapolate from seen patterns to unseen facts, a process prone to plausible-sounding fabrications (Wang et al., 2023). Further, let $q \in \mathcal{K}_{\text{world}} \setminus \mathcal{K}_{\text{train}}$ be a query about missing knowledge. The model generates an answer by maximizing:

$$\hat{y} = \arg\max_{y \in \mathcal{Y}} p_\theta(y \mid q, \mathcal{K}_{\text{train}}), \tag{8}$$

where $p_\theta$ is the learned distribution. Since $q \notin \mathcal{K}_{\text{train}}$, the model relies on spurious correlations or superficial pattern matching, resulting in a high hallucination probability:

$$\mathbb{P}[\hat{y} \neq y^* \mid q \notin \mathcal{K}_{\text{train}}] \gg \mathbb{P}[\hat{y} \neq y^* \mid q \in \mathcal{K}_{\text{train}}], \tag{9}$$

where $y^*$ is the true answer. Note that this is quite distinct from the theoretical limits discussed earlier: even for computable, decidable facts, absence from training data makes hallucination empirically likely.

**Noise, errors, and misinformation.** Training corpora are typically scraped from the internet or aggregated from diverse sources and naturally contain substantial noise, factual errors, and deliberate misinformation (Sahoo et al., 2024). Web text includes unverified claims, outdated information, satirical content misinterpreted as factual, and adversarial misinformation. When LLMs ingest such data, they learn both correct and incorrect associations with no inherent process to distinguish truth from falsehood during pretraining.

Let $\mathcal{D}_{\text{train}} = \{(x_i, y_i)\}_{i=1}^n$ be the training dataset, and let $\eta \in [0, \frac{1}{2})$ denote the *noise rate*, i.e., the fraction of examples with incorrect labels or factual errors. The effect of this noise on hallucination depends critically on whether the underlying knowledge has exploitable structure. When the clean labeling admits low-complexity structure, noise-robust learning can average over many correlated examples to recover the clean decision rule, and the risk converges to the noise-free Bayes rate as $n \to \infty$ (Angluin & Laird, 1988; Natarajan et al., 2013). Atomic factual knowledge provides no such signal: isolated, incompressible facts (specific dates, numerical constants, rare entity attributes) cannot be denoised by pooling, so the corpus noise rate propagates directly into factual error. We formalize this regime below.

**Lemma 2** (Noise floor for atomic facts). *Let $\{f_1, \ldots, f_m\}$ be atomic facts such that (i) the facts are mutually independent and incompressible, so observations of $f_j$ carry no information about $f_i$ for $j \neq i$; and (ii) each $f_i$ appears $r_i \geq 1$ times, every appearance independently mislabeled with probability $\eta \in [0, \frac{1}{2})$. Then for any estimator, the expected hallucination probability on $f_i$ obeys*

$$\mathbb{E}[\mathcal{R}_{\text{hal}}(h); f_i] \geq \beta(r_i, \eta) := \Pr[\text{Bin}(r_i, \eta) \geq \lceil r_i/2 \rceil].$$

*In particular $\beta(1, \eta) = \eta$, and this floor is independent of the total corpus size $n = \sum_i r_i$: enlarging $n$ by adding* new *facts leaves the error on each existing fact unchanged.*

*Proof.* By (i), the only evidence about $f_i$'s label is its $r_i$ noisy observations, which by (ii) are i.i.d. Bernoulli($\eta$)-corrupted copies of the truth. Any estimator is a function of these observations; under a uniform prior the Bayes-optimal decision is the majority vote, with error $\Pr[\text{Bin}(r_i, \eta) \geq \lceil r_i/2 \rceil]$, giving the bound. For $r_i = 1$ this is exactly $\eta$. Independence of $n$ follows since, by (i), observations of $f_j$ ($j \neq i$) are uninformative about $f_i$. □

Lemma 2 isolates where label noise is irreducible: singly-attested atomic facts inherit the full noise rate $\eta$, and additional unrelated data cannot remove it. This floor is specific to atomic memorization and does *not*

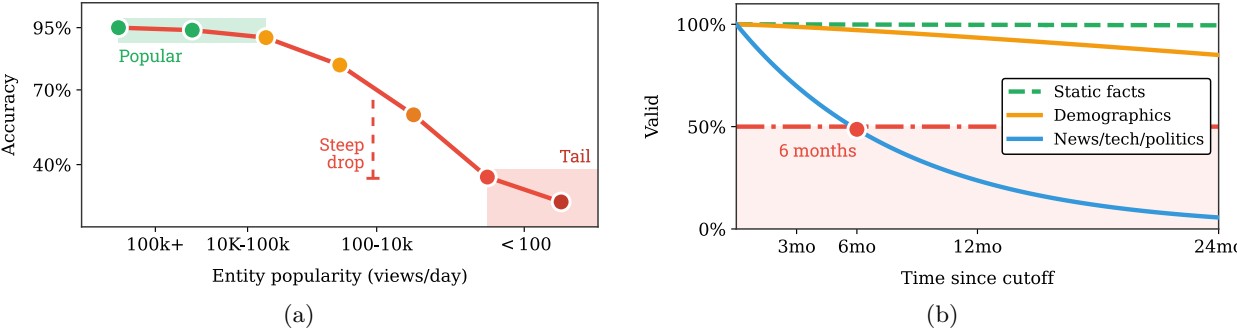

Figure 3: **Empirical evidence of data-induced hallucinations. (a)** Model accuracy exhibits a steep degradation for rare entities, dropping from >95% for highly popular entities (100k+ Wikipedia views/day) to <40% for tail entities (<100 views/day). **(b)** Information validity decays over time since the training cutoff. While static facts remain valid indefinitely and demographics change slowly, rapidly evolving domains cross the 50% validity threshold within 6 months, causing temporally induced hallucinations as models lack explicit temporal reasoning and treat all training data as contemporaneous.

extend to structured prediction, where consistent noise-robust learning applies. In this atomic-fact regime, studies show common web scrapes contain 2–3% demonstrably false factual claims ($\eta \approx 0.02$–$0.03$), conspiracy theories appear with non-negligible frequency, and even curated datasets like Wikipedia exhibit temporal inconsistencies and edit wars that encode conflicting information (Zhang et al., 2025d). The model's objective of "next-token prediction" treats all training text equally, optimizing likelihood rather than veracity:

$$\mathcal{L}(\theta) = -\sum_{i=1}^{n} \log p_\theta(y_i \mid x_i),\tag{10}$$

without regard to whether $y_i$ is factually correct. Consequently, frequently repeated misinformation can dominate the learned distribution, leading models to confidently reproduce falsehoods (Dufour et al., 2024).

**Long-tail distribution and memorization failures.** Real-world knowledge follows a heavily skewed distribution; a small number of entities and facts appear frequently, such as major historical figures or popular scientific concepts, while the vast majority occur rarely. This *long-tail phenomenon* creates a memorization challenge: models must store millions of low-frequency facts with minimal reinforcement (Sahoo et al., 2024).

Empirical evidence shows that LLM factual accuracy degrades sharply for tail entities (Figure 3a). For instance, when asked about individuals with fewer than 10 Wikipedia page views per day, GPT-4's factual precision drops below 40%, compared to >90% for highly popular entities (Kandpal et al., 2023). The model has insufficient exposure to reliably encode these facts and instead generates plausible-sounding but incorrect details by analogy to more common patterns. This aligns with what Theorem 4 states, which in this context implies that without sufficient training examples for each rare fact, generalization bounds guarantee high error rates.

The memorization challenge is further complicated by *interference*: the model's finite capacity means that learning new facts can overwrite or distort previously learned information, a phenomenon known as catastrophic forgetting in continual learning (Gekhman et al., 2024). As models grow and training data expands, managing this interference becomes increasingly difficult, with rare facts being the most vulnerable to degradation.

**Temporal decay.** Training data represents a temporal snapshot, but the world evolves continuously. Let $t_{\text{cutoff}}$ denote the training data cutoff time and $t_{\text{query}}$ the query time. Facts that were true at $t_{\text{cutoff}}$ may become outdated by $t_{\text{query}}$. For example, political leaders change, scientific consensus shifts, companies merge or dissolve, and technologies become obsolete (Zhu et al., 2024a). Define the *temporal staleness* of a fact $f$ as:

$$\tau(f) := \mathbb{P}[f \text{ is outdated at } t_{\text{query}} \mid f \text{ was true at } t_{\text{cutoff}}],\tag{11}$$

which increases with $\Delta t := t_{\text{query}} - t_{\text{cutoff}}$. For time-sensitive domains, empirical studies show $\tau(f)$ can exceed 0.5 within months, i.e., over half of time-dependent facts become stale (Figure 3b) (Lazaridou et al., 2021). Models trained on pre-2023 data confidently assert information valid only in that historical context, producing *temporally induced hallucinations* when queried about current events (Zhu et al., 2024b).

This issue is particularly worse for rapidly evolving domains. A model queried about "the current treatment for disease X" may generate a response reflecting outdated clinical guidelines from its training distribution, presenting obsolete information as current fact (Lazaridou et al., 2021). Unlike incompleteness (which can be addressed via knowledge expansion), temporal decay may require continuous model updates. This presents a difficult trade-off: avoiding the "catastrophic forgetting" of past knowledge is important, yet retaining outdated information leads to temporal hallucinations. This process is costly and technically challenging, and thus requires a careful balance between preserving and updating information.

Worse, models lack explicit temporal reasoning. They cannot reliably distinguish between timeless facts (e.g., mathematical theorems, $\tau(f) \approx 0$) and time-dependent facts (e.g., "the president of the United States", $\tau(f) \gg 0$). When training data contains multiple temporal versions of a fact without clear timestamp annotations, the model may conflate them, producing anachronistic or contradictory outputs (Wang et al., 2023).

**Conflicting information.** In training corpora, different sources disagree on contested facts, present conflicting interpretations of events, or reflect divergent cultural perspectives, and thus, inherent contradictions arise. When such conflicts are unresolved in training, the model learns a superposition of contradictory beliefs, surfacing whichever aligns with the prompt's framing or the model's implicit biases (Huang et al., 2025a).

For example, politically contentious topics have multiple narratives in web text. An LLM trained on this mixture may generate different "facts" depending on subtle prompt variations, revealing that it has not learned a single coherent world model but rather a distribution over conflicting models (Zhang et al., 2025d). Unlike simple factual errors, the model has learned *both* correct and incorrect information, and query-time stochasticity determines which surfaces.

Additionally, systematic biases in training data—gender stereotypes, racial prejudices, and the perspectives of the dominant cultures in the training data—become encoded in model parameters (Gallegos et al., 2024). When generating content about underrepresented groups or non-Western contexts, models often fall back on stereotypes or fabricate details consistent with biased training patterns. This produces culturally skewed hallucinations that reflect and amplify societal biases present in the data.

**Exposure bias and distributional mismatch.** A subtle but critical issue arises from the discrepancy between training and deployment distributions. During training, models see gold-standard prefixes $x_{<t}$ from the corpus and predict the next tokens. At inference, they generate text autoregressively, feeding their own predictions $\hat{x}_{<t}$ back as input. Let $p_{\text{data}}(x_{<t})$ be the training prefix distribution and $p_{\text{model}}(x_{<t})$ be the distribution induced by autoregressive generation. This *exposure bias* means the model never experiences its own errors during training, leading to compounding mistakes at test time (Wang & Sennrich, 2020). The distributional shift can be quantified via the KL divergence:

$$\Delta_{\text{shift}}(t) := \text{KL}(p_{\text{model}}(\cdot \mid x_{<t}) \| p_{\text{data}}(\cdot \mid x_{<t})), \tag{12}$$

which grows with generation length $t$ as errors accumulate.

Concretely, suppose the model makes a small factual error at step $t_0$, i.e., $\hat{x}_{t_0} \neq x_{t_0}^*$. This error shifts the context distribution away from the training distribution, increasing the likelihood of subsequent errors. The error probability at step $t > t_0$ satisfies:

$$\mathbb{P}[\hat{x}_t \neq x_t^* \mid \hat{x}_{t_0} \neq x_{t_0}^*] \geq \mathbb{P}[\hat{x}_t \neq x_t^*] + \Delta_{\text{shift}}(t - t_0), \tag{13}$$

where $\Delta_{\text{shift}}(t - t_0)$ quantifies the compounding effect. Over long generations, these errors accumulate—a phenomenon observed in tasks like multi-hop reasoning and long-form summarization, where factual accuracy degrades as output length increases (Wang & Sennrich, 2020). The model has no mechanism to detect or correct these drifts, as it was never trained on its own potentially erroneous outputs.

**Data contamination and leakage.** Data contamination is not binary but a spectrum, ranging from exact duplication to paraphrastic or semantic overlap with benchmark data. Its effect on measured performance depends on the degree of overlap, whether the model memorizes benchmark formats, and whether the task rewards pattern recall rather than true generalization. Consequently, contamination should be treated as a measurable risk rather than an inevitable artifact of scale, and can be audited or mitigated through techniques such as temporal holdouts, deduplication, canary strings, overlap detection, and perturbation-based retesting (Li et al., 2024c; Jiang et al., 2025).

Related issues include *data leakage* from private or copyrighted sources, where models reproduce verbatim or near-verbatim text from training data, and *model-generated data in training corpora*, where synthetic text produced by earlier LLMs contaminates datasets for subsequent models. This latter issue, termed "model collapse" in some literature, can amplify hallucination patterns; if an earlier model's factual errors are treated as ground truth in a later model's training data, those errors become entrenched and harder to correct (Shumailov et al., 2023).

All these failures are not independent: long-tail facts are more likely to be noisy or outdated, conflicting information is harder to verify for rare entities, and exposure bias amplifies any underlying data quality issue. The compounding effect means that data-induced hallucinations are often more severe in practice than theoretical limits alone would predict. Addressing these issues requires improved data curation, architectural innovations, and deployment-time verification, among other things. However, as with the fundamental limits, complete elimination is infeasible. Data will always be incomplete, imperfect, and outdated relative to the unbounded query space LLMs confront.

## 2.3 Evaluation misalignment and guessing incentives

While fundamental limits and data imperfections guarantee some baseline hallucination rate, evaluation practices and training incentives can systematically *amplify* this problem by rewarding confident fabrication over honest uncertainty. Modern LLM benchmarks, leaderboards, and reward models create perverse incentives that penalize abstention and reward guessing even when the model lacks knowledge. This misalignment between evaluation metrics and deployment desiderata transforms hallucination from an unavoidable failure into a sort of "rational" strategy for maximizing scores.

**The penalty for "I don't know."** The vast majority of present-day LLM evaluations employ binary grading. Responses are scored as either correct (1) or incorrect (0), with no partial credit for expressing uncertainty (Xu et al., 2024a). Under such schemes, abstentions, which can be responses like "I don't know," "I'm uncertain," or "I cannot answer without more information", receive the same zero score as confident but incorrect answers. This creates a rational incentive structure that strictly favors guessing over abstention.

Formally, consider a prompt $c$ with response space $\mathcal{R}_c$ and abstention responses $\mathcal{A}_c \subset \mathcal{R}_c$ (e.g., "I don't know"). A grader $g_c : \mathcal{R}_c \to \{0, 1\}$ is *binary* if $g_c(r) = 0$ for all $r \in \mathcal{A}_c$ and $g_c(r) = 1$ for some correct $r \notin \mathcal{A}_c$. For any belief distribution $\rho_c$ over which responses are correct, the expected score of an abstention is always zero, while even a low-confidence guess has positive expected score if the model assigns any non-zero probability to being correct. The optimal strategy under binary grading is thus to *never* abstain:

$$\mathcal{A}_c \cap \arg \max_{r \in \mathcal{R}_c} \mathbb{E}_{g_c \sim \rho_c}[g_c(r)] = \emptyset. \tag{14}$$

A meta-analysis of influential benchmarks confirms this pervasive issue (Yang et al., 2023). Among the most widely cited evaluations, namely MMLU-Pro, GPQA, Omni-MATH, IFEval, SWE-bench, MATH, BBH, HLE, all employ binary grading with no credit for uncertainty expressions (Li et al., 2023c). MMLU-Pro and GPQA are standard multiple-choice exams with no "I don't know" option. Omni-MATH uses equivalence grading (often via LM judges) that compares outputs to ground-truth answers, assigning full credit for correctness and zero otherwise. IFEval grades instruction-following compliance programmatically, again binarized. Only WildBench, which evaluates real user chats on a 10-point scale, offers *partial* credit for uncertainty, but even there, the rubric suggests that "I don't know" responses score lower than "fair" responses with minor hallucinations, thus still incentivizing guessing (Kirichenko et al., 2025).

**Reward hacking and confident fabrication.** When models are trained via reinforcement learning from human feedback (RLHF) or direct preference optimization (DPO), the reward model inherits biases from human annotators who often prefer confident, fluent answers over hedged or uncertain ones (Xu et al., 2024a). Even when a model is unsure, producing a plausible-sounding fabrication receives higher reward than admitting ignorance. This creates a *reward hacking* dynamic, wherein models learn to maximize perceived confidence and fluency, which correlates with reward, rather than factual accuracy.

Let $R(r \mid c)$ denote the reward for response $r$ to prompt $c$. A model optimizing $\mathbb{E}[R(r \mid c)]$ learns an optimal policy, $\pi^*$, according to the reward function:

$$\pi^*(r \mid c) \ \propto \ \exp\big(\beta \cdot R(r \mid c)\big), \tag{15}$$

where $\beta$ is the inverse temperature. If $R$ rewards verbosity and confidence over accuracy, the learned policy $\pi^*$ shifts toward hallucination-prone policies that generate detailed fabrications. This phenomenon is exacerbated by *LM-as-judge* evaluation, where another LLM grades outputs. LM judges are susceptible to length bias, favoring longer responses even when they contain errors, and often fail to detect subtle factual inaccuracies, grading incorrect but fluent hallucinations as correct (Yao et al., 2024).

**Overconfidence and calibration failure.** Related to reward hacking is the issue of *miscalibration*: models systematically overestimate their correctness probability. Let $p_\theta(c)$ denote the model's self-assessed confidence that its answer to prompt $c$ is correct, and let $\mathbb{P}[\text{correct} \mid p_\theta(c)]$ be the true accuracy conditioned on that confidence level. A well-calibrated model satisfies:

$$\mathbb{P}[\text{correct} \mid p_\theta(c) = p] \ = \ p \quad \forall p \in [0,1]. \tag{16}$$

In practice, LLMs exhibit significant *overconfidence*. They assign high probabilities to incorrect answers, i.e., $\mathbb{P}[\text{correct} \mid p_\theta(c) = 0.9] \ll 0.9$. This miscalibration is particularly severe on long-tail queries and out-of-distribution inputs, where the model has minimal training signal (Sainz et al., 2023).

Overconfidence arises from multiple sources: softmax temperature tuning during fine-tuning often sharpens distributions to increase apparent certainty; RLHF rewards confident responses regardless of correctness; and models lack mechanisms to estimate epistemic uncertainty (what they don't know) versus aleatoric uncertainty (inherent ambiguity). The result is that when a model hallucinates, it does so *confidently*, providing no signal to users or downstream systems that the output is unreliable.

**Illusion of competence.** As already discussed previously, benchmark contamination creates an illusion of reduced hallucination. Models memorize test answers rather than learning generalizable reasoning, scoring well on benchmarks while failing on novel queries. Let $\zeta$ denote the contamination fraction and $\mathcal{R}_{\text{true}}(h)$ the true hallucination rate on uncontaminated data. Observed benchmark performance satisfies:

$$\mathcal{R}_{\text{obs}}(h) \ = \ (1 - \zeta) \cdot \mathcal{R}_{\text{true}}(h) \ + \ \zeta \cdot \mathcal{R}_{\text{mem}}(h), \tag{17}$$

where $\mathcal{R}_{\text{mem}}(h) \ll \mathcal{R}_{\text{true}}(h)$ reflects near-perfect recall of memorized answers. This distorts leaderboard rankings: models with higher contamination rates appear to hallucinate less, while their real-world performance remains poor (Xu et al., 2024a).

**Fluency vs. factuality.** The fundamental training objective, i.e., next-token likelihood maximization, prioritizes fluency and coherence over factual correctness. The loss function:

$$\mathcal{L}(\theta) = -\sum_{t=1}^{T} \log p_\theta(x_t \mid x_{<t}) \tag{18}$$

measures how well the model predicts text continuations from the training distribution, not whether those continuations are factually accurate. A model can achieve low perplexity by learning stylistic patterns, grammatical structures, and common narrative arcs, all while encoding factual errors. Again, when training and evaluation metrics both reward fluency, hallucination becomes a rational strategy: fabricating plausible details improves likelihood while remaining unpunished.

Language modeling treats all tokens equally, whether they convey facts ("The capital of France is Paris") or stylistic filler ("In this essay, I will argue that..."), whereas factuality requires distinguishing knowledge-bearing content from discourse markers. Without explicit factuality signals in the training objective or evaluation metrics, models default to the easier-to-optimize goal of producing text that *sounds right* rather than text that *is right*.

**Toward aligned evaluation.** Addressing evaluation misalignment requires major socio-technical reforms (Kirichenko et al., 2025). First, *confidence-aware grading* should reward models for answering when likely correct and abstaining when not, using a scoring rule whose incentives are explicit rather than ad hoc. Let a model, queried with $c$, either return an answer $\hat{y}$ or abstain, and define its *confidence* $p := \Pr_{\text{model}}[\hat{y} \text{ correct} \mid c]$; the model is *calibrated* if $\Pr[\hat{y} \text{ correct} \mid p = \pi] = \pi$ for all $\pi \in [0,1]$. Fix a wrong-answer penalty $\kappa > 0$ and grade

$$g_c(r) = \begin{cases} 1 & \text{if } r \text{ is a correct answer,} \\ -\kappa & \text{if } r \text{ is an incorrect answer,} \\ 0 & \text{if } r \text{ is an abstention.} \end{cases} \tag{19}$$

The abstention payoff is fixed at 0 and does not depend on any reported confidence, so it cannot be inflated by over-reporting $p$.

**Proposition 1** (Incentive property). *Under equation 19, a calibrated model maximizes expected score by answering if and only if $p > t := \kappa/(1 + \kappa)$, and abstaining otherwise.*

*Proof.* Answering yields expected score $p \cdot 1 + (1 - p)(-\kappa) = (1 + \kappa)p - \kappa$, while abstaining yields 0. Answering dominates iff $(1 + \kappa)p - \kappa > 0$, i.e. $p > \kappa/(1 + \kappa)$. □

The threshold $t$ is set by the domain's error cost: high-stakes tasks take large $\kappa$ (hence high $t$, encouraging abstention unless very confident), while low-stakes tasks take small $\kappa$. Binary grading is the degenerate case $\kappa = 0$, giving $t = 0$, so answering always dominates—precisely the guessing incentive derived. To additionally certify calibration, the reported $p$ can be elicited with a strictly proper scoring rule, e.g. the Brier reward $S(p, Y) = 1 - (Y - p)^2$ for the correctness indicator $Y \in \{0, 1\}$, whose expectation is uniquely maximized by truthful reporting $p = \Pr[Y = 1]$ (Gneiting & Raftery, 2007). Second, *explicit confidence thresholds* should be stated in evaluation instructions, specifying the error tolerance $t$ for each task domain. Third, reward models should be trained to *value honesty over confidence*, penalizing overconfident incorrect answers more than hedged uncertain ones. Finally, benchmark contamination must be actively monitored via canary tokens, adversarial test sets, and temporal segregation of training and evaluation data. Without these changes, evaluation practices will continue to amplify hallucination by rewarding exactly the behaviors that users and applications most want to avoid.

## 2.4 Creativity-factuality trade-off

*Creativity* and *factuality* are fundamentally at odds in probabilistic language generation. Unfortunately, the same mechanisms that enable LLMs to produce novel, engaging, and imaginative text, i.e., exploration of low-probability continuations, deviation from training patterns, and compositional recombination, also create pathways for fabrication.

**Exploration-exploitation dilemma.** LLMs generate text by sampling from a learned probability distribution $p_\theta(x_t \mid x_{<t})$ over next tokens. The sampling strategy governs the balance between *exploitation* (selecting high-probability, safe continuations) and *exploration* (venturing into lower-probability, generative territory). This trade-off is controlled by hyperparameters like temperature $T$ and nucleus (top-$p$) sampling (Peeperkorn et al., 2024).

With temperature scaling, the next-token distribution becomes:

$$p_T(x_t \mid x_{<t}) = \frac{\exp(z_t/T)}{\sum_{v \in \mathcal{V}} \exp(z_v/T)}, \tag{20}$$

where $z_t$ is the logit for token $x_t$. As $T \to 0$, sampling becomes deterministic (greedy decoding), favoring the single most likely token. As $T \to \infty$, the distribution flattens toward uniform (high exploration, minimal exploitation). Empirically, low temperatures ($T \approx 0.1\text{-}0.5$) produce repetitive, conservative text with high factual accuracy but limited novelty. High temperatures ($T \approx 1.0\text{-}2.0$) yield diverse, creative outputs but also frequent hallucinations as the model samples unlikely tokens that lead to fabricated details (Nguyen et al., 2024).

Nucleus or top-$p$ sampling restricts sampling to the smallest set of tokens whose cumulative probability exceeds $p$:

$$\mathcal{V}_p := \arg\min_{\mathcal{V}' \subseteq \mathcal{V}} \left\{ |\mathcal{V}'| \;\Big|\; \sum_{v \in \mathcal{V}'} p_\theta(v \mid x_{<t}) \geq p \right\}. \tag{21}$$

Smaller $p$ (e.g., 0.7) enforces exploitation; larger $p$ (e.g., 0.95) permits exploration. The choice of $T$ and $p$ directly controls the creativity-factuality trade-off, and increasing either parameter boosts creative diversity at the cost of factual reliability.

**Entropy, uncertainty, and hallucination.** A model's predictive uncertainty, measured by entropy, correlates with both creativity and hallucination risk. Define the entropy of the next-token distribution as:

$$H(x_t \mid x_{<t}) = -\sum_{v \in \mathcal{V}} p_\theta(v \mid x_{<t}) \log p_\theta(v \mid x_{<t}). \tag{22}$$

High entropy ($H \gg 0$) indicates the model is uncertain about the next token, with probability mass spread across many candidates. This uncertainty can arise from two sources: *epistemic uncertainty*, which occurs when the model lacks knowledge about the domain, or *aleatoric uncertainty*, which arises when the next token is inherently ambiguous given the context.

When epistemic uncertainty is high, e.g., when the model is queried about rare facts or out-of-distribution topics, high-entropy sampling increases hallucination risk, as the model explores tokens it has weak evidence for, leading to fabrications. Conversely, when aleatoric uncertainty is high, as in creative writing where multiple valid continuations exist, high-entropy sampling is desirable. In this case, the model produces diverse, imaginative outputs without sacrificing correctness (since no single answer is uniquely correct).

The challenge is distinguishing these cases. Models lack explicit mechanisms to estimate epistemic vs. aleatoric uncertainty, so sampling strategies cannot adapt. A fixed high temperature boosts creativity but also hallucination, while a fixed low temperature reduces hallucination but also stifles creativity (Farquhar et al., 2024).

**Accuracy vs. originality.** The creativity-factuality trade-off can be formalized as an optimization problem with competing objectives. Specifically, let $\mathcal{A}(\theta)$ denote a factuality (accuracy) metric which measures how often the model's outputs align with ground truth, and let $\mathcal{C}(\theta)$ be a creativity metric measuring diversity, novelty, or originality. A natural framework posits a constrained capacity:

$$\mathcal{A}(\theta) + \alpha \cdot \mathcal{C}(\theta) = \kappa, \tag{23}$$

where $\alpha > 0$ weights creativity relative to accuracy, and $\kappa$ represents total model capacity. Taking the differential:

$$d\mathcal{A} = -\alpha \cdot d\mathcal{C}, \tag{24}$$

showing a negative correlation: improving creativity ($d\mathcal{C} > 0$) necessitates reducing accuracy ($d\mathcal{A} < 0$), and vice versa (Nguyen et al., 2024).

Empirically, this trade-off manifests in multiple ways. Models fine-tuned for factuality (e.g., via retrieval-augmented generation or fact-checking objectives) exhibit lower perplexity on factual QA but higher perplexity on open-ended creative tasks. Conversely, models optimized for dialogue engagement or storytelling generate more varied, entertaining responses but make more factual errors. The $\alpha$ parameter in the trade-off reflects task requirements; for instance, medical diagnosis or legal advice demand high $\mathcal{A}$ (low $\alpha$), while fiction writing or brainstorming favors high $\mathcal{C}$ (high $\alpha$).

**Improvisation requires hallucination.** A provocative theoretical perspective argues that hallucination is *necessary* for improvisation in LLMs (Jiang et al., 2024b). If a model were constrained to produce only outputs directly supported by its training data, it could never generate truly novel text because every sequence then would be a recombination of observed patterns. Creativity demands extending beyond the training distribution, which by definition involves assigning non-zero probability to unseen or low-probability continuations. These exploratory samples constitute hallucinations when they introduce factual errors, but they are indispensable for creative generation. Formally, let $\mathcal{D}_{\text{train}}$ denote the support of the training distribution. A model that generates only from $\mathcal{D}_{\text{train}}$ produces zero hallucinations but also zero novel outputs. To improvise, the model must sample from regions where $p_{\text{train}}(x) \approx 0$ but $p_\theta(x) > 0$. Whether such samples are creative or hallucinatory depends on the task: in fiction, they are celebrated as originality; in fact-based QA, they are condemned as fabrications. The line between creativity and hallucination blurs, with both stemming from the model's learned ability to extrapolate beyond its training data.

**Practical implications.** The creativity-factuality trade-off implies that *no single model configuration is optimal for all tasks.* Systems requiring high factual precision should use low-temperature, retrieval-augmented generation with strict verification. Tasks valuing creativity benefit from high-temperature sampling with relaxed factuality constraints. Attempting to optimize both objectives simultaneously, i.e., a "do-everything" model, results in suboptimal performance on both, one that is too conservative for creativity and too error-prone for factuality.

Ultimately, the creativity-factuality trade-off reflects a fundamental tension in generative AI: the mechanisms enabling models to produce engaging, original, human-like text are the same mechanisms that produce plausible but false content. Recognizing this inherent duality shifts the question from "How do we eliminate hallucinations?" to "How do we deploy LLMs in ways that use creativity where beneficial and enforce factuality where critical?"

## 3 How Far Do LLMs Really Look Into Context?

### 3.1 Problem description

Transformer-based LLMs have pushed context lengths into the tens or even hundreds of thousands of tokens, enabling applications like book summarization, multi-document QA, and long conversation memory (Vaswani et al., 2017; Beltagy et al., 2020; Zaheer et al., 2020; OpenAI Achiam et al., 2023; Grattafiori et al., 2024; Liu et al., 2024a; Tomczak & Kuppannagari, 2025). Yet despite larger context windows, practical long-context reasoning often falls short of expectations (Qiu et al., 2020; Han et al., 2021). Models frequently cannot effectively utilize the full window. The *effective* usable context of many LLMs is much shorter than their nominal context length (Wang et al., 2024c; Liu et al., 2023c; Wang et al., 2024c; Huang et al., 2023). Even a 70B model trained to 128K context (Llama3.1) was found to only leverage about 64K effectively (Grattafiori et al., 2024; Wang et al., 2024c) and in a comprehensive benchmark (LongBench) with average inputs 6.7K words, even a strong 16K-token model (GPT-3.5-Turbo-16k) *still struggles on longer contexts* (Radford et al., 2019; Brown et al., 2020; Wei et al., 2022). Clearly, simply extending the context window does not guarantee strong reasoning across that entire length. This review explores the reason why. We focus on three core factors (as shown in Figure 5), including *training data positional distribution, positional encoding limits, and attention computation constraints*, and explain how each fundamentally hinders long-range reasoning (Wang et al., 2024c; Huang et al., 2023).

### 3.2 Left-skewed training distribution and undertraining of long positions

One root cause is the left-skewed distribution of token positions in training data (Wang et al., 2024c). During pretraining, model inputs are far more likely to be shorter texts or early segments of long texts, rather than full-length sequences. This creates a severe imbalance: tokens at later positions (far right in the context) are extremely underrepresented (Xiong et al., 2023), as also shown in Figure 4. Recent analyses confirm this "left-skewed position frequency distribution" in large corpora (Wang et al., 2024c;c). For example, in the SlimPajama dataset (a massive webtext corpus), the frequency of examples using very long-range token

distances is vanishingly low: using a 2048 token window, less than 20% of the training pairs involve distances in the upper half of the window, and less than 5% involve the extreme end of the window (Bai et al., 2024a).

In gradient-based training, rarely seen position inter-actions receive only tiny updates (Wang et al., 2024c; Liu et al., 2023c). Intuitively, the model learns to predict the next token mainly using nearby context, and it gets much less practice using very distant context. Formally, one can consider the training loss as an expectation over position indices: if the probability of a position beyond, say, 75% of the max context is extremely small, then the contribution of those positions to the loss (and thus to gradient updates) is negligible. The model thus remains under-trained on long-range dependencies, even if in principle it has the capacity. This results in an effective context length much shorter than the maximum. Indeed, most open-source LLMs end up with effective context well under 50% of what they ostensibly trained for (Wang et al., 2024c).

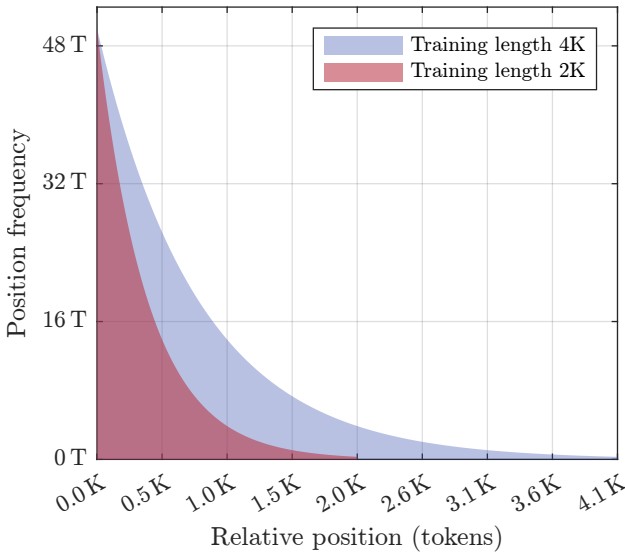

Figure 4: Position-frequency distribution for models trained with 2K vs. 4K sequence lengths after 1T tokens.

**Lemma 3** (Positional undertraining). *Consider a causal transformer trained by (stochastic) gradient descent on sequences of maximum length $L$ with population loss $\mathcal{L}(\theta) = \mathbb{E}_{\mathcal{D}}[\ell(x;\theta)]$. Fix a query position $i$ and a content position $j < i$. Let $p(j) \in [0,1]$ denote the (effective) probability that position $j$ contributes nontrivially to the training signal for predicting $x_i$ (i.e., the event that $x_j$ is present, within the causal window of $x_i$, and survives any attention-masking that yields nonzero loss gradients through the $(i,j)$ pathway). Assume $p(j)$ is left-skewed: $p(j) \to 0$ as $j \to L$.*

*For an attention head with a score $s_{i,j}(\theta) = \frac{1}{\sqrt{d}} q_i(\theta)^\top k_j(\theta)$, and weight $a_{i,j}(\theta) = \exp(s_{i,j})/\sum_{t<i} \exp(s_{i,t})$, suppose there exist constants $B, G > 0$ such that, almost surely,*

$$\left| \tfrac{\partial \ell}{\partial s_{i,j}} \right| \leq B \quad and \quad \left\| \nabla_\theta s_{i,j} \right\| \leq G.$$

*Then there is a constant $C := BG$ for which the expected per-step gradient on parameters impacting $(i,j)$ satisfies*

$$\left\| \mathbb{E}[\nabla_\theta \ell(x;\theta)] \right\| \leq C\, p(j).$$

*Consequently, after $T$ training steps with step size $\eta > 0$,*

$$\left\| \mathbb{E}[\theta_T - \theta_0] \right\| \leq \eta T\, C\, p(j).$$

*If, in addition, $a_{i,j}(\cdot)$ is $L_a$-Lipschitz in $\theta$, then*

$$\left| \mathbb{E}[a_{i,j}(\theta_T) - a_{i,j}(\theta_0)] \right| \leq L_a\, \eta T\, C\, p(j).$$

*Hence, as $p(j) \to 0$ (e.g., for $j$ near $L$), the learned $a_{i,j}$ remains arbitrarily close to its initialization (unoptimized), while weights for nearer positions $j'$ with $p(j') \gg p(j)$ move by a strictly larger amount. Thus, the learned attention from $i$ to distant $j$ is significantly weaker than to nearer $j'$.*

*Proof.* Fix $(i,j)$ with $j < i$. By the chain rule,

$$\nabla_\theta \ell(x;\theta) = \frac{\partial \ell}{\partial s_{i,j}}(x;\theta)\, \nabla_\theta s_{i,j}(\theta) + \text{(terms not passing through $(i,j)$)}.$$

By assumption, on any sample for which the $(i,j)$ pathway is active, $\left\| \frac{\partial \ell}{\partial s_{i,j}} \nabla_\theta s_{i,j} \right\| \leq BG = C$. Let $\mathbf{1}_{i \leftarrow j}$ be the indicator that the sample contributes a nonzero gradient through $(i,j)$. By definition of $p(j)$,

Figure 5: Overview of the three main factors limiting effective long-context reasoning in transformers. (1) **Training distribution skew:** long positions are underrepresented, leaving distant tokens undertrained. (2) **Positional encoding attenuation:** sinusoidal cancellation or RoPE phase misalignment shrinks positional overlap $S_{\text{pos}}(\Delta)$, weakening long-range alignment. (3) **Attention computation limits:** softmax crowding requires $\sim \ln N$ score margins to overcome distractors, while quadratic memory/computation further restricts practical sequence length.

$\mathbb{P}(\mathbf{1}_{i \leftarrow j} = 1) = p(j)$. Taking expectations and using the tower property,

$$\left\| \mathbb{E}[\nabla_\theta \ell(x; \theta)] \right\| \;\leq\; \mathbb{E}\left[ \left\| \tfrac{\partial \ell}{\partial s_{i,j}} \nabla_\theta s_{i,j} \right\| \mathbf{1}_{i \leftarrow j} \right] \;\leq\; C\, \mathbb{E}[\mathbf{1}_{i \leftarrow j}] \;=\; C\, p(j).$$

Under SGD with step size $\eta$, the parameter recursion is $\theta_{t+1} = \theta_t - \eta\, g_t$ with $g_t$ an unbiased stochastic gradient. Summing and taking expectations,

$$\left\| \mathbb{E}[\theta_T - \theta_0] \right\| \;=\; \left\| \sum_{t=0}^{T-1} -\eta\, \mathbb{E}[g_t] \right\| \;\leq\; \sum_{t=0}^{T-1} \eta\, \left\| \mathbb{E}[g_t] \right\| \;\leq\; \eta T\, C\, p(j).$$

Finally, if $a_{i,j}$ is $L_a$-Lipschitz in $\theta$, then

$$\left| \mathbb{E}[a_{i,j}(\theta_T) - a_{i,j}(\theta_0)] \right| \;\leq\; L_a \left\| \mathbb{E}[\theta_T - \theta_0] \right\| \;\leq\; L_a\, \eta T\, C\, p(j).$$

Thus, when $p(j) \to 0$, the attention $a_{i,j}$ remains near its initialization, i.e., effectively unoptimized; whereas for nearer $j'$ with larger $p(j')$, the corresponding attention parameters receive larger cumulative updates and specialize. This proves the claim. $\qquad\square$

### 3.3 Positional encoding saturation and long-range attenuation

Even if we had abundant training data for long contexts, the representation of position itself can become a limiting factor. Transformers rely on positional encodings to inject order information (since self-attention alone is order-invariant) (Vaswani et al., 2017; Huang et al., 2023). However, standard positional encoding schemes have mathematical properties that hinder very long-range discrimination (Touvron et al., 2023). This is called *positional encoding saturation*, which means that beyond a certain context length, the model's ability to distinguish positions or to maintain useful variability in positional signals deteriorates.

In the original transformer, positions are encoded by sinusoids of varying frequencies. These encodings are periodic and bounded. As positions grow, the sinusoids complete many cycles. Two very distant tokens might end up with similar encoding vectors if their positional difference coincides with a period of the

encoding. More critically, dot products between position-encoded **vectors oscillate and diminish with large separations**.

Consider two positions $i$ and $j$ with sinusoidal encodings. Their dot product contains terms of the form $\cos\big((i-j)\,\omega_k\big)$ at multiple frequencies $\{\omega_k\}$. As the separation $|i-j|$ grows, these cosines oscillate rapidly and, when summed across $k$, approximately cancel. Thus, the positional contribution to similarity averages toward zero: widely separated positional vectors become nearly orthogonal.

**Lemma 4** (Sinusoidal encodings attenuation). *Let the $d=2m$-dimensional sinusoidal positional encoding be*

$$\phi(t) \;=\; \big(\sin(\omega_1 t), \cos(\omega_1 t), \ldots, \sin(\omega_m t), \cos(\omega_m t)\big) \in \mathbb{R}^{2m},$$

*with frequencies $\{\omega_k\}_{k=1}^m \subset [\omega_{\min}, \omega_{\max}]$ that (as $m$ grows) form an approximately uniform grid on $[\omega_{\min}, \omega_{\max}]$ with $0 < \omega_{\min} < \omega_{\max} < \infty$. For positions $i, j \in \mathbb{Z}$ with separation $\Delta = |i-j|$, the normalized dot product satisfies*

$$\frac{1}{m}\,\phi(i)\cdot\phi(j) \;=\; \frac{1}{m}\sum_{k=1}^m \cos\big(\omega_k \Delta\big) \;\xrightarrow[\Delta\to\infty]{}\; 0.$$

*In particular, for any fixed frequency band width $\Omega := \omega_{\max} - \omega_{\min} > 0$,*

$$\left|\frac{1}{m}\sum_{k=1}^m \cos\big(\omega_k \Delta\big)\right| \;\leq\; \frac{2}{\Omega\,\Delta} \;+\; o_m(1),$$

*so the expected positional contribution to similarity vanishes as $\Delta \to \infty$.*

*Proof.* Write $\phi(i)\cdot\phi(j) = \sum_{k=1}^m \cos(\omega_k \Delta)$ using $\sin a \sin b + \cos a \cos b = \cos(a-b)$. With the $\omega_k$ forming an asymptotically uniform grid on $[\omega_{\min}, \omega_{\max}]$, the Riemann-sum approximation yields

$$\frac{1}{m}\sum_{k=1}^m \cos(\omega_k \Delta) \;=\; \frac{1}{\Omega}\int_{\omega_{\min}}^{\omega_{\max}} \cos(\omega \Delta)\,d\omega \;+\; o_m(1) \;=\; \frac{\sin(\omega_{\max}\Delta) - \sin(\omega_{\min}\Delta)}{\Omega\,\Delta} \;+\; o_m(1).$$

The numerator is bounded by 2 in magnitude, giving $\left|\frac{1}{m}\sum_{k=1}^m \cos(\omega_k \Delta)\right| \leq 2/(\Omega\,\Delta) + o_m(1) \to 0$ as $\Delta \to \infty$. This proves the claim. $\qquad\square$

**Implication for attention.** Let $w_{ij} \propto \exp(q_i \cdot k_j)$ be an attention weight whose queries/keys inherit an additive positional component aligned with $\phi(\cdot)$. By Lemma 4, for large separations $\Delta$ the positional dot product contributes negligibly, so $w_{ij}$ receives little reinforcement from positional alignment alone. In plain terms, a token at position 20,000 has a near-orthogonal positional vector to one at position 1, attenuating long-range interactions unless content features provide compensating evidence.

**Rotary Position Embedding (RoPE).** Modern LLMs often use RoPE to encode relative positions by rotating query/key vectors. RoPE faces a similar issue as it encodes relative phase shifts such that the inner product of query$_i$ and key$_j$ implicitly includes a factor $\cos(\theta(i-j))$ (for each frequency band). Without adjustment, $\cos(\theta\Delta)$ becomes very small for large $\Delta$. This yields a "long-range attenuation" effect, and attention scores between tokens far apart are exponentially dampened. The base frequency used in RoPE effectively sets the length scale at which attention fades. If the base is not scaled for a larger window, beyond a certain distance, the overlap between rotated vectors is nearly zero. Empirical evidence of this comes from attempts to extend context windows: simply increasing a model's maximum position with RoPE without rescaling leads to steep increases in perplexity and degraded utility beyond the original length (Xiong et al., 2023; Bai et al., 2024a). Recent research identified this as a key limitation and proposed NTK-aware or scaled RoPE, which amplifies or adjusts the rotation frequency to "stretch" out the positional encoding over a longer span (Xiong et al., 2023).

**Learned positional embeddings.** Some models use learned position embeddings (one vector per position up to $L$). Here, the issue is simpler than beyond the positions seen in training; the model has no defined embedding. Even within the training range, if most training sequences were shorter than $L$, the embeddings for the largest indices are poorly trained and often end up near-initialization. They may take on nearly identical or arbitrary values. Thus, the model effectively treats all positions beyond some point as the same *"unknown"* position.

### 3.4 Attention computation limits

The third fundamental limitation lies in the attention mechanism's computation itself when faced with extremely long sequences (as shown in Figure 5). The softmax self-attention in Transformers has a quadratic complexity in sequence length $N$ for both computation and memory, making very large $N$ costly (Dao, 2023). But beyond runtime, there are mathematical reasons why softmax attention struggles as $N$ grows, especially for reasoning tasks (Li et al., 2023a; Liu et al., 2023a).

**Softmax competition and diffusion of focus.** In an attention head, the probability of attending to any given token is $\text{softmax}(e_{ij}) = \frac{\exp(e_{ij})}{\sum_{k=1}^{N} \exp(e_{ik})}$, where $e_{ij}$ is the compatibility of token $i$'s query with token $j$'s key. As $N$ increases, the denominator grows with many terms (Huang et al., 2023). If there is one relevant token among a sea of $N-1$ irrelevant ones, the model's query must assign that relevant token a logit advantage on the order of $\log N$ to maintain a fixed attention probability. For example, to have 50% of the attention mass on one token out of $N$, the score for that token needs to be about $\ln(N)$ larger than the average of the others. This is a steep requirement: as context length grows, the model must sharpen the attention distribution more and more to pick out a single item. If the model's scores for irrelevant tokens have some variance, a large $N$ increases the chance that some distractor token will get a moderately high score by coincidence, eating into the probability of the true relevant token. This effect can be viewed as a type of **combinatorial noise** which means that with many keys, the softmax normalization makes it difficult to preserve a strong signal for the correct one unless the model has learned extremely fine-grained, high-contrast scoring. In practice, as contexts lengthen, attention tends to diffuse, and it often spreads over many tokens or attends mostly to the recent segment, unless a very obvious keyword or cue is present to focus on the far context (Liu et al., 2023c).

**Lemma 5** (Softmax crowding and the need for $\ln N$ margins). *Consider a single attention head over $N$ candidate tokens. Let one* relevant *token have score $s \in \mathbb{R}$ and the remaining $N-1$ irrelevant tokens have scores $X_1, \ldots, X_{N-1}$ that are i.i.d. with finite log-moment $\psi(1) := \ln \mathbb{E}[e^{X_1}] < \infty$. The softmax attention on the relevant token is*

$$P_N = \frac{e^s}{e^s + \sum_{k=1}^{N-1} e^{X_k}}.$$

*If $s$ does not grow with $N$ (e.g., $s - \mu = C$ is fixed, where $\mu := \mathbb{E}[X_1]$), then $P_N \xrightarrow[N\to\infty]{a.s.} 0$.*

*Proof.* By the strong law of large numbers applied to $e^{X_k}$ (which is integrable since $\psi(1) < \infty$),

$$\frac{1}{N-1} \sum_{k=1}^{N-1} e^{X_k} \xrightarrow[N\to\infty]{a.s.} \mathbb{E}[e^{X_1}] =: M \in (0, \infty).$$

Hence,

$$P_N = \frac{e^s}{e^s + \sum_{k=1}^{N-1} e^{X_k}} = \frac{e^s}{e^s + (N-1)\left(\frac{1}{N-1}\sum_{k=1}^{N-1} e^{X_k}\right)} \longrightarrow \frac{e^s}{e^s + (N-1)M} \xrightarrow[N\to\infty]{} 0.$$

Thus, if $s$ is $O(1)$ (including $s - \mu = C$ constant), the denominator grows linearly in $N$ while the numerator is fixed, forcing $P_N \to 0$ almost surely. $\qquad\square$

**Corollary 1** (Required scaling to keep constant attention). *Fix any target $p \in (0,1)$. Under the assumptions of Lemma 5, to have $P_N \to p$ it suffices and is asymptotically necessary that*

$$s = \ln N + \ln M + \ln\left(\frac{p}{1-p}\right) + o(1), \quad \text{where } M = \mathbb{E}[e^{X_1}].$$

*Equivalently, the score margin must scale as $s = \Theta(\ln N)$ up to additive constants depending on the irrelevant-score distribution and the desired probability level $p$.*

**Interpretation.** With many distractors, softmax acts like a competition against the *sum* of irrelevant exponentiated scores, which concentrates near $(N-1)M$. A fixed gap $s - \mu = C$ is overwhelmed as $N$ grows, sending the relevant attention to 0. Maintaining a constant selection probability demands an $O(\ln N)$ growth in the relevant score margin, which is a pressure that generic training may not provide when relevant facts are sparsely embedded among large amounts of filler. Benchmarks with strong, repeated cues make this margin easy to achieve; in information-dense settings without such cues, performance degrades as $N$ increases.

**Memory and precision.** The quadratic memory use of attention means practical implementations struggle with very long inputs. Even if a model supports 100K tokens, performing attention on that many tokens can hit memory limits or require dumping to slower memory, which introduces numerical precision challenges. Additionally, summing over 100K exponentiated scores in softmax can lead to extremely large or small values, testing the limits of floating-point precision (Dao, 2023). Transformers process all tokens in parallel, which means every layer has to recompute interactions across the full sequence. With many layers, the opportunities for error accumulation or gradient diminishing over long ranges increase. In contrast, a recurrent process (like a state-space model) carries information forward iteratively, which has its own challenges (e.g., gradient vanishing through time) but uses a different mechanism for long-term dependency. In transformers, while skip connections help gradients propagate, there is no persistent memory that carries over from token to token beyond what attention redistributes at each layer. If at some intermediate layer the model fails to propagate a piece of information from position $j$ to $i$, later layers can only recover it if some indirect path exists. With very deep contexts, ensuring that all needed long-range links are formed somewhere in the stack is non-trivial.

Empirical benchmark results echo these theoretical concerns. $\infty$ Bench (Zhang et al., 2024a; Chang et al., 2023; Pang et al., 2021; Bai et al., 2024b; Kočiskỳ et al., 2018) which pushes context to 100K+ tokens, finds that current long-context LLMs *"still require significant advancements"* to handle 100K tokens effectively. In LongBench's (Bai et al., 2024b) multitask evaluation, models without explicit long-context training or architectural adjustments see dramatic drops in accuracy on tasks as input length increases, and as NeedleBench's (Li et al., 2025a) Ancestral Trace Challenge shows, even at a relatively modest 2K length, complex logical reasoning across dispersed information is often beyond the reach of today's transformers. The limitations of attention become most apparent when naive long contexts meet tasks requiring synthesis of widely separated pieces; either the transformer tends to focus myopically on one part or gets lost trying to handle everything, revealing a fundamental lack of robust long-range reasoning.

Long-context failures manifest the underlying triad of computability, statistical insufficiency, and finite information capacity distinctly. Positional undertraining (Lemma 3) reflects statistical insufficiency: rare position pairs receive negligible gradient updates. Encoding attenuation (Lemma 4) demonstrates finite information capacity: sinusoidal representations compress poorly over long ranges. Softmax crowding embodies computational constraints: distinguishing signal from noise requires logarithmic score margins that transformers struggle to maintain. Thus, effective context compression below nominal length is not architectural accident but mathematical necessity.

## 4 Reasoning or Recitation? Probing LLM Abilities

Despite broad linguistic competence, LLMs remain brittle in systematic reasoning. Most LLMs optimize the next-token likelihood under an autoregressive factorization:

$$\max_{\theta} \sum_{(x_1,\ldots,x_T) \in D} \sum_{t=1}^{T} \log p_\theta(x_t \mid x_{<t}). \tag{25}$$

Here, $x_t$ is the token at position $t$ and $x_{<t}$ its preceding context. This training objective helps models capture patterns in text and can induce reasoning-like behavior by exploiting regularities in training data, latent decompositions, and prompt-induced intermediate traces. However, it does not explicitly train models to perform algorithmic reasoning, track symbolic state, or verify logical validity of outputs (Huang & Chang, 2023). As a result, LLMs may produce fluent but incorrect answers and often break under distribution shift, compositional recombination, constraint satisfaction, or long-horizon dependency tracking (Huang & Chang, 2023; Lee et al., 2024b; Shojaee et al., 2025; Matarazzo & Torlone, 2025).

Table 2: A comprehensive map of long-context failure modes and the corresponding architectural, training, and inference-time solutions.

| Issue | Failure Mode | Representative Solutions | Key references |
|---|---|---|---|
| Left-skewed training over positions ("positional under-training") | Model underuses late-context tokens; performance drops when evidence appears in the middle/end of long inputs. | *Data-side:* Reweight or resample long-range pairs; curriculum that up-weights large relative distances; extend pretraining/fine-tuning on long sequences. *Index remapping:* Position *redistribution/shift* (e.g., StRing) to reuse well-trained low indices at inference. *Loss shaping:* depth-aware losses to boost gradients at large separations. | (Bai et al., 2024b; Li et al., 2025a; Wang et al., 2024c; Saito et al., 2025) |
| Positional encoding attenuation (sinusoidal & vanilla RoPE saturate) | Dot products between distant positions vanish/oscillate; far tokens receive negligible positional reinforcement. | *RoPE scaling:* NTK-aware or base-frequency scaled RoPE; dynamic/learned scaling per head. *Relative schemes:* ALiBi, T5-style relative attention, KERPLE; monotone distance biases that do not saturate; hybrid absolute+relative mixes. *Content-aided:* strengthen content cues via contrastive span objectives. | (Press et al., 2021; Su et al., 2021; Peng et al., 2023; Bai et al., 2024b) |
| Softmax crowding (logit margin must grow like $\ln N$) | Single relevant token drowned among $N$ distractors; attention mass diffuses as context grows. | *Architectural:* Top-$k$/sparse attention, landmark/sink tokens, hierarchical routing; multi-hop retrieval within the model. *Training:* margin-aware objectives on salient spans; hard-negative mining with long-context distractors; temperature scheduling. *Inference:* focused rereading (two-pass refine), constraint hints. | (Li et al., 2025a; Zhang et al., 2024b; Dao et al., 2022; Child et al., 2019) |
| Quadratic memory/compute of global attention | 128K–1M tokens infeasible or numerically brittle; KV cache and memory blow-ups. | *Algorithmic:* FlashAttention/IO-aware kernels; block-sparse/dilated/linear-time variants; chunked sliding-window with cross-chunk summaries. *System:* paged KV cache, quantization of KV (NF4/FP8), CPU offloading with prefetch; windowed decoding. | (Dao et al., 2022; Child et al., 2019; Beltagy et al., 2020; Dao, 2023; Li et al., 2023a; Liu et al., 2023a) |
| Fragmented long-range reasoning (missing multi-hop chains) | Model fails when evidence is dispersed and must be combined over long spans. | *Planner-reader loops:* iterative reasoning over retrieved segments; scratchpad/state passing across segments. *Supervision:* multi-hop chain objectives with distant supervision; compositional curricula. *Structure:* section headers/summaries injected as anchors. | (Bai et al., 2024b; Lewis et al., 2020b) |
| Context selection is noisy (irrelevant filler dilutes signal) | Performance collapses as irrelevant text increases density. | *Retrieval/compression:* RAG; learned summarization/compression to *evidence sketches*; entropy/gradient-based token pruning; saliency pre-filters before LLM. *Safety net:* late fusion of multiple compressed candidates. | (Bai et al., 2024b; Li et al., 2025a; Rae et al., 2019; Lewis et al., 2020b) |
| Index generalization beyond trained $L$ (learned embeddings) | Positions beyond the seen range are near-initialization or ill-defined. | *Inter/extrapolation:* continuous position functions; Fourier features with scale tuning; spline-based or low-rank position decoders; rope-based annealing during finetune. | (Su et al., 2021; Peng et al., 2023; Press et al., 2021) |
| Chunk boundaries break dependencies | Errors at the window cuts; cross-chunk links lost. | *Bridges:* memory tokens per chunk; summary vectors with attention into prior chunks; overlap windows with learned dedup; cached entity graphs. | (Dai et al., 2019; Wu et al., 2022; Sun et al., 2024) |
| External memory is not persistent across tasks/sessions | Forgets earlier sessions; cannot keep project-scale facts. | *Persistent stores:* vector DB + RAG with typed schemas; tool-augmented retrieval (program-of-thought); long-term key–value memory distilled from transcripts/logs. | (Lewis et al., 2020b; Schick et al., 2023) |
| Evaluation misalignments (proxy tasks, short-context fine-tunes) | Improvements don't transfer to real long-doc reasoning. | *Benchmarks:* use LongBench/NeedleBench/ $\infty$ Bench style tasks with dispersion & density controls; report accuracy vs. length, energy/token, and effective-context curves; ablate position reweighting. | (Bai et al., 2024b; Li et al., 2025a; Zhang et al., 2024b; Wang et al., 2024c) |
| Alternatives to pure Transformers (state-space and hybrids) | Attention saturation or cost dominates at scale. | *SSM class:* S4/Hyena/Mamba for linear-time long-range processing; *Hybrids:* LongMamba (local attention + global state); gated recurrence for persistent memory; cross-architecture distillation from long-attention teachers. | (Gu & Dao, 2023; Gu et al., 2021; Sun et al., 2024) |

Specifically, the below sections explain (i) why likelihood-optimized generation can produce reasoning-like behavior yet fall short of reliable inference and introduce *reasoning efficiency* as a quality-per-compute lens (Section 4.1); (ii) catalog failure modes that undermine reliability (Section 4.2); and (iii) formalize a unified objective with constraints and verification, then instantiate it with practical patterns (Sections 4.3 and 4.4). A related question is whether LLMs truly exhibit compositional reasoning or only approximate it in specific tasks. In semantics, compositional generalization means consistently combining meaning across contexts and representations, not just solving unseen combinations (Abzianidze et al., 2025). Some improvements in reasoning may therefore come from better prompts, tools, or symbolic interfaces rather than fully internalized compositional reasoning (Lappin, 2024). We thus distinguish between *apparent compositional success*, which can arise from pattern matching or prompt scaffolding, and *robust compositionality*, which requires systematic behavior under recombination and abstraction.

## 4.1 Why likelihood falls short

Current LLMs (e.g., GPT-3/ChatGPT/GPT-4, PaLM 2, InstructGPT, LaMDA) function like a "Mad Libs" game: they fill in blanks using statistical associations rather than understanding and logically manipulating facts (Zhang, 2024). The probabilistic modeling approach captures correlations in language but does not guarantee that the model has learned the underlying functions or rules needed to solve reasoning problems (Zhao et al., 2024). Many reasoning tasks (arithmetic, logic puzzles, algorithmic procedures, etc.) require precise intermediate computations and compositional generalization, the ability to combine learned components in novel ways. Models learn the pieces but often fail to compose them: in Boolean logic, accuracy drops sharply as expressions grow deeper or wider, even when the basic primitives are mastered. This suggests they rely on surface patterns rather than underlying rules (Kim & Thorne, 2024).

This aligns with the Language-of-Thought Hypothesis (LoTH), which holds that human-like reasoning depends on internal symbolic structures; by contrast, LLMs store knowledge in continuous vectors rather than discrete, combinatorial representations (Quilty-Dunn et al., 2023). We avoid strong claims that LLM reasoning implies a classical "language of thought" in the cognitive-science sense. A more cautious view is that LLMs may learn task-specific internal patterns that sometimes enable decomposition or variable binding, but these behaviors remain sensitive to prompting, content, and distribution shift. Moreover, both humans and LLMs show content effects on reasoning tasks, suggesting that surface performance alone should not be interpreted as evidence of stable symbolic reasoning (Lampinen et al., 2024). Lee et al. (2024b) evaluated GPT-family models on the Abstraction and Reasoning Corpus (ARC), a challenging inductive reasoning benchmark. They found that while LLMs show some inference ability, they still lag in terms of logical coherence, compositionality, and productivity compared to human problem solvers (Lee et al., 2024b). The stochastic nature of next-token generation can miss the rigid logic needed for puzzles or math problems that have one correct outcome following from premises.

These observations motivate not only improving correctness but also explicitly accounting for the cost of obtaining it. Reasoning efficiency, the expected quality per unit of compute across tasks, is therefore used to assess progress:

$$\eta(M) = \mathbb{E}_{t \sim \mathcal{T}} \left[ \frac{Q(M,t)}{C(M,t)} \right], \tag{26}$$

where $M$ denotes a fully specified inference configuration (model architecture and parameters together with decoding hyperparameters and any tool/execution policy), $\mathcal{T}$ is a distribution over tasks or problem instances, and $t \sim \mathcal{T}$ is a single sampled instance. $Q(M,t)$ is the solution quality, for instance $t$ (e.g., EM/accuracy), and $C(M,t)$ measures compute for that instance, including prompt and generation tokens, a FLOPs proxy, latency, and memory (Guo et al., 2025a). Under this lens, recent reasoning models (e.g., OpenAI o1; DeepSeek-R1) can overthink, producing long chains with redundant steps or shallow branching that raise $C$ without commensurate gains in $Q$ (Jaech et al., 2024; Liu et al., 2024a; Sui et al., 2025; Qu et al., 2025b).

## 4.2 What causes reasoning to fail in LLMs?

LLMs exhibit strong fluency but limited stepwise reasoning: they may produce correct answers accompanied by unsound rationales, over-extend derivations, or shift explanations under minor prompt variations. Advancing

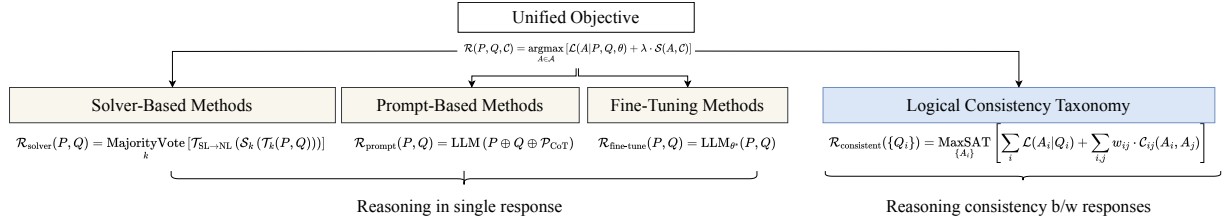

Figure 6: Mathematical Framework Adaptations for LLM Reasoning Approaches.

toward reliable reasoning requires models that deliver correct final answers and procedurally valid intermediate steps. Achieving this requires confronting four persistent failure modes: 1) objective mismatch that makes intermediate reasoning disposable, 2) spurious correlations from unverified knowledge, 3) search pathologies that misallocate compute and derail inference, and 4) fragile interfaces and metrics that distort or obscure reasoning quality.

**Objective mismatch & disposable mediators.** Standard training maximizes answer likelihood rather than step faithfulness. Let $X$ be the input, $Y$ the final answer, and $Z$ the chain-of-thought (CoT). The model marginalizes across traces:

$$P_\theta(Y \mid X) = \sum_Z P_\theta(Y \mid X, Z) \, P_\theta(Z \mid X).$$ (27)

Equation 27 is the optimized quantity, so a fluent but non-causal Z can persist if it maintains a high $P_\theta(Y \mid X)$ (Wei et al., 2022; Barez et al., 2025). Outcome-only RL further entrenches this: optimizing $\max_\theta \mathbb{E}[R(Y, \hat{Y}_\theta)]$ with $R = \mathbb{I}[\hat{Y}_\theta = Y]$ imposes no requirement to use $Z$ (Turpin et al., 2023; Chen et al., 2025d). In mediation terms, the CoT $Z$ often has a small *indirect effect* (IE) on $Y$ relative to the *direct effect* (DE) of $X$:

$$\text{TE} = \underbrace{\mathbb{E}[Y \mid do(X)] - \mathbb{E}[Y \mid do(X'), Z]}_{\text{DE}} + \underbrace{\mathbb{E}[Y \mid do(Z)] - \mathbb{E}[Y \mid do(Z')]}_{\text{IE}},$$ (28)

where the operator $do(\cdot)$ denotes an *intervention* in the sense of Pearl's causal calculus, that is, setting a variable to a specific value while disconnecting it from its natural causes. In this context, $do(X)$ corresponds to externally fixing the model input or prompt, and $do(Z)$ corresponds to enforcing a particular chain-of-thought reasoning path. The expectations $\mathbb{E}[Y \mid do(X)]$ and $\mathbb{E}[Y \mid do(Z)]$ therefore measure the post-intervention outcomes of $Y$ when $X$ or $Z$ are directly manipulated rather than merely observed. With $X', Z'$ representing suitable counterfactual interventions, empirical results show that IE$\approx 0$ on many tasks (Paul et al., 2024), rendering $Z$ a *disposable mediator*. Consequently, systems rewarded solely for final correctness tend to ignore or fabricate intermediate steps; improving faithfulness, therefore, requires step-aware objectives or verifiable rewards that make $Z$ causally indispensable.

**Spurious correlations from knowledge injection.** External knowledge can correlate with answers without being deployed as a causal instrument in reasoning, producing CoTs that cite facts yet do not depend on them for the inference itself. Treating $Z$ as a mediator enables front-door adjustment (Wu et al., 2024a):

$$P(Y \mid do(X)) = \sum_z P(Y \mid do(z)) \, P(z \mid do(X)),$$ (29)

where $z$ ranges over knowledge-aware traces that pass logical checks. Equation 29 re-weights answers so that only causally valid $z$ contribute, mitigating spurious shortcuts induced by mere fact mention. In practice, introducing knowledge should thus be coupled with mechanisms that verify its instrumental use inside $Z$; otherwise, models may appear informed while relying on correlations rather than reasoned application of the information.

**Search pathology & compute misallocation.** Greedy or locally sampled decoding is myopic (one path, no backtracking), whereas naive multi-path expansion bloats search without guidance. Reframing reasoning as planning assigns values to partial states $s$ and actions $a$:

$$\pi(a \mid s) \propto \exp\left(Q^\pi(s, a)/\tau\right),$$ (30)

direct exploration toward high-value traces (Hao et al., 2023). Complementary training prefers concise correctness (e.g., CoPO/AoT-O3) by scaling accuracy $Q$ and cost $C$:

$$\max_{\theta} \; \mathbb{E}\big[Q_\theta - \lambda \tilde{C}_\theta\big], \qquad \tilde{C}_\theta = \text{normalized tokens/FLOPs.} \tag{31}$$

At the meta-level, models must learn when extended reasoning is warranted. Let $c \in \{\texttt{short}, \texttt{think}\}$ be a control token; Thinkless, (Fang et al., 2025) optimizes

$$\mathcal{L} = \underbrace{\mathcal{L}_{\text{ctrl}}(c \mid X)}_{\text{mode selection}} + \underbrace{\mathcal{L}_{\text{resp}}(Y \mid X, c)}_{\text{answer correctness}}, \tag{32}$$

reducing unnecessary long-CoTs while preserving accuracy (Fang et al., 2025). Thus, effective systems guide search toward promising partial derivations, explicitly trade off accuracy against computational expense, and route adaptively between terse and deliberative modes, avoiding both premature commitment and unproductive overthinking (Zhang et al., 2024f). A related approach is hierarchical reasoning, where a model first picks abstract subgoals and then expands them into concrete steps (e.g., Hierarchical Reasoning Model (HRM) (Wang et al., 2025a), HyperTree Planning (HTP) (Gui et al., 2025)). But these methods often falter: errors in planning cascade downward, high-level plans may misalign with executable detail, and the space of possible subplans still grows rapidly unless heavily pruned. Thus, HRMs need subgoal verification, pruning, and cost awareness to be reliable (as discussed in Section 4.3).

**Interface & metric fragility.** Prompt interfaces, especially delimiter tokens for thought, can inadvertently truncate or skew $Z$, exposing *unthinking* vulnerabilities (Zhu et al., 2025b). The same controls can be co-opted defensively to terminate unsafe or redundant chains. On the evaluation axis, answer-only metrics (e.g., Pass@$k$) are largely insensitive to improvements in intermediate steps; verifiable rewards (e.g., CoT-Pass@$k$) demand that both Y and Z be correct, surfacing gains that Pass@$k$ obscures (Wen et al., 2025). Beyond final accuracy, compositional causal tests (CCR) evaluate local↔global causal consistency (Maasch et al., 2025), and length-sensitive rewards must penalize repetition to resist trivial length hacking (Yeo et al., 2025). Robust reasoning, therefore, depends on interfaces that do not perturb cognitive trajectories and on process-aware metrics that reward the correctness of the derivation, not merely its endpoint.

To summarize, reasoning degradation again traces directly to the theoretical limitations of LLMs. The objective mismatch problem reflects computational limits: likelihood training cannot distinguish causal from spurious mediators. Spurious correlations reveal statistical insufficiency: models learn dataset patterns rather than compositional rules. Search pathologies demonstrate finite information capacity: exploring exponentially large reasoning spaces under token budgets forces myopic decisions. These are not fixable through scale alone but require architectural shifts that respect these fundamental constraints.

## 4.3 Unified objective to capture reasoning

The above failure modes show that scaling model size or CoT length alone is insufficient; instead, we need methods that verify correctness at each step, avoid disposable or misleading reasoning chains, and allocate computation where it truly contributes to performance. Framed by reasoning efficiency $\eta = \mathbb{E}[Q/C]$, real progress means improving quality per unit cost rather than simply producing longer reasoning traces (autoregressive manner as in Equation 25). This naturally leads to a unified objective that augments likelihood with verification and cost regularization, ensuring that each intermediate step is causally meaningful, the search remains focused, and consistency is maintained.

$$\mathcal{R}(P, Q, \mathcal{C}) = \underset{A \in \mathcal{A}}{\arg\max} \left[\mathcal{L}(A|P, Q, \theta) + \lambda \cdot \mathcal{S}(A, \mathcal{C})\right], \tag{33}$$

where $P = \{p_1, p_2, \ldots, p_n\}$ represents the set of premises, $Q$ is the query or question, $\mathcal{C}$ represents consistency constraints, $\mathcal{A}$ is the space of possible answers, $\mathcal{L}(A|P, Q, \theta)$ is the likelihood of answer $A$ given premises and query, $\mathcal{S}(A, \mathcal{C})$ is the consistency score, and $\lambda$ balances local correctness versus global coherence. This framework captures the fundamental tension in LLM reasoning between generating locally plausible responses and maintaining global logical consistency across multiple outputs.

### 4.3.1 Instantiate Reasoning using the Unified Objective

**Solver-based methods.** Solver-based approaches transform the reasoning problem into a symbolic manipulation task, where the framework becomes:

$$\mathcal{R}_{\text{solver}}(P, Q) = \mathcal{T}_{\text{SL}\to\text{NL}} \circ \mathcal{S}_{\text{solve}} \circ \mathcal{T}_{\text{NL}\to\text{SL}}(P, Q), \tag{34}$$

where $\mathcal{T}_{\text{NL}\to\text{SL}}$ [Natural Language to Symbolic Language], $\mathcal{S}_{\text{solve}}$ [Symbolic Formulation to Symbolic Answer], and $\mathcal{T}_{\text{SL}\to\text{NL}}$ [Symbolic Answer to Natural Language Answer]. Equation 34 should be viewed as a general reasoning template rather than a fixed pipeline; in practice, different tasks require externalizing different stages (e.g., decomposition, execution, retrieval, or verification) to achieve reliable performance. Methods such as Satisfiability-aided language models (SatLM) (Ye et al., 2023b) translate natural language questions into different formulations and employ language solvers to derive answers, while Language-INtegrated neuro-symbolic reasoning with Constraints (LINC) (Olausson et al., 2023) generates multiple natural language to symbolic language translations with k-majority voting to mitigate translation errors. Logic-LM (Pan et al., 2023) extends this approach by utilizing task-specific formulations, including logic programming, first-order logic, constraint satisfaction problems, and different formulations tailored to different datasets. More recent advances include CLOVER (Ryu et al., 2025), which performs compositional translation via atomic sentences with logical dependency structures, and VERUS-LM (Callewaert et al., 2025), which introduces self-refinement steps using feedback from reasoning engines to correct erroneous logical statements.

These solver-based methods provide deterministic and verifiable outputs, making them attractive for applications requiring high reliability. The framework adaptation for solver-based methods, as shown in Figure 6, becomes:

$$\mathcal{R}_{\text{solver}}(P, Q) = \underset{k}{\text{MajorityVote}} \left[ \mathcal{T}_{\text{SL}\to\text{NL}} \left( \mathcal{S}_k \left( \mathcal{T}_k(P, Q) \right) \right) \right], \tag{35}$$

where $k$ indexes different translation attempts, $\mathcal{T}_k$ represents the $k$-th translation function, $\mathcal{S}_k$ denotes the corresponding symbolic solver, and MajorityVote selects the most frequent answer across multiple solver runs to mitigate translation errors. However, they suffer from several fundamental limitations, including translation brittleness where small errors in natural language to symbolic language conversion severely affect results, information loss during symbolic translation that can render problems unsolvable, and exponential search complexity as problem complexity increases (Feng et al., 2024; Zhang et al., 2024e).

**Prompt-based methods.** For prompt-based approaches, the reasoning occurs within the language model itself:

$$\mathcal{R}_{\text{prompt}}(P, Q) = \text{LLM}(P \oplus Q \oplus \mathcal{P}_{\text{reasoning}}), \tag{36}$$

where $\mathcal{P}_{\text{reasoning}}$ represents reasoning-specific prompts, and $\oplus$ denotes concatenation. Two families are common: (i) explicit chain modeling (CoT/ToT/CR/DoT) and (ii) symbolic expression generation (Symb-CoT/LoT/Aristotle)

The first subcategory focuses on explicit chain modeling, where methods like CoT (Wei et al., 2022) enable step-by-step reasoning traces, Tree-of-Thoughts (ToT) (Yao et al., 2023) provides multi-path exploration with backtracking capabilities, Cumulative Reasoning (CR) (Zhang et al., 2023a) decomposes complex problems into manageable components using directed acyclic graph structures, and Diagram-of-Thought (DoT) (Zhang et al., 2024d) employs role-specific tokens for reasoning navigation. The second subcategory emphasizes symbolic expression generation, with methods such as SymbCoT (Xu et al., 2024b) prompting LLMs to translate natural language problems to symbolic formulations followed by step-by-step solutions with verification, Logic-of-Thought (LoT) (Liu et al., 2025b) instructing models to translate and expand logical expressions based on logic rules, and Aristotle (Xu et al., 2025b) exploiting underlying logical structures for decomposition to improve both efficacy and efficiency. Another relevant decomposition-based strategy is least-to-most prompting, which first decomposes a difficult problem into simpler subproblems and then solves them sequentially, thereby reducing the burden on single-step inference and making the structure of intermediate dependencies more explicit Zhou et al. (2022).

The framework adaptation for prompt-based methods (Figure 6) takes two primary forms:

$$\mathcal{R}_{\text{prompt}}(P, Q) = \text{LLM}\left(P \oplus Q \oplus \mathcal{P}_{\text{CoT}}\right) \text{ or } \underset{\text{path}}{\text{Search}}\left[\text{ToT}(P, Q)\right]. \tag{37}$$

**Fine-tuning methods.** Fine-tuning modifies model parameters to incorporate logical reasoning directly. This targets gaps in logic-rich supervision by adding rules, proofs, and structured derivations, particularly logical multi-step deduction and proofs, in pre-training corpora composed mainly of human-written texts that exhibit reflexive thinking rather than rigid logical reasoning (Morishita et al., 2024). The framework adaptation for fine-tuning methods (as shown in Figure 6) becomes:

$$\mathcal{R}_{\text{fine-tune}}(P, Q) = \text{LLM}_{\theta^*}(P, Q) \text{ where } \theta^* \sim \mathcal{D}_{\text{logic-augmented}}, \tag{38}$$

where $\theta^*$ represents the optimized model parameters, and $\mathcal{D}_{\text{logic-augmented}}$ denotes the logic-augmented training distribution that includes synthetic reasoning samples, formal proofs, and structured logical derivations.

LogicAsker (Wan et al., 2024) formally defines comprehensive sets of atomic and extended logic rules necessary for formal reasoning based on propositional and predicate logic, creating corresponding fine-tuning data to improve reasoning abilities on poorly performing rules. ALT (Morishita et al., 2024) constructs synthetic logic corpora based on diverse logical reasoning rules, including syllogism, contraposition, and De Morgan's laws, comprising numerous multi-step deduction samples with unknown facts and challenging distractors. AMR-LDA (Bao et al., 2024) takes a different approach by converting natural language into Abstract Meaning Representation graphs to capture logical sentence structures before augmenting through logically augmented AMR graphs. LoGiPT (Feng et al., 2024) empowers LLMs by directly learning reasoning processes of logical solvers, avoiding risks of unanswerable questions when facing parsing errors in solver-based methods. Yet reasoning ability on isolated questions is not sufficient. This isolation motivates the need to ensure consistency across multiple queries.

### 4.3.2 Reasoning consistency within the unified objective

The consistency framework ensures outputs satisfy logical constraints across multiple queries:

$$\mathcal{S}(A, \mathcal{C}) = \prod_{c \in \mathcal{C}} \mathcal{I}[c(A) = \text{True}], \tag{39}$$

where $\mathcal{I}[\cdot]$ is the indicator function and $c(A)$ evaluates constraint $c$ on answer $A$. This framework addresses the critical problem that LLMs are prone to producing responses that contradict themselves across different questions, which violates logical consistency and undermines reliability and trustworthiness, particularly in high-stakes scenarios. Logical consistency encompasses multiple constraint types that can be mathematically formalized. 1) Negation consistency requires $\mathcal{C}_{\text{neg}} = \{p \oplus \neg p, \neg(p \wedge \neg p)\}$, ensuring that $p$ and $\neg p$ cannot both be true simultaneously. 2) Implication consistency requires $\mathcal{C}_{\text{imp}} = \{(p \rightarrow q) \wedge p \Rightarrow q\}$ to preserve logical entailment relationships. 3) Transitive consistency enforces $\mathcal{C}_{\text{trans}} = \{(p \rightarrow q) \wedge (q \rightarrow r) \Rightarrow (p \rightarrow r)\}$ to maintain transitive relationships across multiple statements. 4) Factuality consistency requires $\mathcal{C}_{\text{fact}} = \{\mathcal{KB} \models A\}$, ensuring that each generated answer $A$ is *logically entailed ($\models$) by* the external knowledge base $\mathcal{KB}$, such that all statements produced by the model are consistent with verified facts and do not contradict established information sources.

Compositional consistency combines these constraint types as $\mathcal{C}_{\text{comp}} = \mathcal{C}_{\text{neg}} \cup \mathcal{C}_{\text{imp}} \cup \mathcal{C}_{\text{trans}} \cup \mathcal{C}_{\text{fact}}$ for simultaneous satisfaction of multiple logical requirements. The general consistency enhancement framework follows:

$$\text{ConsistencyEnhance}(Q_1, \dots, Q_n) = \underset{A_1, \dots, A_n}{\text{argmax}} \sum_{i=1}^{n} \mathcal{L}(A_i | Q_i) \text{ s.t. } \mathcal{C}(A_1, \dots, A_n). \tag{40}$$

BeliefBank (Kassner et al., 2021) stores raw LLM answers and employs MaxSAT solvers to flip beliefs that clash significantly with others, using modified beliefs as query context via feedback to improve both consistency and accuracy. ConCoRD (Mitchell et al., 2022) generates multiple candidate outputs, estimates

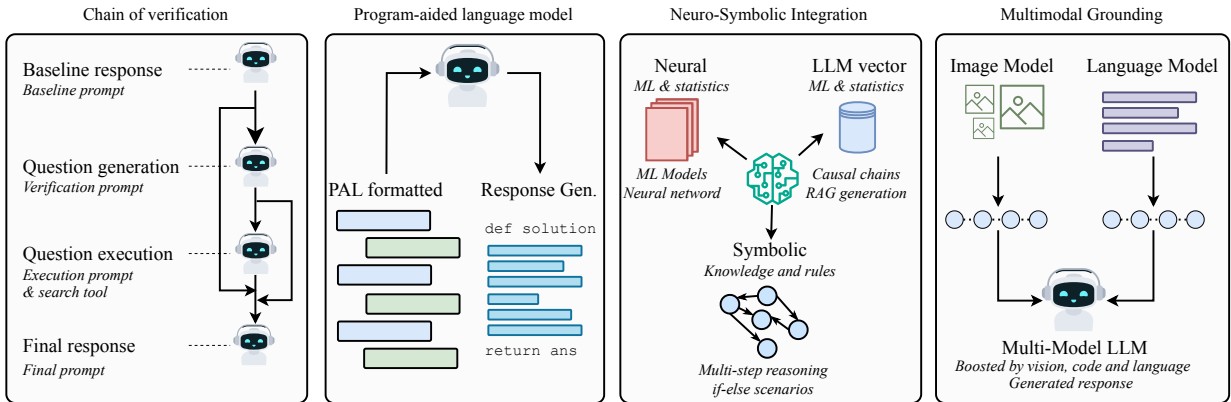

Figure 7: Overview of reasoning strategies, (a) Chain-of-Verification (generate, execute, verify), (b) Program-Aided Language (code-based execution), (c) Neuro-Symbolic Integration (neural + rules for multi-step logic), and (d) Multi-modal LLM (image+language grounding).

soft pairwise inconsistencies using natural language inference, and finds optimal outputs for each question via MaxSAT solvers. The framework adaptation for consistency-enhanced methods (as shown in Figure 6) follows:

$$\mathcal{R}_{\text{consistent}}(\{Q_i\}) = \underset{\{A_i\}}{\text{MaxSAT}}\left[\sum_i \mathcal{L}(A_i|Q_i) + \sum_{i,j} w_{ij} \cdot \mathcal{C}_{ij}(A_i, A_j)\right], \tag{41}$$

where $\{Q_i\}$ represents the set of related queries, $\{A_i\}$ denotes the corresponding answer set, $w_{ij}$ are pairwise constraint weights between answers $A_i$ and $A_j$, $\mathcal{C}_{ij}(A_i, A_j)$ evaluates consistency constraints between answer pairs, and MaxSAT denotes maximum satisfiability optimization that finds the assignment maximizing the number of satisfied constraints. Conceptually, the weights $w_{ij}$ act as *reasoning parity bits*, analogous to parity checks in error-correcting codes, indicating whether the logical relation between $A_i$ and $A_j$ satisfies the required consistency constraints. During optimization, violations of these parity bits signal contradictions in the reasoning chain, prompting iterative adjustments until all parity relations (logical consistencies) are satisfied.

This resembles the *Grandor Chase decoder* paradigm in information theory, where bits are iteratively flipped until all parity checks are satisfied. Here, logical constraints $\mathcal{C}_{ij}$ play an analogous role to parity relations, and consistency enforcement acts as a *reasoning-parity check* that iteratively adjusts answers until the collective chain of reasoning achieves global coherence. In this view, *parity* in AI reasoning corresponds to maintaining causal and logical alignment across multiple inferential steps.

LoCo-LMs (Calanzone et al., 2025) introduces losses based on neuro-symbolic reasoning that teach LLMs logical consistency by maximizing the probabilities that beliefs comply with provided logical constraints during training. REPAIR (Liu et al., 2025c) proposes universal frameworks to quantify compositional logical consistency, including fundamental properties of transitivity, commutativity, and negation invariance, refining noisy pairwise comparisons using rank aggregation and augmenting logically consistent comparisons for instruction-tuning.

### 4.4 Practical patterns operationalize the unified objective

Building on the above formalisms, recent advances introduce approaches to operationalize the unified framework in practical systems (as shown in Figure 7). (1) Program-Aided Language Models (PAL) separate planning from execution by generating code that external engines run, improving precision on arithmetic and algorithmic tasks. (2) Chain-of-verification and self-correction (CoVe) treat initial answers as hypotheses and revise them after targeted checks, reducing hallucinations. (3) Neuro-symbolic integration combines neural generation with symbolic rules or reasoners to enforce global coherence. (4) Multimodal and tool grounding adds external evidence and computation (e.g., search, calculators, databases) to stabilize answers.

**PAL**  Because likelihood training fundamentally favors correlation over entailment (Section 4.1), mitigations must outsource computation to verifiable executors. Program-Aided Language Models (PAL) address this by separating reasoning from execution by having LLMs generate programs in formal languages while delegating computations to external executors (Gao et al., 2023). The approach follows the paradigm

$$\text{Ans}(q) = \text{Exec}\big(\text{LM}_{\text{prog}}(q)\big),$$

where the model writes code that a Python interpreter executes to produce answers. This hybrid approach leverages LLMs' strengths in understanding and planning while avoiding their weaknesses in arithmetic and symbolic manipulation, achieving substantial improvements (e.g., $\sim$72% vs. 55–65% for CoT on GSM8K) (Gao et al., 2023). PAL eliminates the notorious computational errors in LLMs while maintaining their flexibility in problem decomposition and code generation, making it particularly effective for mathematical and algorithmic reasoning tasks. Compared to approaches such as SymbCoT, which still keep much of the reasoning inside the language model, solver-executed methods move more of the inference to external symbolic engines (Lyu et al., 2023; Silver et al., 2024). This reduces the burden on the LLM and can provide stronger guarantees of correctness, although it may introduce challenges in translating natural language problems into formal representations.

**CoVe**  Since models learn dataset correlations rather than causal structure (Section 4.2), Chain-of-Verification (CoVe) introduces explicit verification loops. CoVe treats initial LLM responses as hypotheses requiring systematic verification before acceptance, implementing explicit error-detection loops through a generate-verify-revise cycle (Dhuliawala et al., 2023). The approach decomposes answers into verifiable claims, checks each claim independently using external tools or separate model calls, and revises based on verification results. Self-correction mechanisms similarly enable models to detect and repair reasoning errors through multiple verification rounds.

**Neuro-symbolic integration**  To overcome the sample complexity barrier for compositional reasoning (Theorem 4), neuro-symbolic methods inject explicit rule-based constraints. They pair LLMs with symbolic reasoners to combine flexible language understanding with rule-checked inference (Vsevolodovna & Monti, 2025). Two common patterns are: (i) LLM proposes, symbolics verify, where generated steps are checked for consistency, and (ii) symbolics constrain, LLM generates, where rules or knowledge bases shape generation. This reduces contradictions and enforces precise entailment (Hao et al., 2023).

**Multimodal grounding and tool integration.**  Multimodal grounding augments LLMs with visual, audio, and other signals to reduce ambiguity and add constraints (Zhang et al., 2023b). Tool integration lets models call external resources, calculators, search, databases, and specialized software, so answers rely on evidence and precise computation rather than parametric memory.

## 5   Why Does Retrieval Fail?

Retrieval-augmented generation (RAG) has emerged as a critical paradigm for enhancing LLMs by integrating external knowledge sources (Lewis et al., 2020a). However, the performance of LLMs in RAG settings is heavily contingent upon the quality and relevance of the retrieved information (Gupta et al., 2024). Formally, the retrieved passages condition the model's solution space by acting as contextual evidence during generation:

$$P(y \mid x, D_r) = \int P(y \mid x, z)\, P(z \mid x, D_r)\, dz.$$

Here, $x$ denotes the query or input prompt, $D_r$ represents the set of retrieved documents, $z$ corresponds to latent reasoning variables, and $y$ is the generated response. This formulation highlights that retrieval not only supplements the model with external information but also constrains its generative distribution. As a result, the LLM's output becomes highly dependent on the retrieved context, which effectively steers the token-level generation process and restricts the accessible solution space to regions consistent with $D_r$.

Two fundamental dimensions have been central to the research community's efforts in advancing RAG systems: (1) improving the *quality of retrieval*, and (2) enhancing *LLM robustness to misleading or contradictory*

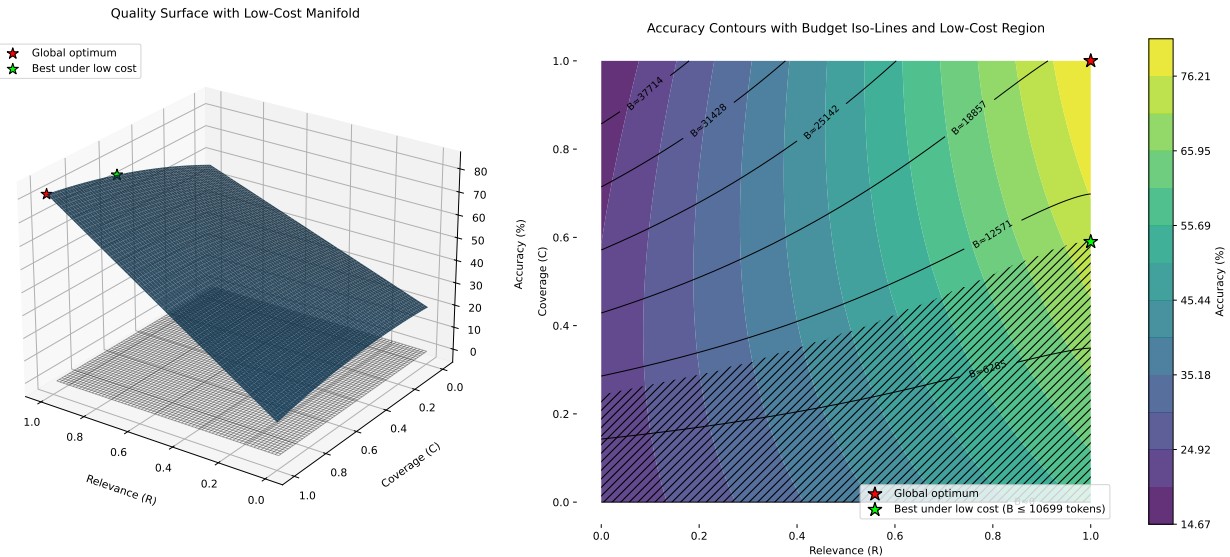

Figure 8: Performance landscape over relevance-coverage space with budget iso-lines and hatched low-cost region. The 3D surface shows how relevance and coverage affect generative quality under token limits, contrasting global (red) and budget-constrained (green) optima.

*retrievals.* These two axes capture the core limitations that dictate the overall reliability and factual consistency of RAG pipelines. In this section, we primarily focus on these dimensions. First, we discuss challenges related to retrieval quality, i.e., how the precision, recall, and contextual alignment of retrieved documents influence generative performance. Second, robustness of LLMs to misretrieved or contradictory information, where retrieval errors can mislead the model's reasoning trajectory or bias its token-level generation toward incorrect conclusions.

## 5.1 Why retrieval fails before generation begins

Although retrieval modules are often treated as static components, their quality fundamentally governs the downstream reasoning and generation capacity of large language models. Even small degradations in retrieval precision or contextual alignment can cascade into incoherent or factually inconsistent generations. To understand these effects, we discuss fundamental issues that result in compromise of retrieval quality and hence generation.

### 5.1.1 Relevance-coverage dilemma

A fundamental constraint in RAG systems arises from the finite *token budget B* imposed by the LLM's context window. Given a retrieved set $D_r$, retrieval must satisfy

$$\sum_{d \in D_r} \text{len}(d) \le B,$$

forcing a trade-off between *relevance* and *coverage*. Precision-oriented retrievers (e.g., dense bi-encoders) maximize local similarity $\text{Sim}(q, d)$ to ensure high relevance, yet often omit peripheral or multi-hop evidence required for compositional reasoning. Conversely, recall-oriented retrievers expand $D_r$ to improve coverage but inject semantically weak or redundant passages, consuming valuable tokens and degrading the signal-to-noise ratio of the conditioning context. As shown in Figure 8, retrieval performance forms a constrained surface where increasing coverage or relevance independently cannot guarantee optimal generation quality. The feasible low-cost region illustrates how token budgets inherently limit the achievable balance between these two dimensions. This tension directly impacts $P(y \mid x, D_r)$, as excessive precision limits inferential completeness, while excessive coverage induces contextual dilution and generation drift.

Recent works address this balance through *structure-aware retrieval*, where compact graph or cluster representations encode semantically related content within fewer tokens (Cheng et al., 2025). Graph-based retrievers leverage node-level relevance propagation to preserve contextual diversity under budget constraints (Zhu et al., 2025a; Guo et al., 2025b). Similarly, KG-guided and hierarchical retrieval methods compress correlated information before injection (Li et al., 2024e). While such designs partially alleviate redundancy, inherent bottlenecks persist: incomplete graph connectivity limits factual recall, graph linearization reintroduces token overhead, and traversal-based ranking adds computational latency. Hence, the relevance–coverage dilemma remains a structural limitation of token-bounded retrieval pipelines.

### 5.1.2 Information discretization in token-bounded retrieval

Retrieval accuracy in RAG pipelines is fundamentally constrained by the imperfect mapping of query to Documents, i.e., $q \mapsto D_r$ under a bounded token context $B$. Given segmented documents $\mathcal{C}(D) = \{c_1, \ldots, c_m\}$, retrieval solves

$$D_r^* = \arg \max_{D_r \subseteq \bigcup_D \mathcal{C}(D)} \sum_{c \in D_r} s(q, c) \quad \text{s.t.} \quad \sum_{c \in D_r} \text{len}(c) \leq B.$$

This optimization assumes chunk independence, yet most natural language dependencies are cross-chunk. When semantically linked units (e.g., condition-action pairs) straddle boundaries, $s(q, c_i)$ and $s(q, c_{i+1})$ both drop below threshold, eliminating essential context. The resulting *fragmentation loss* causes retrieval to return incomplete evidence, which the generator then over-interprets as complete input, degrading factual precision and calibration (Qian et al., 2024; Lu et al., 2025).

Hierarchical and chunk-free retrievals alleviate but do not eliminate this degradation. Hierarchical chunking dynamically merges semantically adjacent spans to preserve local coherence (Lu et al., 2025), while chunk-free in-context retrieval embeds full documents and extracts spans directly (Qian et al., 2024; Brådland et al., 2025). Both improve retrieval fidelity yet remain bounded by the token limit $B$: even if the retriever identifies coherent evidence, it must still serialize it into a finite context, discarding residual dependencies.

Beyond fragmentation, *relevance degradation* arises from ranker-level phenomena. As the retriever expands $D_r$ to increase coverage, average similarity $\bar{s}(q, D_r) = \frac{1}{|D_r|} \sum_{c \in D_r} s(q, c)$ decreases, reducing signal-to-noise in the prompt. Document-level mismatches and embedding sensitivity further compound error, as minor paraphrases can alter embedding neighborhoods, leading to unstable recall@k and inconsistent grounding (Cao et al., 2025; Park & Lee, 2024).

Retrieval accuracy is bounded not only by model capacity but by *representational compression under token budgets*. Evidence selection and chunking convert a continuous knowledge space into a discrete, truncated context sequence; once information is omitted or fragmented, the generator's posterior $P(y \,|\, x, D_r)$ cannot recover it. Thus, despite improved chunking, hierarchical merging, or structure-aware retrieval, the *irreducible gap between real-world evidence continuity and token-bounded context serialization* remains the central bottleneck in RAG relevance.

**Ranking failures and positional bias**

Retrieval in RAG systems suffers from two intertwined limitations: *rank truncation*, where relevant documents fall below the retrieval cutoff, and *positional bias*, where the Large Language Model (LLM) underutilizes information even when it is retrieved. Let $s(q, d)$ denote the retrieval score and $S^\star \subset \mathcal{D}$ the set of truly relevant documents. The retriever selects

$$D_r = \{d_{(1)}, \ldots, d_{(k)}\} = \arg \max_{|D_r| = k} \sum_{d \in D_r} s(q, d),$$

yielding an exposure likelihood reflected by the retrieval metric Recall@$k = \frac{|D_r \cap S^\star|}{|S^\star|}$. Whenever $\exists\, d^\star \in S^\star$ with rank $> k$, the model's generation is upper-bounded by missing evidence, regardless of reasoning capacity. This manifests as the *scattered-evidence failure*, where the rationale spans multiple documents but only a subset appears in $D_r$, fragmenting multi-hop inference. Even multi-passage fusion techniques (e.g., FiD-style decoding) remain constrained by ranker exposure and token budgets (Izacard & Grave, 2020; Agrawal et al., 2024).

Positional or order bias further compounds this issue. Long-context analyses reveal a consistent "lost-in-the-middle" effect: tokens located near the start and end of the prompt receive disproportionately higher attention weights than mid-sequence tokens. Empirically, relocating gold passages from the extremes to mid-context reduces answer recall, confirming a primacy-recency weighting curve $w(t)$ with $w(0), w(1) \gg w(0.5)$ (Liu et al., 2023c; Li et al., 2024b). This bias arises from causal masking and positional encoding decay in Transformer layers—RoPE-based encodings exhibit long-range attenuation that weakens middle-context dependency modeling (Kazemnejad et al., 2023). Consequently, document ordering within the retrieved set materially affects output; reordering equally scored passages alters factual accuracy, demonstrating the model's sensitivity to serial position (Cuconasu et al., 2025; Hsieh et al., 2024).

The combined retrieval-generation failure probability can be bounded as

$$\Pr[\text{failure}] \geq 1 - \Pr[d^\star \in D_r] \cdot \Pr[\text{LLM attends to } d^\star],$$

illustrating that generation fidelity depends jointly on exposure (ranking) and utilization (attention). While reranking, iterative decoding, and attention-based reordering alleviate some degradation (Reddy et al., 2024; Li et al., 2024b), the bottleneck persists: the serialized, token-bounded prompt enforces both top-$k$ truncation and position-dependent conditioning. Once critical evidence is omitted or placed unfavorably within the context, the posterior $P(y \mid x, D_r)$ cannot recover it. Retrieval quality in RAG is therefore limited not only by what is retrieved, but by *where* and *how* it is positioned within the LLM's representational scope.

### 5.1.3 Memory contamination.

RAG systems inherit a critical vulnerability from their external memory: any retriever that ranks based on similarity $s(q, d) = \langle f_Q(q), f_D(d) \rangle$ can be adversarially biased through minimal perturbations of the knowledge base (Xian et al., 2024; Zou et al., 2025). Let the retrieval corpus be $\mathcal{D} = \mathcal{D}_{\text{clean}} \cup \mathcal{P}$, where $\mathcal{P}$ is a small poisoned subset. The retriever returns

$$D_r(q) = \arg \operatorname*{top-k}_{d \in \mathcal{D}} s(q, d),$$

and the generator produces an answer distribution $P(y \mid x{=}q, D_r(q))$. The attacker seeks to inject poisons that maximize

$$\mathcal{L}_{\text{attack}}(\mathcal{P}; q^\star, a^\star) = \Pr_{\text{retrieval}}\big[\mathcal{P} \cap D_r(q^\star) \neq \varnothing\big] \cdot \Pr_{\text{gen}}\big[y{=}a^\star \mid q^\star, D_r(q^\star)\big],$$

subject to a small injection budget $|\mathcal{P}| = N \ll |\mathcal{D}_{\text{clean}}|$. Empirically, *PoisonedRAG* demonstrates that inserting only five poisoned documents per target query in million-scale KBs achieves $\sim 90\%$ attack success rates, establishing a highly efficient adversarial regime (Zou et al., 2025).

*Orthogonal augmentation* is the key mechanism: poisoned items are optimized such that

$$s(q^\star, p) \gg s(q^\star, c), \quad \text{for many } c \in \mathcal{C}_{\text{clean}}, \quad \text{while} \quad \langle f_D(p), f_D(c) \rangle \approx 0,$$

i.e., they remain orthogonal to clean evidence yet maximally aligned with the query embedding. This displaces authentic evidence from the top-$k$ frontier under the same token budget $B$, corrupting retrieval without overtly altering corpus semantics. Because retrievers reuse the same embedding manifold for paraphrased queries $q' \sim \mathcal{Q}(q^\star)$, poisons generalize across lexical variants, producing a *knowledge corruption cascade* (Chang et al., 2025; Ha et al., 2025):

$$\Pr[\text{poison retrieved for } q'] \approx \Pr[\text{poison retrieved for } q^\star], \quad \text{for most } q' \in \mathcal{Q}(q^\star),$$

thereby inducing systemic bias in subsequent generations.

Post-hoc defenses, including activation-level poisoning detectors (Tan et al., 2024; Zhang et al., 2025a) and forensic traceback via removal-based counterfactuals, can flag compromised passages but do not eliminate the structural vulnerability.

The fundamental risk persists: retrieval operates over a mutable, token-bounded corpus. Any poisoned $p$ occupying a top-$k$ slot both *displaces clean evidence* and *dominates generation conditioning*. The lower bound

$$\Pr[\text{failure}] \geq \Pr[p \in D_r(q^\star)] \, \Pr\big[P(y{=}a^\star \mid q^\star, \{p\}) > \theta\big]$$

captures the multiplicative nature of exposure and utilization vulnerabilities. Hence, memory contamination remains an intrinsic limitation of retrieval-based augmentation; the system's factual reliability is only as secure as the integrity of its external memory.

## 5.2   To what extent do LLMs believe everything they read?

While the previous section focused on the fundamental causes that degrade retrieval quality, this section examines the LLM's own capacity to withstand such imperfections. Even with flawed or incomplete evidence, an ideal model should reason cautiously, calibrate confidence, and prioritize consistent information. In practice, however, large language models show limited robustness to imperfect retrieval, often absorbing noise or contradictions without discrimination. This section examines the boundaries of that resilience.

**Attention is distraction too**

In retrieval-augmented LLMs, the same mechanism that enables contextual reasoning, multi-head self-attention, introduces a fundamental vulnerability: *distraction by irrelevant or misleading context* (Amiraz et al., 2025; Shi et al., 2023; Yang et al., 2025). Let the input sequence be a concatenation of the query $q$ and retrieved passages $D_r = \{d_1, \ldots, d_k\}$. Within each Transformer layer $l$, attention operates as

$$A^{(l)} = \text{softmax}\left(\frac{Q^{(l)}K^{(l)\top}}{\sqrt{d}}\right)V^{(l)},$$

where $Q^{(l)} = W_Q^{(l)}h_q$ and $K^{(l)} = W_K^{(l)}h_{D_r}$. The effective relevance of each passage to the generation process is determined not by retrieval similarity $s(q, d_i)$ but by the aggregate attention mass

$$\alpha_i = \frac{1}{L}\sum_{l=1}^{L}\sum_{h \in \mathcal{H}} \text{mean}\left(A_{q \to d_i}^{(l,h)}\right),$$

representing how strongly the model attends to passage $d_i$. When semantically related distractors $d_j$ satisfy $\alpha_j \approx \alpha_i$ or even $\alpha_j > \alpha_i$ for relevant $d_i$, the conditional distribution

$$P(y \mid q, D_r) = \sum_{i=1}^{k}\alpha_i P(y \mid q, d_i)$$

is biased toward irrelevant evidence, even if the retrieval itself is correct. This attention-relevance mismatch systematically distorts the generative posterior and yields confident but erroneous reasoning.

Empirical analyses confirm this vulnerability: adding a single irrelevant passage can reduce accuracy by up to 30%, with degradation correlating to the distractor's cosine similarity in embedding space (Yang et al., 2025; Amiraz et al., 2025). Such distractibility arises because pretraining and supervised fine-tuning maximize next-token likelihood conditioned on *all* tokens' attention is optimized for coverage, not discrimination. Consequently, attention weights $\alpha_i$ do not encode epistemic confidence, and the model's aggregation of contextual signals remains indiscriminate.

Mathematically, distraction is inherent to the softmax attention mechanism. For tokens $t_1, t_2$ from relevant and irrelevant passages, the ratio of normalized weights is

$$\frac{\alpha_{t_2}}{\alpha_{t_1}} = \exp\left(\frac{q^\top (k_{t_2} - k_{t_1})}{\sqrt{d}}\right),$$

implying that small perturbations in inner-product similarity exponentially amplify attention imbalance. In high-dimensional spaces, spurious query-key alignments yield over-attention to distractors, and the resulting activations propagate across layers, entangling noise and evidence. Since subsequent layers operate on these contaminated representations, post-hoc filtering cannot easily reverse the effect.

Mitigation thus demands data-driven attention calibration. Robust training requires examples where irrelevant passages $D_r^-$ coexist with relevant ones $D_r^+$, enforcing selective focus through penalties on non-relevant

attention mass:

$$\mathcal{L}_{\text{robust}} = -\sum_{(q, D_r, y)} \log P(y \mid q, D_r) + \lambda \sum_{d_j \in D_r^-} \alpha_j.$$

Such training, explored in noise-augmented fine-tuning and entailment-consistency filtering (Xiang et al., 2024; Yoran et al., 2023), reduces distraction but at significant computational cost, requiring large curated datasets and layerwise attention attribution. Fundamentally, however, the vulnerability persists: attention aggregates information linearly across all tokens and lacks an intrinsic mechanism to down-weight misinformation. Once a distractor occupies token space within the fixed context window, its representation is fused into the model's latent state, irreversibly influencing generation. Thus, the same attention mechanism that enables flexible reasoning also guarantees susceptibility to irrelevant or misleading context, a structural limitation of current LLM architectures.

**Parametric and retrieved knowledge conflicts**

RAG generation performs an implicit *source arbitration* between an LLM's parametric prior and its retrieved evidence. Let $s_{\text{param}}(y \mid q)$ denote the closed-book logit for answer $y$, and let $s_{\text{ctx}}(y \mid q, D_r)$ denote the logit obtained when conditioning on the retrieved passages $D_r$. We estimate $s_{\text{ctx}}$ and $s_{\text{param}}$ via paired forward passes with and without retrieved context, following standard contrastive attribution protocols. The model's decision can be abstracted as:

$$y^\star = \arg\max_y \left[ \lambda(q) s_{\text{param}}(y \mid q) + (1 - \lambda(q)) s_{\text{ctx}}(y \mid q, D_r) \right],$$

where $\lambda(q) \in [0, 1]$ is an *implicit trust weight* influenced by entity familiarity and prompt framing. Empirically, $\lambda(q)$ varies sharply: models rely on parametric memory for well-known facts but defer to external context when internal confidence is low (Du et al., 2024; Wu et al., 2024b).

Under *knowledge conflict* ($y_{\text{param}} \neq y_{\text{ctx}}$), dominance follows the margin

$$\Delta(q, D_r) = \left[ s_{\text{param}}(y_{\text{param}}) - s_{\text{param}}(y_{\text{ctx}}) \right] - \left[ s_{\text{ctx}}(y_{\text{param}}) - s_{\text{ctx}}(y_{\text{ctx}}) \right],$$

whose sign determines which source prevails. Because both scores depend on context order and evidence composition, small perturbations in $D_r$ or prompt layout can flip $\text{sign}(\Delta)$, producing unpredictable reliance on either source. Benchmarks such as *ConflictBank* and *ClashEval* report all four regimes, model-right/context-wrong, model-wrong/context-right, both wrong, and both right across architectures and domains (Su et al., 2024; Wu et al., 2024b).

Mechanistically, *attention mediates the arbitration*: $s_{\text{ctx}}$ emerges from token-level aggregation where retrieved passages accumulate attention mass that can amplify or suppress the parametric signal. Cutting or reweighting early attention heads reduces context dominance when priors are strong, revealing that the LLM's fusion of sources is coherence-driven, not truth-driven (Jin et al., 2024). Even with realistic, time-updated corpora, these behaviors persist, retrieval may override correct priors or reinforce outdated ones (Kortukov et al., 2024).

Mitigation strategies fall into three limited families: **Training-time biasing** involves fine-tuning with conflict-labeled data to calibrate $\lambda(q)$, which is effective but costly and domain-sensitive. **Inference-time arbitration** uses dual decoding or controller policies to compare parametric and contextual outputs via entailment or consistency, offering robustness yet being latency-heavy and prone to over-deferring to fluent context (Wang et al., 2024a). **Structure-aware fusion** applies graph- or claim-level gating to down-weight $s_{\text{ctx}}$ when contradictions arise, which reduces overt clashes but depends on the quality of structured evidence (Kortukov et al., 2024).

**Why this remains fundamental.** The parametric prior is a frozen distribution $s_{\text{param}}(y \mid q)$; retrieval injects a mutable, position-biased likelihood $s_{\text{ctx}}(y \mid q, D_r)$. Their mixture weight $\lambda(q)$ is emergent, not optimized for factuality but for token-likelihood consistency. Hence, minor shifts in $D_r$, prompt phrasing, or ranking can cross the decision boundary

$$\Delta(q, D_r) = 0,$$

causing unstable source arbitration (Xu et al., 2024c). Even after calibration or structured fusion, attention still aggregates conflicting signals under likelihood maximization, leaving $\lambda(q)$ ungrounded. Hence, all current LLMs, whether closed-book, open-book, or hybrid, remain vulnerable to contradictions; resolving them requires explicit verifiers or external truth supervision beyond probabilistic conditioning.

**Pipeline limitations**

A user query $q$ is often an underspecified representation of the underlying intent $z$ (Srinivasan et al., 2022). Let $p(z \mid q)$ denote the distribution over possible interpretations. Ideally, retrieval should maximize expected coverage across intents:

$$D_r^\star \in \arg \max_{|D_r| \leq k} \mathbb{E}_{z \sim p(z|q)} \sum_{d \in D_r} s(q(z), d),$$

but most RAG pipelines collapse this distribution to a single interpretation, approximating $p(z \mid q) \approx \delta(z = z_0)$. Methods such as *RQ-RAG* and *Plan-RAG* attempt to broaden coverage by generating rewrites $\{q_i\}_{i=1}^m$, yet rewriting introduces new error sources and increases ranking complexity (Chan et al., 2024; Lee et al., 2024a; Li et al., 2024d).

The core theoretical gap is that attention-based fusion performs an implicit mixture

$$P(y \mid q, \{D_r^{(i)}\}) = \sum_{i=1}^m \sum_j \alpha_{i,j} \, P(y \mid q, d_j^{(i)}),$$

where $\alpha_{i,j}$ are attention weights, whereas the Bayes-optimal decision marginalizes over latent intents:

$$y^\star = \arg \max_y \mathbb{E}_z \left[ \log P(y \mid q(z), \, D_r(z)) \right].$$

These coincide only if $\alpha_{i,j} \propto p(z \mid q)$ and passages are conditionally independent; conditions rarely met. Two structural limitations follow: (1) **non-identifiability of intent**, because without explicit modeling the model cannot determine which latent intent to prioritize; and (2) **unprincipled fusion**, in which attention merges token-level signals without grounding in a calibrated posterior over intents, optimizing for fluency rather than correctness (Izacard & Grave, 2020; Ye et al., 2023a; Chan et al., 2024; Verma et al., 2024).

Retrieval fragility thus once again exemplifies the underlying triad of limitations on LLM performance under bounded resources. The relevance-coverage dilemma embodies finite information capacity: token budgets force lossy compression of evidence. Ranking failures reflect statistical insufficiency: semantic drift accumulates as retrieval breadth increases. Adversarial contamination exploits computational constraints: orthogonal augmentation defeats similarity-based ranking. Thus, RAG systems inherit fundamental limits rather than transcending them.

## 6 When Seeing Fails to Mean Understanding

By grounding LLMs in perceptual experience, i.e., enabling direct interpretation of visual and auditory inputs, multimodal LLMs (MLLMs) are posited to overcome the hallucinations and abstraction biases that constrain text-only systems (Hurst et al., 2024; Tong et al., 2024a; Wu et al., 2025d). DeepMind's Flamingo architecture was designed to "bridge powerful pretrained vision-only and language-only models" with the goal of enabling few-shot visual reasoning across diverse tasks without fine-tuning (Alayrac et al., 2022). Similarly, OpenAI's GPT-4V(ision) was promoted for its "grounded visual understanding" through the ability to "encode, integrate, and reason over arbitrarily interleaved language and vision signals" (Hurst et al., 2024). Google's Gemini models have positioned multimodality as central to overcoming the grounding problem, i.e., the lack of correspondence between abstract semantic content and real physical objects (Team et al., 2023).

The rationale is intuitive: if text-only models falter from a lack of perceptual grounding, then integrating direct sensory input should alleviate these failures, allowing vision to anchor language and richer modalities to yield more reliable reasoning (Xu et al., 2025c). Yet a paradox emerges from empirical evaluation, i.e., despite richer inputs, MLLMs inherit and often amplify the fundamental limitations of their language-model

backbones (Tong et al., 2024a; Cui et al., 2023). The holistic evaluation of GPT-4V shows that regional, language, and prompt framing biases persist, showing that visual input does not eliminate inductive bias from the training data (Cui et al., 2023; Brin et al., 2024; Senkaiahliyan et al., 2023). This is primarily because most MLLMs are built by coupling pretrained vision and language models through modality adapters (Li et al., 2023b; Alayrac et al., 2022). Consequently, the inductive and representational biases embedded in these pretrained components propagate into downstream MLLMs (Tong et al., 2024a), as similarly observed in text models where encoder-level failures transfer to generative tasks (Tong et al., 2023). Furthermore, research on compositional reasoning finds that state-of-the-art MLLMs exhibit the same brittleness documented in text-only models (Ahn et al., 2025; Dziri et al., 2023). The models struggle with systematic generalization, fail on minor perturbations to visual scenes(Zhang et al., 2025e; Clusmann et al., 2025b;a), and collapse when forced to integrate multiple reasoning steps across modalities. Perhaps most troublingly, multimodal architectures introduce novel failure modes, such as visual object hallucinations stemming from "over-reliance on bag-of-objects representations and language priors" (Li et al., 2025c), that are absent in unimodal systems.

Multimodality does not resolve the fundamental computational and epistemic limits of LLMs (Wang et al., 2025b). Apparent benchmark gains often mask persistent brittleness, as visual inputs introduce new bottlenecks while preserving pretrained linguistic biases. This highlights the need to examine architectural constraints and representational failures that govern what these models cannot do, regardless of sensory input.

## 6.1 Cross-modal bottlenecks and linguistic priors

Despite the integration of vision encoders, audio processors and sensory modules, MLLMs exhibit *language dominance*, where the linguistic representations systematically dominate, distort, and/or compress non-linguistic modalities. This dominance is demonstrated through four fundamental mechanisms: (i) representation imbalance in learned embeddings, (ii) alignment noise from vision-language pretraining, (iii) information loss at modality fusion boundaries, and (iv) semantic distortion from tokenization granularity mismatches.

### 6.1.1 Representation imbalance

MLLMs typically project visual features into a token space compatible with pretrained language models, but this projection is asymmetric and corrupts visual semantic integrity (Wu et al., 2025c). Let $\mathbf{V} \in \mathbb{R}^{n_v \times d_v}$ represent a sequence of $n_v$ visual tokens with dimension $d_v$, and $\mathbf{T} \in \mathbb{R}^{n_t \times d_t}$ represent a sequence of $n_t$ text tokens with dimension $d_t$. In the architectures like LLaVa (Liu et al., 2023b) and BLIP-2 (Li et al., 2023b), a trainable projection layer $\mathbf{W}_{\mathrm{proj}} : \mathbb{R}^{d_v} \to \mathbb{R}^{d_t}$ maps visual embeddings into text token space:

$$v_i' = W_{\mathrm{proj}} v_i + b, \quad v_i \in \mathbf{V} \tag{42}$$

The feature dimensions between the two components are aligned using a projection layer (Liu et al., 2023b). A two-layer MLP enhancing the vision-language connector representation can improve multimodal capabilities over simple linear projections (Liu et al., 2024b). However, the optimization objective used to train $\mathbf{W}_{\mathrm{proj}}$ is principally text generation, specifically minimizing the causal language modeling loss via modifications in vision to language mapping rather than joint representational adaptation. Empirical analysis reveals the extent of representational imbalance. In VideoLLaMA-7B, output tokens attend to text tokens 157 times more than to visual tokens on a per-token basis (Wu et al., 2025a). This observation of linguistic dominance, driven by dataset and parameter imbalance, can be formalized as the following empirically testable claim.

**Proposition 2.** *(Representational dominance). Let $\mathcal{H}_v$ denote the hypothesis class of visual representations and $\mathcal{H}_t$ denote the hypothesis class of text representations in an MLLM with a frozen pretrained LLM backbone. If the pretraining corpus $\mathcal{D}_{\mathrm{LLM}}$ has cardinality $|\mathcal{D}_{\mathrm{LLM}}| \gg |\mathcal{D}_{\mathrm{align}}|$, where $\mathcal{D}_{\mathrm{align}}$ is the vision-language alignment dataset, then the effective capacity of $\mathcal{H}_t$ dominates $\mathcal{H}_v$ in the sense that:*

$$\mathbb{E}_{(\mathbf{v},\mathbf{t}) \sim \mathcal{D}_{\mathrm{test}}} \left[ \|\nabla_{\mathbf{v}} \mathcal{L}\|_2 \right] \ll \mathbb{E}_{(\mathbf{v},\mathbf{t}) \sim \mathcal{D}_{\mathrm{test}}} \left[ \|\nabla_{\mathbf{t}} \mathcal{L}\|_2 \right], \tag{43}$$

*where $\mathcal{L}$ is the task loss and gradients are measured with respect to visual and text token representations.*

*The frozen LLM parameters encode strong linguistic priors from $\mathcal{D}_{\mathrm{LLM}}$. During multimodal finetuning, updates are confined to the adapter parameters and projection layer $\mathbf{W}_{proj}$, which yields a posterior dominated by this prior. Since $|\mathcal{D}_{\mathrm{LLM}}| \gg |\mathcal{D}_{\mathrm{align}}|$, the gradient magnitude w.r.t visual empirical is suppressed relative to text embeddings, consistent with empirical findings (Wu et al., 2025a).*

Pretrained LLMs encode strong linguistic priors, causing MLLMs to overrely on text while underutilizing visual inputs (Liu et al., 2025a; Tong et al., 2024a). Training paradigms further prioritize textual tokens, relegating images to a subordinate representational subspace and limiting their dynamic integration (Wu et al., 2025a). To counteract this trend, recent work proposes modifying the instruction-tuning phase with targeted reward functions designed to encourage a more balanced use of both modalities (Liu et al., 2025a).

### 6.1.2 Alignment noise

Most contemporary MLLMs rely on vision encoders pretrained with contrastive objectives, such as CLIP (Contrastive Language Image Pretraining) (Radford et al., 2021). CLIP learns a joint embedding space by maximizing the similarity between matched image-text pairs while minimizing similarity for mismatched pairs:

$$\mathcal{L}_{\mathrm{CLIP}} = -\frac{1}{N} \sum_{i=1}^{N} \log \frac{\exp\left(\mathrm{sim}(v_i, t_i)/\tau\right)}{\sum_{j=1}^{N} \exp\left(\mathrm{sim}(v_i, t_j)/\tau\right)}, \tag{44}$$

where $v_i$ and $t_i$ are the visual and text embeddings of sample $i$, $\mathrm{sim}(\cdot,\cdot)$ is the cosine similarity, $\tau$ is the temperature, and $N$ is the batch size. While effective for retrieval, this objective introduces *semantic drift* (Spataru et al., 2024), which is that the learned visual representations are not grounded in perceptual properties of objects, but rather in their co-occurrence statistics with text descriptions from web-scraped datasets. The concept of semantic drift can be illustrated through the following bound, which formally connects the representational distortion to the statistical divergence of the training data.

**Proposition 3.** *(Alignment drift bound). Let $f_v : \mathcal{X}_v \to \mathbb{R}^d$ and $f_t : \mathcal{X}_t \to \mathbb{R}^d$ be the CLIP vision and text encoders, respectively. Let $p_{true}(v,t)$ denote the true joint distribution of visual concepts $v$ and their linguistic descriptions $t$, and let $p_{data}(v,t)$ denote the empirical distribution in web-scraped data. Then the expected semantic drift $\Delta$ for a concept $c$ satisfies:*

$$\Delta(c) := \mathbb{E}_{v \sim p_{true}(v|c)}\left[\|f_v(v) - \mu_c^{true}\|_2\right] \geq \sqrt{D_{KL}(p_{true}(v|c) \,\|\, p_{data}(v|c))} \cdot \sigma_c \tag{45}$$

*where $\mu_c^{true}$ is the true perceptual centroid of concept $c$, and $\sigma_c$ is the within-concept standard deviation.*

*This inequality states that the expected distortion between learned and true perceptual representations grows proportionally to the divergence between real-world and dataset distributions, scaled by within-class variability.*

The embedding spaces inherit the statistical biases, cultural associations, and spurious correlations of their text corpora, and this semantic drift propagates into downstream MLLMs. For example, the concept *"dog"* does not represent the visual entity itself but instead approximates the linguistic descriptions most frequently associated with dogs in internet text.

An alternative approach focuses on improving the quality of the visual representations before they are fused. Generative methods like Masked Image Modeling (MIM), used in models such as BEiT, force the vision encoder to reconstruct masked image patches (Bao et al., 2021).

### 6.1.3 Modality fusion and structural loss

Multimodality fusion in MLLMs typically occurs through late concatenation or attention-based pooling (Laurençon et al., 2024). In architectures such as Flamingo (Alayrac et al., 2022), visual tokens $\{\mathbf{v}'_1, \ldots, \mathbf{v}'_{n_v}\}$ and text tokens $\{\mathbf{t}_1, \ldots, \mathbf{t}_{n_t}\}$ are concatenated and fed into a transformer $\mathbf{Z} = [\mathbf{v}'_1, \ldots, \mathbf{v}'_{n_v}, \mathbf{t}_1, \ldots, \mathbf{t}_{n_t}]$, with cross-modal attention computed as:

$$\mathrm{Attention}(\mathbf{Q}, \mathbf{K}, \mathbf{V}) = \mathrm{softmax}\left(\frac{\mathbf{Q}\mathbf{K}^T}{\sqrt{d_k}}\right)\mathbf{V}, \tag{46}$$

where queries $\mathbf{Q}$ typically come from text tokens and keys/values $\mathbf{K}, \mathbf{V}$ from both modalities. While this is effective, the joint attention over text and a large set of visual tokens creates a representational bottleneck in which fine-grained spatial information cannot be fully preserved, i.e., the vision signal undergoes a form of *lossy compression*, which limits downstream precision.

**Information-theoretic formulation.** Using the Information Bottleneck (IB) principle (Tishby et al., 2000), let $V$ denote visual input, $T$ text input, $Y$ the target output (e.g., caption or answer), and $Z$ the fused representation. The IB objective seeks a compressed representation that maximizes task-relevant information while minimizing total information:

$$\min_{p(z|v,t)} \big[ I(V, T; Z) - \beta I(Z; Y) \big], \tag{47}$$

where $\beta > 0$ controls the trade-off between compression and task performance, and $I(\cdot; \cdot)$ denotes mutual information.

**Proposition 4.** *(Cross-modal information loss). Let $\mathcal{G} = (V, E)$ represent the relational structure of a visual scene, where $V$ are objects and $E$ are spatial or semantic relations. Under attention-based fusion with $k$ attention heads, the mutual information between the fused representation $Z$ and the relational structure $\mathcal{G}$ is bounded by:*

$$I(Z; \mathcal{G}) \leq k \cdot \log(n_v + n_t) + H(\mathcal{G}) - H(\mathcal{G} \mid \text{co-occurrence}), \tag{48}$$

*where $H(\mathcal{G})$ is the entropy of the graph structure and $H(\mathcal{G} \mid \text{co-occurrence})$ is the conditional entropy given object co-occurrence statistics.*

*Attention computes weighted sums over token embeddings, which can be viewed as a rate-distortion compression (Tishby et al., 2000). The capacity is limited by the number of attention heads $k$ and sequence length. Since attention weights rely on dot products of embeddings, they primarily encode token co-occurrence rather than relational structure. By the data processing inequality, $I(V; \mathcal{G}) \geq I(Z; \mathcal{G})$, and the bound follows from finite attention capacity and co-occurrence statistics.*

Pooling visual tokens collapses spatial or temporal structure, whereas cross-attention applied after pooling loses fine-grained interplay between modalities, such as spatial relationships between visual regions, temporal dynamics in video, or hierarchical scene composition. Fusion bottleneck architectures (Nagrani et al., 2021) route cross-modal interactions through a small set of latent tokens, reducing computation but risking loss of relational and visual detail as bottleneck size decreases.

### 6.1.4 Tokenization granularity mismatch

Another mismatch exists between the discrete, symbolic nature of language tokens and the continuous, high-dimensional nature of visual inputs. To interface with language models, visual data must be tokenized, typically through patch embeddings (as in ViTs) or vector quantization (VQ). Non-text modalities contain redundant tokens, limiting their contribution to cross-modal attention (Wu et al., 2025a), referred to as *attention dilution*.

**Patch embeddings and spatial downsampling.** ViTs (Dosovitskiy et al., 2021) divide an image into fixed-size patches $\mathbf{P} \in \mathbb{R}^{(H/p) \times (W/p) \times (p^2 \cdot C)}$, where $H \times W$ is the image resolution, $p$ is the patch size, and $C$ is the number of channels. Each patch is linearly projected into a $d$-dimensional token:

$$\mathbf{v}_i = \mathbf{W}_{\text{patch}} \cdot \text{flatten}(\mathbf{P}_i) + \mathbf{b}_{\text{patch}} \tag{49}$$

This process introduces a *resolution bottleneck*, i.e., fine-grained visual details smaller than the patch size are irreversibly lost. This granularity mismatch means visual information is inherently underrepresented compared to text (Wu et al., 2025c).

**Vector quantization.** VQ-based tokenizers (e.g., VQ-VAE (van den Oord et al., 2017), VQGAN (Esser et al., 2021)) discretize continuous visual features by mapping them to a learned codebook $\mathcal{C} = \{\mathbf{e}_1, \ldots, \mathbf{e}_K\} \subset \mathbb{R}^d$.

Given an encoder output $\mathbf{z}_e(\mathbf{x})$, the quantized representation is:

$$\mathbf{z}_q = \mathbf{e}_k, \quad k = \arg\min_j \|\mathbf{z}_e(\mathbf{x}) - \mathbf{e}_j\|_2.$$

While VQ enables discrete tokenization, it introduces semantic distortion. When a VQ tokenizer is used, multimodal understanding performance is lower compared to using a specialized semantic tokenizer (Jia et al., 2025). The quantization process forces continuous visual manifolds onto a finite discrete set, inducing quantization error:

$$\epsilon_{\text{quant}} = \|\mathbf{z}_e(\mathbf{x}) - \mathbf{z}_q\|_2 \tag{50}$$

that compounds when tokens are processed by downstream models.

**Proposition 5.** *(Codebook collapse and semantic coverage). Let $\mathcal{M} \subset \mathbb{R}^d$ be the manifold of visual features with intrinsic dimension $d_{\mathcal{M}}$ and volume $Vol(\mathcal{M})$. For a codebook of size $K$ trained with quantization loss $\mathcal{L}_{VQ}$, the expected reconstruction error $\mathbb{E}[\epsilon_{quant}^2]$ is lower-bounded by:*

$$\mathbb{E}[\epsilon_{quant}^2] \geq \frac{1}{K^{2/d_{\mathcal{M}}}} \cdot \left(\frac{Vol(\mathcal{M})}{\omega_{d_{\mathcal{M}}}}\right)^{2/d_{\mathcal{M}}} \tag{51}$$

*where $\omega_{d_{\mathcal{M}}}$ is the volume of the unit ball in $d_{\mathcal{M}}$ dimensions. Moreover, if the training distribution is non-uniform, at most $K_{eff} \ll K$ codebook entries are utilized (codebook collapse), degrading the bound by a factor of $K/K_{eff}$. The $K^{-2/d_{\mathcal{M}}}$ scaling reflects the curse of dimensionality. Codebook collapse occurs when gradient updates concentrate on frequently-accessed entries, leaving others untrained.*

VQ suffers from codebook collapse: during training, only a subset of codebook vectors may be utilized, leaving large regions of visual space poorly represented (Yu et al., 2024). Recent works have explored alternatives like FSQ (Finite Scalar Quantization) (Mentzer et al., 2024) and Factorized Quantization (FQ) (Bai et al., 2024c), but these methods still face fundamental granularity mismatch issues. Philosophically, the issue is ontological i.e., language is inherently compositional and discrete (morphemes, words, sentences), while vision is continuous and analog (pixel intensities, spatial gradients, object boundaries). Tokenization schemes attempt to bridge this gap by discretizing vision, but no finite vocabulary can fully capture the infinite variability of visual perceptions.

More sophisticated architectures, such as Q-Former's learnable query bottleneck (Li et al., 2023b) and DeepStack's layer-distributed visual tokens (Meng et al., 2024), enhance cross-modal interaction but still operate within a computational framework optimized for text generation, rearranging rather than resolving linguistic dominance.

## 6.2 Post-training alignment

Multimodal instruction tuning (Liu et al., 2023b; Dai et al., 2023) fine-tunes MLLMs on (image, instruction, response) triplets, often semi-synthetic, to teach complex visual reasoning and dialogue. While effective at aligning outputs with human expectations, this process teaches statistical patterns of desired responses rather than resolving underlying biases: models may reinforce spurious correlations present in synthetic training data (Hosseini et al., 2025). Multimodal limitations thus amplify rather than resolve the unified framework constraints: projection layers compress visual semantics into linguistically-dominated spaces (information capacity limits), caption-mediated training learns co-occurrence rather than perceptual structure (statistical insufficiency), and ambiguous inputs propagate uncertainty bidirectionally (computational undecidability).

## 6.3 Epistemic and cognitive pitfalls

We examine fundamental epistemic failures, the perceptual illusion of grounding, spurious statistical associations masquerading as spatial understanding, and compositional brittleness that defeats systematic generalization, alongside two amplifying factors: cross-modal hallucination and cognitive overfitting to synthetic training data, which rewards dataset exploitation over genuine understanding.

### 6.3.1 Perceptual illusion of grounding

MLLMs are built on the hypothesis of *perceptual grounding* that by processing images (or other modalities) directly, models develop representations anchored in sensory experience rather than purely symbolic abstractions. However, contemporary MLLMs exhibit what we term the *perceptual illusion*, i.e., they appear grounded while continuing to reason over embeddings fundamentally detached from perceptual reality (Cao et al., 2024). The visual embeddings generated are projected into the language model's token space via $\mathbf{W}_{\mathrm{proj}}$, concatenated with text tokens, and processed by the LLM backbone (2). Critically, the LLM never accesses raw pixel intensities, spatial gradients, or any continuous perceptual features; it operates entirely on discrete token sequences in a learned embedding space (5).

**Proposition 6.** *(Symbolic detachment of multimodal representations). Let $\mathcal{P}$ denote the set of perceptually grounded scene properties that often support downstream reasoning, such as 3D spatial layout, object permanence, and visually observable interaction regularities; some causal judgments may depend on such properties but are not reducible to perception alone (Jin et al., 2023).*

$$I(\mathcal{R}_{MLLM}; \mathcal{P}) \leq I(\mathcal{C}; \mathcal{P})$$

*where $\mathcal{C}$ denotes caption semantics and $I(\cdot; \cdot)$ is the mutual information operator. This implies that perceptual properties not describable in captions remain unrepresented, upper-bounded by the mutual information between captions and those properties.*

A natural objection is that if MLLMs only generate text, caption-level representations should suffice. However, this conflates *output modality* with *representational requirements*. Consider the query: "If this object is rotated 90 degrees clockwise, which face will be visible?" While the answer is text, correct reasoning requires internal representations of 3D geometry and mental rotation, perceptual properties $\mathcal{P}$ underspecified in typical captions $\mathcal{C}$. Formally, let $\mathcal{T}$ denote a task requiring reasoning over $\mathcal{P}$ with textual output. If $I(\mathcal{C}; \mathcal{P}) < H(\mathcal{P})$, then optimal performance requires $I(\mathcal{R}_{\mathrm{MLLM}}; \mathcal{P}) > I(\mathcal{C}; \mathcal{P})$, contradicting Proposition 6. Caption-mediated representations may memorize common patterns for high in-distribution performance, yet fail on compositional generalization, systematic spatial or causal reasoning, and adversarial robustness. While MLLMs achieve *behavioral adequacy* (plausible text), they lack *representational adequacy* (internal perceptual models), evident when tasks demand genuine perceptual reasoning beyond pattern matching. Empirical evidence supports this symbolic detachment, demonstrating that even multimodal models like GPT-4V rely predominantly on textual associations rather than direct visual input when predicting human perceptual judgments (Hirano et al., 2024). The illusion of grounding arises because MLLMs excel at tasks where linguistic priors suffice (e.g., object recognition, scene classification) (Abdou et al., 2021) while failing on tasks requiring genuine perceptual inference (e.g., physical stability, spatial navigation, fine-grained visual reasoning) (Rahmanzadehgervi et al., 2025; Hirano et al., 2024; Zhang et al., 2025e; Clusmann et al., 2025a).

### 6.3.2 Grounding and reasoning failures

Even when MLLMs integrate textual and other information (image, video, audio, etc.), the learned associations are predominantly spurious (Hosseini et al., 2025), which means that the associations are mere correlations in training data rather than causal or compositional structures. This manifests most clearly in spatial reasoning tasks, where models learn frequent co-occurrences (e.g., "cat on sofa") but not the underlying geometric or physical relations (Rahmanzadehgervi et al., 2025; Hou et al., 2025). Let $\mathcal{S} = \{o_1, \ldots, o_n, r_{ij}\}$ denote a visual scene with objects $o_i$ and spatial relations $r_{ij}$ (e.g., "on top of," "left of," "inside"). An ideally grounded model would learn a representation $\phi(\mathcal{S})$ that factorizes as:

$$\phi(\mathcal{S}) = \bigotimes_{i=1}^{n} \phi_{\mathrm{obj}}(o_i) \otimes \bigotimes_{i,j} \phi_{\mathrm{rel}}(r_{ij})$$

where $\otimes$ denotes compositional combination and $\phi_{\mathrm{rel}}$ explicitly encodes geometric constraints (e.g., vertical displacement for "on top of"). However, MLLM embeddings instead capture:

$$\psi(\mathcal{S}) \approx \sum_{i,j} w_{ij} \cdot \mathbb{1}[\text{co-occur}(o_i, o_j)]$$

where $w_{ij}$ are learned weights reflecting dataset co-occurrence frequencies and $\mathbb{1}[\text{co-occur}(o_i, o_j)]$ indicates whether objects $o_i$ and $o_j$ appear together in training images.

**Proposition 7.** *(Spurious spatial associations). Let $\mathbf{x}$ denote an image and $\mathbf{c}$ its caption, with $p_{train}(o_i, o_j, r_{ij})$ the joint distribution of objects and relations in training data. Suppose $p_{train}(o_i, o_j, r_{ij}) \neq p_{train}(o_i, o_j) \cdot p_{train}(r_{ij} \mid o_i, o_j)$ (i.e., relations are not conditionally independent of co-occurrence). Then an MLLM trained with maximum likelihood on captions will satisfy:*

$$\mathbb{E}_{(\mathbf{x},\mathbf{c}) \sim p_{test}} [\mathcal{L}(MLLM(\mathbf{x}), \mathbf{c})] \geq \mathbb{E}_{(\mathbf{x},\mathbf{c}) \sim p_{train}} [\mathcal{L}(MLLM(\mathbf{x}), \mathbf{c})] + \lambda \cdot D_{KL}(p_{test} \| p_{train})$$

*where $\lambda > 0$ quantifies sensitivity to distributional shift and $D_{KL}$ is the Kullback–Leibler divergence. This implies that performance degradation scales with the divergence between training and test distributions of object–relation co-occurrences. The model's learned conditional distribution $q(\mathbf{c} \mid \mathbf{x})$ minimizes $D_{KL}(p_{train}(\mathbf{c} \mid \mathbf{x}) \| q(\mathbf{c} \mid \mathbf{x}))$, but generalizes poorly when test data violates training correlations.*

This spurious grounding is evident in compositional reasoning benchmarks. When presented with novel spatial configurations (e.g., "a sofa on top of a cat," inverting the typical arrangement), MLLMs produce nonsensical outputs or refuse to generate descriptions, because the training data contains no instances of this configuration (Thrush et al., 2022; Yuksekgonul et al., 2023). The model learns a statistical pattern, not a geometric relationship that can be flexibly recombined. Research on compositional generalization shows that multimodal models often perform poorly when tested on compositional reasoning tasks that require understanding novel combinations of familiar concepts (Thrush et al., 2022). This limitation arises because the number of possible multi-way combinations grows exponentially, while training coverage is inherently limited (Lake et al., 2017; Bahdanau et al., 2019). It has been shown that even when models are explicitly trained using reinforcement learning (RL), compositional gaps with visual inputs remain (Li et al., 2025b). The failure is architectural: adding visual tokens simply expands the space of spurious correlations without imposing compositional structure.

**Cross-modal hallucinations.** Consider the predictive variance decomposition:

$$\text{Var}[Y \mid V, T] = \text{Var}_V[Y \mid V] + \text{Var}_T[Y \mid T] + 2\,\text{Cov}_{V,T}[Y].$$

The covariance term captures *cross-modal contamination*, where uncertainty or bias in one modality inflates variance in the other. Ambiguous visual inputs ($\text{Var}_V[Y \mid V]$) can trigger *textual hallucinations*, while strong linguistic priors ($\text{Var}_T[Y \mid T]$) can bias visual interpretation (Yue et al., 2024). Object hallucination occurs when MLLMs generate text that is semantically coherent but inconsistent with the image. Contrastive decoding mitigates this by reducing over-reliance on language priors, yet multimodal models only marginally reduce such bias. Ambiguous visual embeddings (e.g., occlusion or low resolution) lead the language model to "fill in" plausible but non-existent objects, whereas spurious visual patterns can induce incorrect textual associations (Chen et al., 2025c). This *bidirectional contamination* worsens with modality imbalance, as the dominant modality's errors propagate unchecked (Leng et al., 2024).

Consequently, a significant line of research has focused on hallucination mitigation. For example, Multi-Modal Mutual-Information Decoding (M3ID) is an inference-time method that explicitly amplifies the influence of the reference image by favoring tokens with higher mutual information with the visual prompt, effectively re-weighting the generation process towards the visual modality (Favero et al., 2024). On the other hand, models like GROUNDHOG ground language to holistic segmentation masks, creating a much richer connection between visual entities and textual phrases. This requires curating new datasets with fine-grained, segmentation-based annotations but can significantly reduce object hallucination by design (Zhang et al., 2024c). While these methods can reduce the frequency of hallucinations, they function primarily as safety overlays rather than foundational cures.

**Cognitive overfitting.** Let $D_{\text{synthetic}}$ be synthetic paired data and $D_{\text{real}}$ true perceptual data. The bias of the fused representation $Z$ scales with their relative sizes:

$$\text{Bias}(Z) \propto \frac{|D_{\text{synthetic}}|}{|D_{\text{real}}|}$$

Heavy reliance on synthetic captions or web-scraped image-text pairs causes MLLMs to overfit anthropomorphic and stylistic priors, propagating cognitive biases (Shumailov et al., 2023). Human-annotated captions often impose subjective interpretations (e.g., "the dog looks happy"), while synthetic captions generated by earlier models amplify errors across generations. As a result, MLLMs reproduce these biases, producing homogenized outputs that reflect training artifacts rather than robust multimodal perception.

## 6.4 Scaling limitations in multimodality

A persistent belief is that the architectural limitations of MLLMs can be overcome through scaling, i.e., adding more parameters, training on larger multimodal datasets, or increasing compute. This may be justified for unimodal systems, where loss scales as a power law with model size, dataset size, and compute, leading to monotonic improvements (Kaplan et al., 2020; Hoffmann et al., 2022b). However, when vision and language are combined, scaling laws become fundamentally fragmented, introducing non-linearities that predictable unimodal scaling cannot accommodate.

### 6.4.1 Divergent scaling laws by modality

For language models, loss (or perplexity) decreases with dataset size $D$ and model size $N$ following a power law (Kaplan et al., 2020; Hoffmann et al., 2022b). Vision models exhibit similar behavior (Zhai et al., 2022). Formally,

$$L_{\text{text}}(N, D) = C_{\text{text}} N^{-\alpha_N^{\text{text}}} D^{-\alpha_D^{\text{text}}},$$
$$L_{\text{vision}}(N, D) = C_{\text{vision}} N^{-\alpha_N^{\text{vision}}} D^{-\alpha_D^{\text{vision}}}, \qquad (\alpha_N^{\text{text}}, \alpha_D^{\text{text}}, \alpha_N^{\text{vision}}, \alpha_D^{\text{vision}} > 0)$$

where $C_{\text{text}}$ and $C_{\text{vision}}$ are modality-specific constants.

When modalities are fused, the naive expectation is that combined loss would follow:

$$L_{\text{multi}}(N, D) = g(L_{\text{text}}(N, D), L_{\text{vision}}(N, D)).$$

Empirical work on mixed-modal scaling laws (Aghajanyan et al., 2023) shows that multimodal performance depends not only on individual modality scaling but also on cross-modal synergy and competition. Consequently, naive parameter or dataset scaling fails to yield uniform improvements, as the slower-scaling modality can dominate the aggregate error. This scaling trend is also observed in native multimodal models (NMMs), which demonstrate scaling laws comparable to modular approaches that employ separate tokenizers and image encoders (Shukor et al., 2025).

**Proposition 8.** *(Fractured multimodal scaling). For a multimodal system combining language and vision, the observed loss $L_{MLLM}(N, D_t, D_v)$ deviates from the linear combination model as:*

$$L_{MLLM} = \lambda L_{text}(N, D_t) + (1 - \lambda) L_{vision}(N, D_v) + \Delta_{interaction}(N, D_t, D_v)$$

*Consequently, the effective scaling exponent $\alpha_{eff}$ satisfies:*

$$\alpha_{eff} = \frac{\partial \log L_{MLLM}}{\partial \log N} \in [\min(\alpha_{text}, \alpha_{vision}), \max(\alpha_{text}, \alpha_{vision})].$$

*Here, $\alpha_{text}$ and $\alpha_{vision}$ denote the scaling exponents for unimodal language and vision models, respectively. This implies that multimodal scaling is bounded by, and does not exceed, the unimodal exponents.*

*The interaction term arises from modality competition during training. When scaling exponents differ $(\alpha_{text} \neq \alpha_{vision})$, the optimizer must allocate gradient capacity asymmetrically. At smaller scales, language dominates (faster initial loss decrease); at larger scales, vision scales slower, creating interference. The* max *term captures this competition.*

The system exhibits anti-scaling: increased data in one modality can paradoxically worsen overall performance if modality imbalance is worsened. While modality balancing approaches, such as Liu et al. (2025a), propose a solution during preference tuning on a much smaller dataset, it is unclear how such an approach can be applied for pretraining at scale.

### 6.4.2 Data alignment noise

A critical phenomenon in large-scale multimodal datasets is the *non-linear compounding of the alignment noise.* Let $p_{\text{misalign}}$ be the fraction of misaligned image-text pairs. The effective noise in the fused pair scales as:

$$\epsilon_{\text{alignment}} \sim 1 - \prod_{i=1}^{|D|}(1 - p_{\text{misalign}})_i \approx 1 - (1 - p_{\text{misalign}})^{|D|}.$$

Even small misalignment fractions accumulate rapidly as $|D|$ grows. This leads to the degradation of cross-modal grounding (Shu et al., 2025). At small scales, high-quality pairs dilute noise, but at billion-scale datasets, misalignment dominates, causing the effective signal-to-noise ratio to decay roughly as

$$\text{SNR}(N) \propto \log N^{-1},$$

propagating into MLLM hallucinations (Wu et al., 2025d).

Larger models and datasets amplify misalignment and hallucinations, higher-resolution inputs raise computational cost quadratically, and linguistic priors continue to dominate reasoning. Multimodal brittleness is qualitative, arising from architectural constraints, information-theoretic bottlenecks, and misaligned objectives and scaling merely replicates these failures at higher fidelity.

## 6.5 Implications for robustness and deployment

The architectural, representational, and scaling failures converge on a conclusion that contemporary MLLMs are *synthetic correlators*, not grounded reasoners. As such, they are limited by representation mismatch, absence of causal reasoning, and biases inherited from web-scale datasets (Bommasani et al., 2022; Wu et al., 2025a). This leads to implications for trustworthiness and deployment risks, such as reliability, interpretability, and scalability limitations. *Reliability.* Cross-modal hallucinations and spurious grounding challenge model outputs in critical domains such as medicine, autonomous systems, or scientific reasoning (Hirano et al., 2024; Zhang et al., 2025e; Clusmann et al., 2025b; Cui et al., 2024). *Interpretability.* Token-level dominance and attention imbalances impede clear understanding of how models integrate visual, audio, or textual information (Wu et al., 2025a;c). *Scalability.* Computational cost of cross-attention and non-linear growth of alignment noise constrain practical deployment of large-scale MLLMs in resource-limited settings (Hoffmann et al., 2022b; Tay et al., 2022).

### 6.5.1 Promising directions

Addressing these limitations requires departures from current paradigms. **Neuro-symbolic grounding** integrates multimodal embeddings with explicit symbolic reasoning to enforce compositional structure and causal constraints, as in Cosmos (Sehgal et al., 2024) and NeSyGPT (Cunnington et al., 2024). **Real-world embodied data**, including sensorimotor interactions and robotics datasets (Mon-Williams et al., 2025; Chen et al., 2025a), provides perceptually grounded supervision that caption-mediated training cannot, as demonstrated by EO-1 (Qu et al., 2025a) and Gemini Robotics (Team et al., 2025). Both directions point toward the same conclusion: multimodal understanding requires embodiment and compositional structure, not merely richer data fusion.

## 7 Benchmark Limitations

Having established that LLM failures stem from five fundamental theoretical limits, namely hallucination, context compression, reasoning degradation, retrieval fragility, and multimodal misalignment, a natural question emerges; why do current benchmarks suggest continuous progress despite these intrinsic ceilings? This section demonstrates that contemporary evaluation practices systematically obscure these limits rather than measure them. Data contamination inflates scores by rewarding memorization over reasoning; judge bias incentivizes confident fabrication aligned with evaluator priors; compute-agnostic metrics hide the cost of marginal gains; and evaluation instability masks the saturation of genuine capability. Together, these artifacts

create an illusion of progress that conflates benchmark score increases with fundamental capability advances. Understanding evaluation limitations is thus essential to interpreting where scaling helps versus where it merely exploits measurement artifacts. Despite substantial progress in evaluation frameworks, benchmarking LLMs remains fundamentally constrained by issues of (i) data contamination, (ii) judge and protocol bias, (iii) compute and efficiency, (iv) stability, and (v) equity and safety.

**Data contamination.** As models are trained on increasingly large corpora of web-scraped text, the probability of benchmark test sets appearing in training data increases substantially (Apicella et al., 2025; Ni et al., 2025).

Let $\mathcal{D}_{\text{train}}$ denote the pre-training dataset, and $\mathcal{B}$ the benchmark dataset used for evaluation. Let $\mathcal{D}_{\text{train}}^{\text{info}}$ and $\mathcal{B}^{\text{info}}$ represent the sets of all informational content (text, semantics, or paraphrased variants) contained in $\mathcal{D}_{\text{train}}$ and $\mathcal{B}$, respectively. Benchmark Data Contamination (BDC) occurs when there exists overlap between $\mathcal{D}_{\text{train}}$ and $\mathcal{B}$, causing the model to have prior exposure to evaluation data or semantically related knowledge. This overlap can substantially bias evaluation outcomes. Xu et al. (2025a) quantifies the contamination risk as:

$$\text{BDC} = \frac{|\mathcal{D}_{\text{train}}^{\text{info}} \cap \mathcal{B}^{\text{info}}|}{|\mathcal{B}^{\text{info}}|}. \tag{52}$$

An empirical study by Lunardi et al. (2025) identified potential data contamination, especially in older benchmarks, indicating memorization rather than true understanding. This underscores the need for contamination-resistant benchmarks that are temporally updated and are semantically novel. The *TS-Guessing protocol* (Deng et al., 2024) and *PaCoST* statistical framework (Zhang et al., 2025b) enable contamination detection, though with inherent limitations in identifying sophisticated paraphrasing or semantic equivalence. Longitudinal analysis of contamination shows that performance drops after models' training cutoff dates provide temporal evidence of contamination (Roberts et al., 2024). Beyond detection, recent work has explored mitigation strategies, with mixed results regarding the effectiveness of decontamination approaches (Sun et al., 2025). More fundamentally, Chen et al. (2025b) introduced a dynamic data generation method to benchmark reasoning capabilities. Similarly, *LiveBench* (White et al., 2025) introduces a benchmark with automatically generated, frequently-updated questions and objective scoring, substantially reducing contamination risk through temporal dynamics.

**Judge and protocol bias.** Recent work highlights that benchmark outcomes for LLMs are heavily shaped by judge bias and protocol sensitivity, rather than genuine model capability. In LLM-as-a-judge evaluations, models used as evaluators often exhibit self-preference bias, favoring outputs generated in their own style or by the same model family (Wataoka et al., 2024). GPT-4, for instance, has been shown to rate its own responses higher than human judges do and to prefer text with low perplexity, that is, language it could have plausibly generated itself (Wataoka et al., 2024). Additional artifacts include position bias, where the order or labeling of options (A/B) alters preferences, and verbosity bias, where longer or list-formatted responses receive inflated scores even when quality does not improve (Wataoka et al., 2024). These effects were empirically observed in GPT-4-based benchmarks such as AlpacaEval, where longer answers and stylistic conformity correlated spuriously with higher ratings; subsequent length-controlled variants achieved far better alignment with human judgments (Dubois et al., 2024). Cross-model inconsistencies also emerge: different judges, or even prompt variants of the same judge, yield different rankings, a phenomenon visible in LMSYS Chatbot Arena, where Elo scores drift depending on whether comparisons are made by humans or AI (Dubois et al., 2024). Together, these biases illustrate "judge drift," in which over-reliance on a single LLM judge skews benchmarks toward its linguistic and stylistic priors.

Beyond judge effects, evaluation results are acutely protocol-sensitive. Minor variations in prompt wording, template formatting, or decoding parameters can swing model accuracy. A large-scale study tested 20 models on 39 tasks with 6.5 million prompt variants and found that performance rankings frequently reversed under semantically equivalent prompts (Mizrahi et al., 2024). Yet most papers still report single-prompt results without disclosing templates or decoding settings, impeding reproducibility and exaggerating apparent differences (Mizrahi et al., 2024). Even superficial changes, such as formatting the same input in plain text versus JSON, can shift accuracy by up to 40 percentage points (He et al., 2024). Such findings underscore that leaderboard scores are often artifacts of unreported evaluation choices, calling for multi-prompt evaluation

regimes and standardized documentation of prompts, context windows, and decoding parameters to mitigate protocol bias and restore comparability across studies.

**Compute and efficiency.** A growing body of work cautions that headline benchmark scores can reflect test-time compute inflation rather than genuine capability. Many recent evaluations allow models to consume vast computational resources via multi-sample reranking, extended chain-of-thought (CoT) reasoning, or external tool use, without normalizing for inference cost. As a result, higher accuracy often comes from spending more FLOPs, not from improved reasoning. For instance, complex reasoning strategies like multi-agent debate or Reflexion only surpassed simpler baselines when given larger compute budgets; under equal computation, a basic CoT + self-consistency baseline matched or outperformed them (Wang et al., 2024b). Similarly, a recent study found that prompting models to "think step-by-step" can slow inference by 35-600% (5-15 s vs. a few s) while yielding little or no accuracy benefit for stronger models (Meincke et al., 2025). These findings suggest that some CoT-augmented evaluations and retrieval-augmented setups may overstate ability: the same accuracy might be achievable with fewer samples or no external tools if the model were better calibrated.

Beyond compute inflation, public leaderboards often neglect efficiency, cost, and latency. Accuracy remains the dominant metric, while inference time, token usage, and energy cost, which are critical for real-world deployment, are rarely reported. The HELM (Liang et al., 2023) initiative explicitly treats efficiency as one of seven core metrics, measuring latency and token consumption, and finds top-scoring models like GPT-4 and Claude 3 Opus to be highly token-heavy and slow. Analyses of HELM results show smaller models (e.g., Mistral-7B, Cohere Command) deliver lower latency and higher throughput despite modestly lower accuracy, yet such trade-offs are invisible in typical leaderboards (Liang et al., 2023). This omission incentivizes "large-and-slow" models that dominate by brute force, even when less resource-intensive models may be more practical. Some initiatives, like EfficientQA (Chaybouti et al., 2025) competitions, now constrain model size or hardware budgets to promote cost-aware evaluation, but such standards remain rare. Moving toward multi-objective leaderboards that jointly report accuracy, inference time, and cost would better align benchmarking with deployment realities and reward models that are both capable and efficient.

**Stability.** A major limitation of current LLM benchmarks is the high variance and poor reproducibility of results. Small test sets, stochastic decoding, and prompt sensitivity can lead to wide performance fluctuations, sometimes large enough to reverse leaderboard rankings (Madaan et al., 2024). On math-reasoning benchmarks such as AIME, AMC, and MATH, merely changing the random seed can shift Pass@1 scores by 5-15 percentage points, with single-question differences moving aggregate results by 2-3 points (Hochlehnert et al., 2025). Sensitivity to prompt formatting and decoding parameters further compounds this fragility; minor prompt paraphrases or temperature changes can meaningfully alter benchmark outcomes (Sclar et al., 2024). Collectively, these results reveal that many leaderboard gains are statistically fragile, motivating multi-run or ensemble-of-seeds reporting as standard practice (Blackwell et al., 2024).

Benchmark instability also stems from model version drift. API-based systems such as GPT-4 or Claude evolve through unannounced updates, causing identical evaluations at different times to yield inconsistent results. documents significant score shifts across months and introduces contamination-resistant and resampling methods to stabilize longitudinal rankings (Zhang et al., 2025c). Even for static open-source models, minor hardware or library differences can introduce nondeterminism in parallel computation, underscoring the need for version tracking and controlled evaluation environments. Evidence from a recent report shows that single-run accuracy can be misleading: when each question was answered 25 times, outcome distributions revealed large intra-model variability, prompting metrics such as "$\geq 51\%$ correct" instead of one-off accuracies (Meincke et al., 2025).

**Equity and safety.** Benchmark equity and safety robustness remain among under under-evaluated areas of LLMs. Despite the broad adoption of multilingual and domain-specific benchmarks, many were created in English or other high-resource languages and later translated, introducing artifacts that distort difficulty and cultural meaning. Studies find that translation-based evaluation can over- or underestimate capability due to phrasing differences, ambiguity, or loss of nuance. Accuracy disparities across languages strongly correlate with training data abundance and translation quality rather than intrinsic reasoning gaps (Gupta et al., 2025). Similarly, MM-Eval demonstrates that mere translations of benchmarks like MMLU or GSM8K yield

inconsistent results across linguistic variants because prompts differ in fluency and familiarity (Son et al., 2025). These issues reflect broader concerns about Western-centric benchmark design as translated content often fails to capture local references or idioms (Pawar et al., 2025; Talat et al., 2022). BenchMAX and related frameworks propose using native-authored test sets and language-specific calibration to mitigate such bias (Huang et al., 2025b). Furthermore, domain-specific benchmarks (e.g., PubMedQA, BigBench-Legal) are often English-only, leading to inflated generalization claims. The literature thus calls for culture-grounded benchmarks and transparent documentation of supported languages (Singh et al., 2024; Wu et al., 2025b).

Safety evaluations remain brittle. Many rely on static "red-team" prompt lists, allowing models to overfit to known attacks (Perez et al., 2022). Studies show that even RLHF- or constitutionally aligned models remain vulnerable to paraphrased or multi-turn jailbreaks. A recent study finds that simple perturbations like synonym swaps, role-play framing, or code-switching can bypass filters with high success rates (Xu et al., 2024d). Further, JailbreakBench (Chao et al., 2024) demonstrates that defenses effective in one release can fail in the next. Moreover, parameter changes can degrade refusal behavior without affecting the task performance (Wei et al., 2024), confirming that alignment is not structurally robust. Finally, a multi-turn study on LLM defenses to jailbreaking reveals that conversational persistence erodes safety consistency (Li et al., 2024a).

## 8 Discussion and Future Work

Existing surveys on LLM reliability and scaling catalog empirical pathologies such as hallucination, reasoning failure, or retrieval brittleness, yet stop short of unifying them under a formal theoretical account. Our work closes this gap by establishing a rigorous, proof-based framework that connects these recurring failures to fundamental theoretical ceilings. By integrating computability, information theory, and statistical learning, we demonstrate that certain classes of errors are not removable artifacts of training or architecture—but are necessary consequences of computation itself. This synthesis reframes LLM research: the challenge is not to eliminate failure, but to understand, bound, and allocate it optimally.

**Interpretation and implications.** Our results formalize necessary limits on LLM capability under scaling: diagonalization and uncomputability guarantee irreducible error sets; information-theoretic and sample-complexity bounds impose compression and generalization constraints; and long-context dynamics, like positional undertraining, encoding attenuation, and softmax crowding compress the effective context window far below its nominal size. These ceilings are architecture-agnostic in principle but operationally shaped by training distributions, inference policies, and system design. Engineering innovations (better curricula, retrieval, routing, or memory) can help approach these ceilings, but cannot surpass them. Hence, the future of scalable, reliable AI lies not in chasing asymptotic perfection but in designing systems that fail gracefully, predictably, and transparently.

Because some error is intrinsic, systems should optimize where and how failure occurs. This includes calibrated abstention over confident fabrication, task-aware decoding that modulates entropy when factual accuracy is critical, and retrieval used as bounded oracle access under finite token budgets. Separating creative from factual modes, or introducing confidence-aware grading and contamination-resistant benchmarks, can realign incentives that otherwise reward confident guessing.

**Methodological scope.** Our theoretical arguments necessarily rely on stylized assumptions like independence in long-tail distributions, simplified distractor models for softmax competition, and idealized capacity measures to maintain analytical tractability. While these abstractions clarify mechanisms, deriving tight, domain-specific constants remains an open challenge. Furthermore, we survey representative mitigation avenues, not an exhaustive catalogue; integrating these ideas into production-scale architectures requires both formal and empirical co-design.

**Future work.** We identify five promising fronts for extending this framework:

1. **Quantifying intrinsic limits:** Move from existence proofs to measurable lower bounds on irreducible failure rates under realistic query distributions, enabling models to report *how often* failure is inevitable.

2. **Reliable systems and evaluation:** Develop selective-prediction pipelines with calibrated abstention and coverage guarantees (e.g., conformal prediction), enabling selective uncertainty control and linguistic calibration so models abstain outside reliable regions and produce appropriately hedged answers (Jiang et al., 2025; Wang et al., 2025c).

3. **Long-context and memory:** Design objectives and curricula that maintain logarithmic-in-context attention margins, and combine them with recurrent, SSM, or external-memory architectures guaranteeing efficient long-range information transport.

4. **Retrieval under token budgets:** Formalize retrieval-augmented generation as a constrained optimization problem, deriving approximation guarantees for multi-hop coverage and robustness under noisy or adversarial retrieval.

5. **Reasoning and multimodality objectives:** Go beyond likelihood by enforcing process consistency (intermediate checks, constraint satisfaction) with explicit sample–compute trade-offs; add controllable creativity–factuality mode switches with regret bounds; and rebalance multimodal gradients via information-bottleneck regularization to reduce text dominance without harming language fluency.

By grounding empirical scaling laws in formal impossibility results, this paper provides a principled basis for future progress in large-scale modeling. Understanding that these limitations are inherent, not incidental invites a paradigm shift from asking how to make models infallible to asking how to make their fallibility quantifiable, predictable, and aligned with task goals.

## 9 Conclusions

This work identifies and formalizes five fundamental limitations of LLMs that persist even under scaling: hallucination, context compression, reasoning degradation, retrieval fragility, and multimodal misalignment. Drawing from computability theory, information theory, and statistical learning, we establish that these are not merely empirical artifacts but principled limits inherent to the design and deployment of LLMs. We present theoretical impossibility results, bounded performance theorems, and capacity-aware diagnostics to characterize where scaling helps, where it saturates, and where it fails. Existing surveys and empirical analyses of LLM behavior document recurring failure cases without connecting them to the formal limits of computation or learnability. By contrast, our framework unifies these observations under a proof-based theoretical foundation, showing that many observed pathologies are not accidental engineering outcomes but inevitable consequences of computability, finite information, and sample constraints. This connection between theory and practice provides a rigorous synthesis of scaling limits across architectures, modalities, and objectives. Our synthesis suggests that the future of reliable language modeling lies not in unbounded scaling, but in architecture-aware, theoretically grounded design guided by an understanding of what models cannot possibly learn, what data cannot provide, and where system-level interventions must operate.

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
