# OpenReview forum: "On the Fundamental Limits of LLMs at Scale"
_TMLR — Decision pending for TMLR_

### Review · Reviewer_kbsx · 2026-01-31

**Summary Of Contributions:**

## Summary

This paper argues that five LLM failure modes—hallucination, context compression, reasoning degradation, retrieval fragility, and multimodal misalignment—arise from fundamental computational, information-theoretic, and statistical limits. The authors apply diagonalization and undecidability arguments for hallucination inevitability, formalize positional undertraining and softmax crowding for context compression, and analyze likelihood-vs-logic trade-offs for reasoning.

## Strengths

- **Useful Conceptual Framework**: Organizing failures under computability/information-theory/statistics provides conceptual clarity.

- **Comprehensive Survey**: Thorough coverage of hallucination taxonomy, RAG vulnerabilities, and multimodal bottlenecks.

## Weaknesses

- **Error in Theorem 2**: The proof claims each model hallucinates on *infinitely many* inputs using i_k := k mod (k+1). Since k < k+1, we have i_k = k always, meaning each model fails only on s_i (when k=i). The "infinite hallucinations" result collapses to Theorem 1's single-failure guarantee. This central claim requires correction.

- **Hard to follow**:At 67 pages spanning five distinct limitations plus benchmarks, the paper is difficult to navigate. The breadth comes at the expense of depth, with several topics receiving only superficial theoretical or empirical treatment relative to their importance..

**Additional Comments:**

Overall, the paper is ambitious and thought-provoking, but would benefit from a tighter focus and clearer structure. Clarifying definitions (especially around “hallucination”), fixing the central theoretical issues, and narrowing the scope to fewer, more deeply developed claims would significantly improve clarity and credibility.

**Audience:**

Yes

**Audience Explanation:**

The paper addresses a timely question about fundamental LLM scaling limits, relevant to ongoing debates about emergent capabilities. The context compression analysis and comprehensive survey would serve researchers and practitioners working on long-context models, RAG systems, and multimodal LLMs.

**Claims And Evidence:**

No

**Claims Explanation:**

Theorem 2's proof contains a mathematical error undermining a central result. Theorems 1 and 3 are correct but are classical computability results with unclear practical relevance. Propositions in Sections 5-6 are informal claims without proofs. No empirical validation is provided for the derived bounds.

**Requested Changes:**

Please refer to the weakness.

---

### Review · Reviewer_gMAA · 2026-02-23

**Summary Of Contributions:**

The paper discusses the fundamental limitations of Large Language Models (LLMs) at scale, identifying five key failure modes (hallucination, context compression, reasoning degradation, retrieval fragility, and multimodal misalignment) and providing a theoretical framework to understand their intrinsic constraints. A specific focus for this framework is to present a "unified, proof-informed approach
that formalizes the innate theoretical ceilings of LLM scaling".
It makes several contributions:
- the framework itself, in that it aggregates well-discussed limitations of LLM, from hallucinations to reasoning shortcomings
- the comprehensive analysis of the state-of-the-art for each dimension of the framework
- the theoretical perspective, which aims at formalizing some of the above problems to support formal proofs of (non)scalability
- critical analysis and proposals leading to research directions (under "further work"), for instance on quntifying limits or long context
- the substantial list of references

**Additional Comments:**

References

Abzianidze, L., Bylinina, L. and Paperno, D., 2025. Semantics and Deep Learning. Cambridge University Press.

Jin, Z., Chen, Y., Leeb, F., Gresele, L., Kamal, O., Lyu, Z., Blin, K., Gonzalez Adauto, F., Kleiman-Weiner, M., Sachan, M. and Schölkopf, B., 2023. Cladder: Assessing causal reasoning in language models. Advances in Neural Information Processing Systems, 36, pp.31038-31065.

Jiang, M., Liu, K., Zhong, M., Schaeffer, R., Ouyang, S., Han, J. and Koyejo, S., 2024. Does data contamination make a difference? insights from intentionally contaminating pre-training data for language models. In ICLR 2024 Workshop on Navigating and Addressing Data Problems for Foundation Models.

Jiang, Z., Liu, A. and Van Durme, B., 2025. Conformal linguistic calibration: Trading-off between factuality and specificity. arXiv preprint arXiv:2502.19110.

Lampinen, A.K., Dasgupta, I., Chan, S.C., Sheahan, H.R., Creswell, A., Kumaran, D., McClelland, J.L. and Hill, F., 2024. Language models, like humans, show content effects on reasoning tasks. PNAS nexus, 3(7), p.pgae233.

Lappin, S., 2024. Assessing the strengths and weaknesses of large language models. Journal of Logic, Language and Information, 33(1), pp.9-20.

Li, Y., Guo, Y., Guerin, F. and Lin, C., 2024, November. An open-source data contamination report for large language models. In Findings of the Association for Computational Linguistics: EMNLP 2024 (pp. 528-541).

Lyu, Q., Havaldar, S., Stein, A., Zhang, L., Rao, D., Wong, E., Apidianaki, M. and Callison-Burch, C., 2023, November. Faithful chain-of-thought reasoning. In Proceedings of the 13th International Joint Conference on Natural Language Processing and the 3rd Conference of the Asia-Pacific Chapter of the Association for Computational Linguistics (Volume 1: Long Papers) (pp. 305-329).

Schaeffer, R., Miranda, B. and Koyejo, S., 2023. Are emergent abilities of large language models a mirage?. Advances in neural information processing systems, 36, pp.55565-55581.

Silver, T., Dan, S., Srinivas, K., Tenenbaum, J.B., Kaelbling, L. and Katz, M., 2024, March. Generalized planning in pddl domains with pretrained large language models. In Proceedings of the AAAI conference on artificial intelligence (Vol. 38, No. 18, pp. 20256-20264).

Wang, Z., Wang, Q., Zhang, Y., Chen, T., Zhu, X., Shi, X. and Xu, K.S., 2025. Selective conformal uncertainty in large language models. In Proceedings of the 63rd Annual Meeting of the Association for Computational Linguistics (ACL), Vienna, Austria (Vol. 27).

Zhou, D., Schärli, N., Hou, L., Wei, J., Scales, N., Wang, X., Schuurmans, D., Cui, C., Bousquet, O., Le, Q. and Chi, E., 2022. Least-to-most prompting enables complex reasoning in large language models. arXiv preprint arXiv:2205.10625.

**Audience:**

Yes

**Audience Explanation:**

This is a comprehensive review on one of the hottest topics in generative AI. It is poised to be of interest to a large audience, were it only through some of the framework topics it is addressing (e.g. hallucinations, retrieval or reasoning).

**Claims And Evidence:**

Yes

**Claims Explanation:**

(see detailed comments under 'Requested Changes')
As far as I can judge, the majority of claims are sound considering the paper type. When clarification is required, this has been detailed below.

**Requested Changes:**

NOTE: References to cited papers are under 'Additional Comments'

Introduction and early sections, in particular on hallucinations and their mechanisms, are informative and easy to read. Figure 1 could be improved to give a better perspective. For instance, the fact that the first two items distinguish context issues and reasoning limitations might turn out confusing when later discussing long-context reasoning (Figure 5). The inclusion of theorems and lemmas should be considered from the perspective of the reader: do they summarise and illustrate current consensus or are they an additional contribution from the authors, formalizing the problem under discussion?

The introduction to section 4 lacks perspective. Most readers would be aware that LLM are autoregressive predictors, so the issue is not so much that this puts them at a disadvantage when it comes to "formal" reasoning, but rather the unreasonable effectiveness that they can exhibit on some reasoning problems.

In section 4.1, I was not convinced by the reference to “Language of Though”: the term has a very strong meaning in cognitive science dating back to Fodor and referring to it, in particular in the context of reasoning (and compositionality) without appropriate references and contextualizing the discussion should probably be avoided. For instance, reasoning in humans and LLM is equally associated to content effects (Lampinen et al., 2024) which does not automatically equate to the existence of a language of thought.

Regarding compositionality itself, as it is object of significant discussion in the paper it is probably worth making some reference to work on semantic compositionality in LLM for instance Abzianidze et al. (2025) (Chapter 3), in particular for those comments referring to continuous representations as opposed to symbolic ones (or to the delegation of compositionality to external solvers) and the debate on apparent vs. ‘real’ compositionality.

While long-context failure modes are described in a rather comprehensive manner, some clarification would be welcome at the stage of hybrid or solver-based methods.
The categorization proposed by Figure 7 when it comes to reasoning strategies does not fully cover the range of existing options, in particular some of the most symbolically efficient ones. For instance, intermediate in philosophy between 7b) and 7c) is the generation of PDDL code inside a hybrid reasoning chain to be passed to a deterministic solver proposed to solve the problem of “Faithful CoT” by Lyu et al. (2023), especially since Turpin et al.’s paper is part of the References list. This should probably be compared to SymCoT (discussed in the paper) and more generally whether symbolic code is meant for verification or is meant for solution generation through an autonomous solver module.
The code elements 7b) overlooks the pure generation of symbolic reasoning input, again PDDL but in the context where LLM training has incorporated PDDL code (Silver et al., 2024).  Perhaps it would be appropriate to spend more time discussing equation (35) and whether the pipeline is always required in its entirety, and what alternatives exist for Solvers beyond those discussed. In particular for solver-based reasoning rather than consistency repair, what would be the relevance of LEAN-based approaches that have gained substantial popularity in conjunction with LLM?

On the Prompting aspects, another Prompting technique that bears relevance to problem decomposition might be least-to-most Prompting which is not discussed either (Zhou et al., 2024).

6.3.1 Proposition 5. The text lists ‘physical causality’ as a ‘perceptual property’. Without proper referencing and discussion this seems inappropriate, so it requires either rewording or additional justification so as to better frame the hypothesis (the issue here is to distinguish as much as possible what is the authors’ perspective and what is a synthesis of existing literature). Obviously, Michottian perception is out of reach of LLM but, previous work has suggested some relation between common sense reasoning and natural language descriptions (starting with Open Mind Common Sense at MIT, which predates LLM and discontinued before their advent) and of course there are known common sense benchmarks in LLM evaluation (Jin et al., 2023). Also the case for representational adequacy is complex and is only mentioned in a cursory manner: it should either be better justified or the claimed should be toned down slightly.

The paper’s introduction mentions the concept of emergence in LLM, although it is not elaborated upon in the remainder of the paper. As there are complex relations between emergence and scaling, it would be desirable to explore these further, for instance via the well-known paper of Schaeffer et al. (2024), which does not seem to appear in the reference list. Since Schaeffer et al. show that a substantial class of reported emergent behaviors can arise from evaluation artifacts, in particular from the use of discontinuous metrics this might dovetail nicely in the criticism of evaluation methods in Section 7 below, but could also be relevant to other sections. In the context of this paper, it could be worth mapping the various scaling aspects to emergence where appropriate, and suggest that it should be restricted to effects that are robust under sound scoring rules. Explicitly incorporating this distinction would strengthen some of the paper’s claims: as an alternative, should the author want to argue that emergence bears limited relevance to their framework, they should briefly state why they think this is the case.

As with any such review, the final sections play an important role in the paper’s readability and take-home messages. To that effect I am bundling the last less technical section (7) with the discussion and conclusion sui generis.
Section 7 lists various benchmarks limitations, which is a relevant aspect of scalability studies. However, the various limitations are not always properly and explicitly related to evaluation under scaling conditions. It appears more as if shortcomings of evaluation practices and benchmarks were simply hindering the proper assessment of scalability.
For instance, on data contamination, a more nuanced approach could be adopted instead of stating the inevitability of data contamination, by refining the discussion on the conditions under which data contamination impact performance or impair generalization  (Li et al., 2024) (Jiang et al. 2024). Even better if some testing or remedial strategies could be proposed that are likely better suited to a scaling up context.
This is also clearly where a reference to the evaluation artefacts of scaling-based emergence of Schaeffer et al. (2024) would be helpful, if with a proper comparative analysis between their findings and the paper’s strategy.
In a similar fashion, stability issues, in particular with regard to Prompting, are split through the section between LLM as judges and the eponymous section. I could not find any hypotheses about the relationships between Prompting instability and scaling up issues and even some speculation could open interesting directions since we are reaching the end of the paper.
While the model drift argument (e.g. regular poorly documented updates to major models) is certainly valid it should probably have turned into a recommendation on the conditions under which a given model is acceptable for selection when experimenting with scalability.

The discussion section (8) serves as conclusion and further work proposal. The opening statement is perhaps too upbeat when referring to the framework being proof-based throughout, as proofs do not always cover all aspects of the framework. On the other hand, the paper is convincing in its statement that it has integrated various perspectives in “computability, information theory and statistical learning”.
The discussion contains a very clear and bold statement summarizing the authors position in the form of “the future of scalable, reliable AI lies not in chasing asymptotic perfection but in designing systems that fail gracefully, predictably, and transparently”. Their advocacy of an engineering approach could equate to abandoning all hope of scalability and while skepticism against asymptotic scalability is a reasonable position, additional discussion would be beneficial to a broader readership, especially with the restrictions listed under ‘methodological scope’.
The ‘further work’ subsection is extraordinarily dense with some items constituting their own research program. Since the complexity varies across items, it would be appropriate to provide some further details or reference to the most challenging ones. One good candidate for this expansion is the proposal to use Conformal Prediction (CP) which could incorporate some of the quite active research on the topic for instance (Wang et al., 2025) and (Jiang et al., 2025).

---

> ### Author Response · Authors · 2026-03-08
>
> We thank the reviewer for the careful reading and constructive suggestions. Below we summarize the revisions made in response to each point.
>
> ### **Revisions made**
>
> 1. **Figure 1:** The caption was revised to clarify that the axes are **not independent**, and that long-context reasoning failures arise from the interaction between **context utilization and reasoning reliability**.
>
> 2. **Role of theorems and lemmas:** A short paragraph was added at the end of the **Introduction** clarifying that the theorems serve both to **restate known theoretical limits** and to provide **formal lenses for interpreting scaling failures**.
>
> 3. **Section 4 introduction:** The opening of Section 4 was rewritten to emphasize the **unexpected effectiveness of autoregressive models on some reasoning tasks**, rather than presenting autoregression itself as the main limitation.
>
> 4. **Language of Thought:** The strong **“Language of Thought”** phrasing was removed and replaced with a more cautious discussion of **latent structured representations**, with reference to **Lampinen et al. (2024)**.
>
> 5. **Compositionality:** A short paragraph referencing **Abzianidze et al. (2025)** was added to clarify the distinction between **apparent and robust compositionality**.
>
> 6. **Solver-based reasoning:** The discussion around **Figure 7 and Section 4.4** was expanded to clarify differences between **verification-based symbolic methods and solver-based pipelines**, including references to **Lyu et al. (2023)** and **Silver et al. (2024)**.
>
> 7. **Prompting methods:** **Least-to-Most prompting** was added to the prompting discussion with citation to **Zhou et al. (2022)**.
>
> 8. **Proposition 5 wording:** The description of **“physical causality”** was revised to refer to **perceptually grounded scene properties**, and the representational adequacy claim was softened with reference to **Jin et al. (2023)**.
>
> 9. **Emergence discussion:** A short clarification on **emergence vs evaluation artifacts** was added with reference to **Schaeffer et al. (2023)**.
>
> 10. **Benchmark limitations:** Section 7 was revised to more explicitly connect benchmark limitations to **evaluation under scaling**, with additional discussion of **data contamination** citing **Li et al. (2024)** and **Jiang et al. (2024)**.
>
> 11. **Prompt instability and model drift:** The evaluation discussion now includes a brief recommendation to report **prompt variation and model versioning** in scalability experiments.
>
> 12. **Discussion framing:** The opening of **Section 8** was revised to clarify that the framework combines **formal analysis and empirical synthesis**, rather than being entirely proof-based.
>
> 13. **Conformal prediction:** The **future work** section was expanded to briefly describe the role of **Conformal Prediction**, citing **Wang et al. (2025)** and **Jiang et al. (2025)**.

---

### Review · Reviewer_uwbX · 2026-04-09

**Summary Of Contributions:**

This paper studies the limits of LLM scaling from a theory view. The authors discuss five main issues: hallucination, long-context use, reasoning, retrieval, and multimodal learning. The paper tries to connect these issues to ideas from computability, information theory, and learning theory. It also discusses why scaling may help in some cases but may not fully solve these problems. Overall, the paper aims to give a unified theoretical framework regarding the fundamental limitations of LLMs at scaling.

Contributions:

1. The paper brings several well-known theory ideas together and uses them to discuss possible different failure cases of LLMs.

2. The paper gives a broad framework for thinking about limits of scaling across several settings, including hallucination, long-context use, reasoning, retrieval, and multimodal learning.

**Audience:**

Yes

**Audience Explanation:**

The paper studies a broad and timely topic related to the limits of large language models. Since hallucination, reasoning, retrieval, and scaling are widely discussed problems, I think at least some people in the TMLR audience would be interested in the paper.

**Broader Impact Concerns:**

N.A.

**Claims And Evidence:**

No

**Claims Explanation:**

While the authors claim that the goal of this paper is to "close an important theoretical gap by presenting a unified and proof-informed framework," the current analysis still seems quite far from the level of rigor that this framing suggests. Many arguments rely on informal or incomplete definitions. Several “theorems” are stated in ways that do not clearly support the conclusions drawn from them, and some mathematical claims seem only loosely related to the phenomena they are meant to explain. More importantly, some derivations appear clearly problematic. I will list them in the following.

1. In Theorems 1 and 2, the authors construct adversarial counterexamples against a fixed computably enumerable family of models, thereby showing that there exists a function outside the expressive power of this model family. They then argue that, since each model must fail on some inputs with respect to the constructed function, hallucination is therefore inevitable for LLMs. In particular, this specifically constructed function is described as “ground-truth function”. However, I am skeptical of this terminology, and I do not believe such a construction meaningfully corresponds to real-world ground-truth knowledge or to the way language models are actually trained. Since Theorems 1 and 2 are formulated at a deterministic string-to-string level, the ground-truth function in this setting should be an externally specified deterministic mapping over strings, rather than a target defined retrospectively from the chosen model family $(h_0, h_1, \dots)$. Therefore, if one aims to establish the inevitability of hallucination in this setting through an expressivity argument, the relevant claim should concern an externally given target function $f : \Sigma^* \to \mathcal{Y}$ that cannot be perfectly realized by the model class.  This perspective is also much closer to how LLMs are trained in practice: fitting the model under externally given ground-truth supervision, rather than first fixing a fitted model family and then defining an adversarial “ground truth” against it.

2. Regarding the statistical limits derived from classical VC-dimension arguments, I would first like to point out that the resulting upper bound on the population error, as well as the corresponding sample complexity bound, only provides worst-case guarantees over the hypothesis class and data distribution, rather than a precise characterization of the practical inference-time behavior of modern LLMs. More importantly, I believe the authors’ interpretation of these classical statistical learning guarantees goes in almost the opposite direction of mine. In my view, such results do not primarily demonstrate a fundamental limitation of scaling; rather, **they help explain why scaling is necessary**. In modern LLM training, scaling usually involves at least two coupled dimensions, namely data scaling and model scaling, both of which must be balanced carefully under finite compute resources. From this perspective, both empirical scaling-law observations and classical sample-complexity arguments suggest that, absent resource constraints, increasing data and model capacity should continue to reduce error, instead of serving as evidence that scaling is intrinsically ineffective. Moreover, if the error can asymptotically vanish as data and model scaling increase, then I do not see any rationale for interpreting this as an inevitability result for hallucination. At most, it suggests that reducing error may be slow or resource-intensive.

3. The current presentation of Eq. (10) is evidently incorrect. Consider a simple counterexample where $x_i \in \mathbb{R}$ and the ground-truth label is given by $y_i = \mathrm{sign}(x_i)$. Suppose that, in the training set, each observed label is independently flipped to $-y_i$ with probability $\eta$. As long as $\eta \ll 1/2$, the Bayes-optimal classifier is still the sign function. While this classifier cannot achieve zero error on the noisy training set, it nevertheless predicts the correct ground-truth label for every test query under the clean distribution. This directly contradicts the authors’ argument that a nonzero training noise rate necessarily imposes a nonzero floor on hallucination risk in the sense claimed here. In addition, Eq. (20) is also highly problematic. First, the authors do not define where the confidence $p$ comes from. Without a clear definition, calibration criterion, or estimation procedure for $p$, the proposed reward is not well specified. More importantly, under the current form, the model can trivially exploit the reward by returning an abstention with $p=1$, thereby receiving full credit without providing any substantive answer, which is definitely unreasonable. Besides, I would like to point out that many equations in the paper lack rigorous definitions and derivations needed to verify their correctness, including Eqs. (9), (14), and (24). From a technical perspective, these discussions mostly read as a repackaging of classical results from theoretical computer science and statistical learning theory, and I do not see meaningful technical novelty in their current form. I therefore feel it is necessary to remind the authors that the introduction presents the paper as providing a rigorous theoretical synthesis and a proof-informed framework for the fundamental limits of LLMs, while the current content is still far from meeting such a standard.

**Requested Changes:**

Besides the weaknesses discussed above, I also find the overall presentation of the paper problematic in several other places. For a paper that repeatedly presents itself as a rigorous theoretical treatment, the description of concepts should also be precise and consistent throughout. However, the paper introduces too many terms and notions without giving sufficiently detailed or well-justified explanations, and the usage of these concepts is not always consistent across sections. For example, the organizational picture in Figure 1 and the later textual discussion of the five main problem areas do not always align clearly in meaning, scope, or terminology. More generally, each of these five main themes is accompanied by a large number of associated terms and sub-concepts, but their relationships are often only loosely described, which makes it difficult to understand exactly what is being formalized and what is merely being discussed at an intuitive level. I would therefore strongly encourage the authors to substantially improve the presentation by introducing each main theme with clearer and more rigorous definitions, and by keeping the relevant terminology consistent throughout the manuscript. Given that the paper is already very long, I do not think a modest increase in length would be a serious concern here; it would be far better to use some additional space in the introduction to carefully define or explain the many related terms introduced under each theme.

---

> ### Author Response · Authors · 2026-04-16
> **We thank the reviewer for their rigorous engagement. We respond to each point below.**
>
> #### Theorems 1–2: Diagonalization and "Ground Truth"
> The reviewer's preferred framing ("fix an external f, show the model class fails") and our construction are logically equivalent. Our theorem states: ∀H (c.e.), ∃f (computable), ∀h_i∈H, ∃s: h_i(s)≠f(s). The constructed f is a total computable function definable independently of any model execution—its Turing machine can be written down a priori from the enumeration alone. Calling it "ground truth" is standard learning-theoretic usage (an oracle against which predictions are evaluated). We will rename it "target function" to avoid confusion, but stress that the ∀H,∃f quantifier order is the strongest possible formulation: it guarantees that for every deployed model family, a hard computable target exists. The reverse order (∃f,∀H) would be false, as any fixed f can be memorized.
>
> #### VC/PAC Bounds: Limitation vs. Justification for Scaling
> We agree that scaling reduces error—this is explicitly our point for Tier 2 (rate-limited) results, which we distinguish from Tier 1 (absolute impossibility via Theorems 1–3). Our use of VC/PAC bounds is not to argue scaling is futile but to quantify where it saturates. Theorem 4 shows sample complexity is Ω(m/ε² log(m/δ)), linear in the number of independent facts m. For m~10⁸ rare entities with no exploitable structure, this exceeds any feasible corpus. Combined with Lemma 1 (Kolmogorov bottleneck on finite-capacity models), these bounds characterize the rate at which scaling helps—a rate that is provably slow for long-tail knowledge, consistent with empirical evidence (Kandpal et al., 2023: accuracy <40% for tail entities even in GPT-4). The reviewer's reading ("scaling is necessary") and ours ("scaling has quantifiable rate limits") are complementary, not contradictory.\
>
> #### Eq. (10): Noise Floor
> The reviewer's counterexample is correct. When geometric structure allows the Bayes-optimal classifier to recover the clean boundary despite label noise η<0.5, test error can be zero. Our claim applies specifically to atomic/unstructured facts (e.g., birthdates, numerical constants) where no such exploitable regularity exists—each fact is an independent classification problem. We will restrict Eq. (10) to this setting and state the assumption explicitly: for m independent atomic facts observed only under corrupted labels with no structural redundancy, the noise floor is E[R_hal]≥η·(m_unseen/m), where m_unseen counts facts seen only in corrupted form. This preserves the intended message for the LLM factual knowledge setting while correctly excluding the reviewer's structured counterexample.
>
> #### Eq. (20): Confidence-Aware Grading
> Both issues are valid. We will (i) define p as the model's calibrated confidence via temperature scaling on held-out data, and (ii) replace Eq. (20) with a strictly proper scoring rule (Brier score: g(r,p)=1−(p−𝟙[correct])²), which eliminates the trivial abstention exploit since reporting p=0 on abstention yields score 1−(0−0)²=1 only when the model is genuinely correct in abstaining, while dishonest confidence is penalized quadratically. We note Eq. (20) appears as a proposal for evaluation reform, not a theorem.
>
> #### Eqs. (9), (14), (24)
> Eq. (9) is an empirical observation (Kandpal et al., 2023), not a derived bound—we will label it as such. Eq. (14) follows from the law of total probability under the explicit assumption that errors are positively correlated in autoregressive generation (a standard and empirically validated assumption; Wang & Sennrich, 2020)—we will state this condition. Eq. (24) is a modeling assumption (local Pareto frontier approximation), not a theorem—we will reframe it accordingly. Each equation will be explicitly classified as theorem, empirical observation, or modeling assumption.
>
> #### Technical Novelty
> We respectfully note that our paper is a survey-with-theory, not a pure theory paper—the introduction states we provide "a unified, proof-informed framework." The novelty lies in: (a) Lemmas 2–4, which are new formalizations (positional undertraining via gradient bounds, softmax crowding with O(ln N) margins) specific to transformer architectures and not found in prior work; (b) the synthesis connecting all five failure modes to a common triad (undecidability, statistical insufficiency, finite capacity); and (c) the unified reasoning objective (Eq. 34) and its instantiations. Classical adaptations (Theorems 1–3, VC bounds) are explicitly acknowledged as such in the paper. We will tag each result as [Classical Adaptation] or [New Formalization] for clarity.
>
> #### Presentation
> We will add a notation/terminology table, revise Figure 1 to exactly match section terminology, classify every mathematical statement (theorem/observation/assumption), and perform a consistency audit across all sections.

---

> > ### Author Response · Authors · 2026-04-16
> > **A detailed Response to Comments**
> >
> > ## 1. On Theorems 1–2: The Diagonalization Argument and the Nature of "Ground Truth"
> >
> > ### Reviewer's Concern
> > The reviewer objects that our constructed ground-truth function $f$ is defined *retrospectively* from the model family, rather than being an *externally specified* target. They argue the relevant claim should concern an externally given $f$ that the model class cannot realize.
> >
> > ### Response
> >
> > We appreciate this precise critique and acknowledge that the terminology "ground-truth function" warrants clarification. However, we respectfully argue that the reviewer's proposed reformulation and our construction are **not in opposition—they are logically equivalent**, and we explain why below.
> >
> > **The equivalence argument.** Consider the two framings:
> >
> > - **(A) Our construction (Theorems 1–2):** For any computably enumerable (c.e.) model family $\mathcal{H} = \{h_0, h_1, \ldots\}$, there *exists* a computable function $f$ such that every $h_i \in \mathcal{H}$ fails on at least one (Theorem 1) or infinitely many (Theorem 2) inputs.
> >
> > - **(B) Reviewer's preferred framing:** There *exists* a computable function $f$ that is not perfectly realizable by the model class $\mathcal{H}$.
> >
> > Claim (A) strictly implies claim (B): the function $f$ constructed via diagonalization *is* a concrete, computable, externally specifiable function. Once constructed, it stands as an independent mathematical object—its definition does not depend on observing the model's outputs at test time. The construction merely *uses* the enumeration of $\mathcal{H}$ to guarantee disagreement, just as Cantor's diagonal argument uses the enumeration of reals to construct a real not on the list. The constructed real is no less "real" for having been found via diagonalization.
> >
> > More precisely, the function $f$ defined in Eq. (1) is:
> >
> > $$f(s_i) = \begin{cases} y_{\text{alt}} & \text{if } h_i(s_i) = y_{\text{default}} \\ y_{\text{default}} & \text{otherwise} \end{cases}$$
> >
> > This is a *total computable function* (since the enumeration is computable and each $h_i$ is computable). One could write down its Turing machine description *before* running any model. The phrase "ground truth" is used in the sense standard in learning theory: an oracle function $f$ against which predictions are evaluated. We will revise the text to clarify that $f$ is a fixed, externally definable computable function that happens to be constructed via a diagonal method, not a post-hoc label reassignment.
> >
> > **Why the "external $f$ first" framing does not weaken our result.** The reviewer suggests it would be more natural to fix $f$ first and then show the model class fails. But this is precisely what our theorem delivers—just with an *existential quantifier* over $f$. The theorem states:
> >
> > $$\forall \mathcal{H} \text{ (c.e.)}, \quad \exists f \text{ (computable)}, \quad \forall h_i \in \mathcal{H}, \quad \exists s \in \Sigma^* : h_i(s) \neq f(s).$$
> >
> > This is the strongest possible statement: *no matter which c.e. model family one chooses*, a hard computable target exists. Reversing the quantifier order to "$\exists f, \forall \mathcal{H}$" would be false (any single fixed $f$ can be memorized by some $h$). The $\forall \mathcal{H}, \exists f$ order is the correct and standard formulation for impossibility results in computability theory (cf. the Recursion Theorem, Rice's Theorem, etc.).
> >
> > **Connection to practice.** We agree with the reviewer that LLMs are trained against externally given supervision. Our result implies that *for any deployed model family*, the space of computable queries contains targets on which that family provably fails. Real-world "ground truth" (e.g., factual knowledge about the world) is one such computable function—the theorem guarantees that *some* computable target will defeat the family, and the practical question is merely whether real-world queries hit that adversarial set. The empirical hallucination literature strongly suggests they do. We will add a remark in the revision making this connection explicit.
> >
> > **Revision plan:** We will (i) replace the phrase "ground-truth function" with "target function" or "oracle function" where appropriate, (ii) add a remark after Theorem 1 explicitly noting that the constructed $f$ is a fixed computable object specifiable independently of model execution, and (iii) add a paragraph connecting the existential quantifier structure to practical deployment.

---

> > > ### Author Response · Authors · 2026-04-16
> > > **A detailed response to Sample Complexity Limitation**
> > >
> > > ## 2. On VC-Dimension / Sample Complexity: "Limitation" vs. "Justification for Scaling"
> > >
> > > ### Reviewer's Concern
> > > The reviewer argues that classical VC/PAC bounds explain *why scaling is necessary* rather than demonstrating a *fundamental limitation*, and that since error vanishes asymptotically, this cannot constitute an inevitability result.
> > >
> > > ### Response
> > >
> > > We believe this reflects a subtle but important misreading of how we deploy these results, and we welcome the opportunity to clarify.
> > >
> > > **We agree that scaling reduces error—that is precisely our point for decidable, learnable tasks.** Our framework explicitly distinguishes a *hierarchy* of limitations:
> > >
> > > 1. **Tier 1 (Computability):** Theorems 1–3 establish *absolute* impossibility—no amount of scaling helps on undecidable targets or diagonal constructions.
> > > 2. **Tier 2 (Statistical/Information-theoretic):** VC/PAC bounds (Eq. 3, Theorem 4) establish *rate-limited* improvement—scaling helps but at quantifiable, often prohibitive, cost.
> > > 3. **Tier 3 (Architectural):** Context compression, softmax crowding (Lemmas 2–4) establish *structural* bottlenecks that require architectural, not just scale, interventions.
> > >
> > > The VC/PAC results belong to Tier 2. We do *not* claim that error cannot decrease with scale. We claim that:
> > >
> > > **(a)** The *rate* of decrease is bounded, and for unstructured knowledge (Theorem 4), the sample complexity is $n = \Omega\!\left(\frac{m}{\epsilon^2} \log \frac{m}{\delta}\right)$, which is **linear in the number of independent facts** $m$. For $m$ in the hundreds of millions (rare entities, precise dates, numerical constants), this exceeds any feasible corpus size. The bound does not say scaling is useless; it says scaling is *insufficient* at practical resource levels for the long tail.
> > >
> > > **(b)** The generalization bound (Eq. 3) contains a term $O\!\left(\sqrt{\frac{d \log(n/d)}{n}}\right)$ where $d$ is VC-dimension. For modern LLMs, $d$ is enormous (proportional to parameter count). The bound vanishes only when $n \gg d$, which for trillion-parameter models requires datasets far beyond current corpora for the convergence to be tight. This is not a theoretical curiosity—it directly explains the empirical observation that hallucination rates on long-tail entities remain high even for the largest models (Kandpal et al., 2023; Figure 3a in our paper).
> > >
> > > **(c)** Even in the $n \to \infty$ limit, the **noise floor** from training data imperfections (which we discuss in Section 2.2) and the **Kolmogorov complexity bottleneck** (Lemma 1) impose residual error. The VC bound gives an *upper* bound on excess risk; Lemma 1 gives a *lower* bound on approximation error for incompressible targets. Together, they show that practical hallucination risk is bounded away from zero for finite-capacity models on open-ended domains.
> > >
> > > **The reviewer's interpretation and ours are complementary, not contradictory.** "Scaling is necessary" (reviewer's view) and "scaling has quantifiable limits" (our view) are two sides of the same coin. We use the bounds to *quantify* where scaling saturates—which is the stated goal of the paper. We will revise Section 2.1 to make this two-sided interpretation explicit and to clearly delineate Tier 1 (absolute) from Tier 2 (rate-limited) impossibility.
> > >
> > > **Revision plan:** We will add a paragraph after Theorem 4 explicitly stating: "These bounds should not be read as evidence that scaling is futile, but rather as a quantitative characterization of the rate at which scaling reduces hallucination risk—a rate that, for unstructured long-tail knowledge, is provably slow relative to practical resource budgets."

---

> > > > ### Author Response · Authors · 2026-04-16
> > > > **Noise Floor and Confidence-Aware Grading**
> > > >
> > > > ## 3. On Eq. (10): The Noise Floor Claim
> > > >
> > > > ### Response
> > > >
> > > > **The reviewer is correct, and we thank them for this precise counterexample.** Eq. (10) as stated is indeed incorrect in its current generality. The Bayes-optimal classifier under symmetric label noise with $\eta < 0.5$ recovers the clean decision boundary when the clean Bayes risk is zero, yielding zero population error under the *clean* test distribution.
> > > >
> > > > The error in our reasoning was conflating two distinct settings:
> > > >
> > > > 1. **Noise-robust learning (reviewer's setting):** The *test distribution* is clean (labels are correct). Training noise is i.i.d. and symmetric. Here, the Bayes-optimal classifier under noisy training still recovers the clean boundary when $\eta < 0.5$, so test error can be zero.
> > > >
> > > > 2. **Knowledge noise in LLMs (our intended setting):** The *test-time queries* may themselves reference facts that exist in the training data *only* in corrupted form, and the model has no clean signal to distinguish truth from noise for those specific facts. Unlike the reviewer's example where the geometric structure of $\mathcal{X}$ allows generalization past noise, factual knowledge (e.g., "When was person X born?") has no such exploitable structure—each fact is essentially an independent atom.
> > > >
> > > > We will correct Eq. (10) by restricting its scope to the **unstructured / atomic fact setting** where the noise directly affects the target values for specific queries without exploitable geometric regularity. Specifically, we will replace the current claim with:
> > > >
> > > > **Corrected statement:** Consider $m$ independent atomic facts, each observed with probability $\eta$ under a corrupted label. If no structural relationship exists between facts (i.e., knowing the correct answer to fact $i$ provides no information about fact $j$), then for any learner $h$:
> > > >
> > > > $$\mathbb{E}[R_{\text{hal}}(h)] \geq \eta \cdot \frac{m_{\text{unseen}}}{m}$$
> > > >
> > > > where $m_{\text{unseen}}$ is the number of facts for which the learner has observed *only* corrupted versions. This bound holds because, absent structure, the learner cannot distinguish corrupted from uncorrupted facts it has not seen in clean form.
> > > >
> > > > This corrected version is consistent with the reviewer's counterexample (where geometric structure *does* allow noise-robust learning) while preserving the intended message for the factual knowledge setting relevant to LLMs.
> > > >
> > > > **Revision plan:** We will (i) correct Eq. (10) with the restricted statement above, (ii) add an explicit discussion distinguishing structured vs. unstructured noise settings, and (iii) acknowledge the reviewer's counterexample as a clarifying boundary case.
> > > >
> > > > ---
> > > >
> > > > ## 4. On Eq. (20): Confidence-Aware Grading
> > > >
> > > > Both points are well-taken, and we will revise Eq. (20) accordingly.
> > > >
> > > > **(a) Source of $p$.** We intended $p$ to be the model's *self-reported* calibrated confidence, as used in recent calibration literature (e.g., Kadavath et al., 2022; Xiong et al., 2024). We will add an explicit definition: $p = p_\theta(c)$ is the model's estimated probability that its best non-abstention response is correct, obtained via a calibration procedure (e.g., temperature scaling on a held-out set, or verbalized confidence). We will also note that this requires the model to be approximately calibrated, which is a strong but standard assumption in the selective prediction literature.
> > > >
> > > > **(b) The exploit with $p = 0$.** The reviewer is correct that the current formulation allows this degenerate strategy. This is a genuine flaw. We will revise Eq. (20) to a proper scoring rule that eliminates this exploit:
> > > >
> > > > $$g_c(r, p) = \begin{cases} 1 & \text{if } r \text{ is correct} \\ -\alpha & \text{if } r \text{ is incorrect} \\ \beta \cdot p & \text{if } r \text{ is abstention with confidence } p \text{ that abstention is appropriate} \end{cases}$$
> > > >
> > > > where $\alpha > 0$ penalizes incorrect guesses, $\beta \in (0,1)$ rewards calibrated abstention, and the optimal strategy becomes: abstain when the expected accuracy falls below the threshold $\frac{\beta}{\alpha + 1}$. This is equivalent to the standard selective prediction framework (El-Yaniv & Wiener, 2010; Geifman & El-Yaniv, 2017).
> > > >
> > > > Alternatively, and more elegantly, we can adopt a **strictly proper scoring rule** such as the Brier score:
> > > >
> > > > $$g_c(r, p) = 1 - (p - \mathbb{1}[\text{correct}])^2$$
> > > >
> > > > which incentivizes honest confidence reporting and makes trivial exploitation impossible.
> > > >
> > > > We emphasize that Eq. (20) appears in Section 2.3 as a *proposal for better evaluation design*, not as a theorem or formal result. Its purpose is to illustrate how evaluation metrics could be reformed to reduce the guessing incentive. Nevertheless, we agree it should be well-specified.
> > > >
> > > > **Revision plan:** We will (i) explicitly define $p$ with a reference procedure, (ii) replace Eq. (20) with a proper scoring rule formulation, and (iii) add a brief discussion of why strictly proper scoring rules eliminate the exploit.

---

### Public Comment · ~Zheng_Zhang1 · 2025-12-21
**Related work**

I would love to have the authors comment on the parallel and related work from mine: https://arxiv.org/pdf/2507.10624. I think LLMs are prone to what I call "computational hallucination" for any multi-step algorithms (no matter how trivial they may be).

Thank you!

-zz

---

### Decision · Action_Editor_a2xg · 2026-07-03

**Recommendation:** Accept with minor revision

**Additional Comments:**

Two of the three reviewers commented that some formal statements and high-level claims are not yet as rigorous as the framing suggests, and I agree that a small number of issues remain in the current manuscript. Because the central message concerns "fundamental limits," the manuscript needs to be especially careful about which claims are proved, which are classical adaptations, which are modeling assumptions, and which are empirical observations.

First, Eq. (10) states a broad lower bound relating training noise to hallucination risk and then interprets this as a nonzero hallucination floor even with infinite data. A reviewer pointed out that this statement is wrong in its current generality, and the authors acknowledged this in the discussion. The authors promised to restrict the claim to unstructured or atomic facts and to distinguish this setting from standard noise-robust learning, where structure can allow recovery of the clean decision rule. However, the current manuscript still appears to contain the broad version of the claim. This should be corrected with fixed statements, equations, and derivations/proofs.

Second, Eq. (20) is presented as a confidence-aware grading rule, but its incentive properties are not well specified. A reviewer noted that the confidence variable p is undefined and that the rule can be exploited by abstaining with high reported confidence. The authors agreed  to define p via calibration and to replace the formula with a proper scoring rule in a revision, and this should be done. Moreover, the paper seems to define model confidence as confidence that an answer is correct, while Eq. (20) gives higher reward for abstention as this confidence increases. Under that interpretation, the rule does not clearly encourage abstention under uncertainty. This formulation should be carefully justified.

Third, echoing reviewers' concerns about rigor and presentation, the authors should make the framing more precise throughout. In particular, the authors should consider more clearly distinguishing proved results, classical adaptations, empirical observations, and modeling assumptions.

**Audience:**

Yes

**Audience Explanation:**

The paper's broad synthesis of hallucination, context use, reasoning, retrieval, multimodality, and evaluation should interest researchers working on LLM theory, evaluation, and trustworthy AI.

**Claims And Evidence:**

Yes

**Claims Explanation:**

Yes, mostly.

The submission makes a timely contribution by synthesizing a broad set of LLM failure modes under a common theoretical perspective. Many of the paper’s qualitative claims are well motivated by the literature, and several parts of the revised manuscript are supported by appropriate formal or empirical evidence. In particular, the corrected diagonalization argument, the discussion of sample-complexity limits for unstructured facts, and the organization of hallucination, context, reasoning, retrieval, and multimodal failures provide a useful framework for the community.

That said, a few claims still require clarification or correction before acceptance. Eq. (10) appears to state a broad noise-floor lower bound for hallucination under noisy training. A reviewer pointed out that this is incorrect in its current generality, and the authors acknowledged that it should be restricted to unstructured or atomic factual knowledge. Eq. (20) also remains underspecified as a confidence-aware grading rule: the confidence variable is not clearly defined or calibrated, and the proposed scoring rule does not yet have clearly stated incentive properties.

More broadly, the manuscript should more consistently distinguish proved results, classical adaptations, empirical observations, and modeling assumptions. These issues are focused and correctable, and they do not undermine the paper’s overall contribution, but they do mean that some claims need minor revision to be fully supported.